# CAUSAL DISCOVERY IN THE WILD: A VOTING-THEORETIC ENSEMBLE APPROACH

**Vy Vo**[1,2]  **Haoxuan Li**[3,4,5]  **Mingming Gong**[6,7]

[1]Monash University    [2]CSIRO's Data61
[3]Peking University    [4]University of Oxford    [5]Institute for Decentralized AI
[6]The University of Melbourne    [7]Mohamed bin Zayed University of Artificial Intelligence

## ABSTRACT

Causal discovery is a critical yet persistently challenging task across scientific domains. Despite years of significant algorithmic advances, existing methods still struggle with inconsistent outcomes due to reliance on untestable assumptions, sensitivity to data perturbations, and optimization constraints. To this end, ensemble-based causal discovery has been actively pursued, aiming to aggregate multiple structural predictions for increased stability and uncertainty estimation. However, current aggregation methods are largely heuristic, lacking theoretical guarantees and guidance on how ensemble design choices affect performance. This work is proposed to address there fundamental limitations. We introduce a principled voting-based framework for structural ensembling, establishing conditions under which the aggregated structure recovers the true causal graph. Our analysis yields a theoretically justified weighted voting mechanism that informs optimal choices regarding the number, competency, and diversity of causal discovery experts in the ensemble. Extensive experiments on synthetic and real-world datasets verify the robustness and effectiveness of our approach, offering a rigorous alternative to existing heuristic ensemble methods.

## 1 INTRODUCTION

Discovering the causal relationships among factors in a system is an active research interest across scientific disciplines. A plethora of causal discovery algorithms have been proposed over the years, which however tend to produce different graph predictions when applied to the same dataset. This is mainly because methods often rely on strong assumptions that are rarely testable in practice, while diagnostic tools for detecting violations of these assumptions are scarce (Prakash et al., 2024). Even algorithms with formal guarantees can suffer from various other issues such as sensitivity to data perturbation, finite-sample estimation errors or inherent constraints in the optimization procedures due to the combinatorial search space of graphs. This consequently complicates the selection of algorithms for real-world problems when the true causal structure is unknown.

In response to these multifaceted challenges, ensemble-based causal discovery has emerged as a promising solution. This approach essentially aims to aggregate multiple output structures, either from heterogenous algorithms or from repeated runs of the same one on different data partitions, into a single consensus graph. Structural ensembles help mitigate data variability and algorithmic instability, while offer scalability benefits for large datasets (Tang et al., 2019; Guo et al., 2021; Sun et al., 2026). Ensemble learning is also known to effectively capture epistemic uncertainty (Kendall & Gal, 2017; Hüllermeier & Waegeman, 2021)[1]. In this context, it pertains to assessing the reliability of inferred causal structures, allowing practitioners to distinguish between confident causal links and more tentative ones that may require additional investigation. This approach to uncertainty quantification is well-suited for frequentist algorithms that produce point estimates of the graph, offering an alternative to Bayesian causal discovery approaches, which natively account for uncertainty yet relying on strong parametric assumptions to compute graph posteriors (Lorch et al., 2021; Annadani et al., 2023).

---

[1]uncertainty due to a lack of knowledge about the true model, which can be reduced with extra information

Despite the advantages of ensemble-based approaches, existing structural aggregation techniques remain largely heuristic and under-theorized. These methods can be broadly categorized as *distance-based* (Wang & Peng, 2014; Malmi et al., 2015) or *voting-based* (Tang et al., 2019; Zhang et al., 2023). The former line of works treat graph aggregation as a clustering problem, searching for a "centroid" graph of minimal distance to all structures in the ensemble, whereas the latter view algorithms in an ensemble as voters, wherein weighted voting rules are designed to score individual structures and obtain the collective decision based on some decision thresholds. In either approach, the selection of distance measures or weighting schemes are made without theoretical justification as to how well the ensembled graph approximates the true causal structure. Furthermore, there is no principled understanding of how key factors such as the size of ensemble, the diversity or performance of individual voters influence the accuracy of the aggregated graph. As a result, practitioners are left without guidance on how to configure ensemble strategies for optimal performance.

Another source of uncertainty that is often overlooked in the existing literature is the selection of the set of graphs used for aggregation. Typically, graph collections are generated from fixed data partitions or limited runs of causal discovery algorithms. This raises a fundamental question: how stable are the ensemble results given variations in the input graph composition? This oversight, coupled with the heuristic nature of current aggregation methods, has undermined the reliability of ensemble-based causal discovery in real-world applications.

**Contributions.** In this work, we address these key limitations in the literature by introducing a principled voting-based framework for ensembling the structures from heterogeneous causal discovery algorithms. We characterize the generative process underlying given voting profiles within a structural ensemble and formally analyze the conditions under which the aggregated structure aligns with the true causal graph. The analysis enables a theoretically grounded weighted voting mechanism that informs optimal design choices for the size, competency and diversity levels of causal discovery experts. We validate the theoretical advantages as well as the practical robustness and effectiveness of our approach through extensive experiments on synthetic and real-world systems[2].

## 2 PRELIMINARIES

**Notation.** We use upper case letters ($X$) to denote random variables, lower case letters ($x$) for values and bold lower case ($\boldsymbol{x}$) for vectors. We reserve calligraphic letters ($\mathcal{X}$) for spaces/sets, italic bold capital ($\boldsymbol{X}$) for matrices and roman bold capital ($\mathbf{G}$) for notations related to graphs. Finally, we use $[d]$ to denote a set of integers $\{1, 2, \cdots, d\}$.

**Structural Causal Model.** A directed graph $\mathbf{G} = \langle \mathbf{V}, \mathbf{E} \rangle$ consists of a set of (ordered) nodes $\mathbf{V}$ and an edge set $\mathbf{E} \subseteq \mathbf{V}^2$ of pairs of nodes with $(v, v) \notin \mathbf{E}$ for any $v \in \mathbf{V}$ (one without self-loops). For a pair of nodes $u, v$ with $(u, v) \in \mathbf{E}$, there is an arrow pointing from $u$ to $v$ and we write $u \rightarrow v$. Two nodes $u$ and $v$ are adjacent if either $(u, v) \in \mathbf{E}$ or $(v, u) \in \mathbf{E}$. If there is an arrow from $u$ to $v$ then $u$ is said to be a parent of $v$ and $v$ is a child of $u$. Let $\mathrm{pa}_v$ and $\mathrm{ch}_v$ respectively denote the parents and children of a node $v$ in $\mathbf{G}$. A *directed acyclic graph* (DAG) is a directed graph without cycles.

Let $X_{\mathrm{pa}_i}$ be the set of direct causes of $X_i$ (so-called *parents*), and $\{U_i\}_{i \in [d]}$ be mutually independent noise variables with strictly positive density. The data-generating process for a set of $d$ random variables $X = \{X_i\}_{i \in [d]}$ is characterized by a *structural causal model* (SCM, Pearl, 2009), which is defined over a tuple $\langle U, X, \mathcal{F} \rangle$ and consists of a sets of assignments

$$X_i := f_i\big(X_{\mathrm{pa}_i}, U_i\big), \quad i \in [d]. \tag{1}$$

The set of causal mechanisms $\mathcal{F} = \{f_i\}_{i \in [d]}$ deterministically assign values to each $X_i$ respectively given its causes and the corresponding error term $U_i$. An SCM without feedback loops gives rise to a DAG structure $\mathbf{G}$ where each variable $X_i$ corresponds to the $i$-th node in the graph. The recursive application of (1) induces a joint probability distribution over $X$, denoted as $\mathbb{P}(X)$ such that the Markov factorization holds: $\mathbb{P}(X) = \prod_{i \in [d]} \mathbb{P}(X_i | X_{\mathrm{pa}_i})$.

**Optimal Transport.** Let $\alpha = \sum_{j=1}^m a_j \delta_{x_j}$ be a discrete measure with weights $\boldsymbol{a}$ and particles $x_1, \cdots, x_m \in \mathcal{X}$ and $\beta = \sum_{j=1}^m b_j \delta_{y_j}$ be another discrete measure defined similarly. The Kan-

---

[2]Codes are released at `github.com/isVy08/Causal-Bayes-Ensemble`

torovich (1960) formulation of the optimal transport (OT) distance between $\alpha$ and $\beta$ is

$$W_c(\alpha, \beta) := \min_{\mathbf{P} \in \mathbb{U}(\boldsymbol{a}, \boldsymbol{b})} \langle \mathbf{C}, \mathbf{P} \rangle, \tag{2}$$

where $\langle \cdot, \cdot \rangle$ denotes the Frobenius dot-product; $\mathbf{C} \in \mathbb{R}_+^{m \times m}$ is the cost matrix of the transport; $\mathbf{P} \in \mathbb{R}_+^{m \times m}$ is the transport matrix/plan; $\mathbb{U}(\boldsymbol{a}, \boldsymbol{b}) := \left\{ \mathbf{P} \in \mathbb{R}_+^{m \times m} : \mathbf{P}\mathbf{1}_m = \boldsymbol{a}, \mathbf{P}\mathbf{1}_m = \boldsymbol{b} \right\}$ is the transport polytope of $\boldsymbol{a}$ and $\boldsymbol{b}$; $\mathbf{1}_m$ is the $m-$dimensional column of vector of ones. Given a measurable map $T : \mathcal{X} \mapsto \mathcal{Y}$, $T\#\alpha$ denotes the push-forward measure of $\alpha$ which satisfies $T\#\alpha(B) = \alpha(T^{-1}(B))$ for any measurable set $B \subset \mathcal{Y}$. For discrete measures, the push-forward operation is equivalent to $T\#\alpha = \sum_{j=1}^m a_j \delta_{T(x_j)}$.

# 3 STRUCTURE LEARNING FROM NOISY EXPERTS

Suppose that a set of algorithms (so-called "experts") of size $n$ are selected to infer the causal graph $G$ underlying a system of $d$ variables. We use $\mathcal{D}$ to represent some fixed input accessible to all experts. For statistical causal discovery experts, which is the focus in our paper, $\mathcal{D}$ refers to a collection of data samples. In the general case with diverse types of experts, $\mathcal{D}$ can be arbitrary e.g., the combination of numerical data and semantic description of the variables, the latter of which is typically processed by human or LLM-based experts.

Let $\mathcal{G}$ denote the space of possible DAG structures. In the general case, $|\mathcal{G}|$ can be as large as $2^{\Theta(d^2)}$. However, our analysis here does not require $\mathcal{G}$ to contain all possible configurations; rather it is sufficient to consider purely non-degenerate structures i.e., $\mathcal{G} := \{\mathbf{G} : \mathbb{P}(G = \mathbf{G} \mid \mathcal{D}) > 0\}$, where $\mathbb{P}(G \mid \mathcal{D})$ is the hidden prior distribution. Let the graph space $\mathcal{G}$ be arbitrarily indexed $1, 2, \cdots |\mathcal{G}|$, with $\mathbf{G}_k$ denoting the graph realization associated with an index $k$. Let $\widetilde{G}_i \in \mathcal{G}$ denote the random variable associated with the estimated graph from an $i$-th expert. Each estimation $\widetilde{G}_i$ can be viewed as a noisy version of the true graph.

We then define for every $i$-th expert a fixed (unknown) **competence transition matrix** $\boldsymbol{T}_i := \left[ \mathbb{P}(\widetilde{G}_i = \mathbf{G}_j \mid G = \mathbf{G}_k, \mathcal{D}) \right]_{\mathbf{G}_j, \mathbf{G}_k \in \mathcal{G}}$. These quantities encode the noise rates of the expert's prediction, which are also indicators of the expert's **competence** in predicting the causal graph. We denote $\boldsymbol{T}_{i,|\mathbf{G}_j} \triangleq \boldsymbol{T}_i[\mathbf{G}_j, :]$ as the $j$-th row vector, $\boldsymbol{T}_{i,\mathbf{G}_j|} \triangleq \boldsymbol{T}_i[:, \mathbf{G}_j]$ as the $j$-th column vector, and $\boldsymbol{T}_{i,\mathbf{G}_k|\mathbf{G}_j} \triangleq \boldsymbol{T}_i[\mathbf{G}_j, \mathbf{G}_k]$ as the $k$-th entry in the $j$-th row.

Without loss of generality, we assume the correct structure corresponds to index 1. Hence, the generative distribution for $\widetilde{G}_i$ is determined by $\mathbb{P}(\widetilde{G}_i \mid G = \mathbf{G}_1, \mathcal{D})$. We denote $p_{i,\mathbf{G}_1} := \boldsymbol{T}_{i,\mathbf{G}_1|\mathbf{G}_1}$ as the probability that the $i$-th expert makes a *correct* prediction given data $\mathcal{D}$ and oppositely, $q_{i,\mathbf{G}_j|\mathbf{G}_1} := \boldsymbol{T}_{i,\mathbf{G}_j|\mathbf{G}_1}, \forall j > 1$ as the probability that expert $i$ makes an *incorrect* prediction.

## 3.1 RECOVERY OF CAUSAL STRUCTURE VIA BAYES VOTING RULE

Essentially, we have a body of $n$ decision-making experts, each voting for one alternative $\widetilde{G}_i$ out of $|\mathcal{G}|$ possible options. The voting setup can be cast as a general problem of multi-class classification with unbalanced truth over an exponentially large space of $|\mathcal{G}|$ classes. Thus, recovering the true causal structure in our setting is equivalent to finding the optimal decision rule mapping from the data of $n$ noisy predictions to the ground-truth graph label, a classic unsupervised learning problem now placed over a combinatorially large label space typical of causal discovery settings. We first introduce our proposed vote aggregation rule and establish the conditions for recovering the true causal structure from its noisy predictions.

Formally, a voting rule refers to a function $g(\cdot)$ mapping a set of votes $\widetilde{G} = \{\widetilde{G}_i\}_{i \in [n]}$ to an estimate of the true state of $G$. In this work, we consider a **weighted linear voting rule** that aggregates the choices of the body of $n$ experts into a collective decision by assigning a weighted score to each alternative and selecting the one with the highest score. More concretely, the scoring function for a candidate graph $\mathbf{G} \in \mathcal{G}$ is given as

$$S_{n, \boldsymbol{w}_{\mathbf{G}}}(\widetilde{G}) = \sum_{i=1}^n w_{i,\mathbf{G}} \, \mathbf{1}\left[\widetilde{G}_i = \mathbf{G}\right] + b_{\mathbf{G}}, \tag{3}$$

where $\boldsymbol{w}_{\mathbf{G}} = [w_{1,\mathbf{G}}, \cdots, w_{n,\mathbf{G}}] \in \mathbb{R}^n$ are weights assigned to each expert reflecting their voting power or reliability, and $b_{\mathbf{G}} \in \mathbb{R}$ is the bias term towards a candidate solution $\mathbf{G}$. The predicted structure $\widehat{\mathbf{G}}$ from this voting rule is given as $\widehat{\mathbf{G}} = \arg\max_{\mathbf{G} \in \mathcal{G}} S_{n,\boldsymbol{w}_{\mathbf{G}}}(\widetilde{G})$.

The rule reduces to **plurality voting**[3] in the unweighted case i.e., $\forall \mathbf{G}, \mathbf{G}' \in \mathcal{G}$: $\boldsymbol{w}_{\mathbf{G}} = \boldsymbol{w}_{\mathbf{G}'}$ and $b_{\mathbf{G}} = b_{\mathbf{G}'}$. For brevity, we define $\pi_{\mathbf{G}} := \mathbb{P}(G = \mathbf{G} \mid \mathcal{D})$.

It is a well-established result (Nitzan & Paroush, 1982) that the error-minimizing decision rule corresponds to **Bayes voting rule**, where $S_{n,\boldsymbol{w}_{\mathbf{G}}}(\widetilde{G}) := \log \mathbb{P}(G = \mathbf{G} \mid \widetilde{G}, \mathcal{D})$ where the weight of each expert $i$ and the bias in (3) are given by:

$$w_{i,\mathbf{G}} = \log p_{i,\mathbf{G}} - \log q_{i,\widetilde{G}_i|\mathbf{G}}, \quad b_{\mathbf{G}} = \sum_{i=1}^{n} \log q_{i,\widetilde{G}_i|\mathbf{G}} + \log \pi_{\mathbf{G}}. \tag{4}$$

See the detailed derivation in Appendix E.1. For the Bayes' voting rule to be defined, we adopt the convention that $w_{1,\mathbf{G}}, \cdots, w_{n,\mathbf{G}} = b_{\mathbf{G}} = 0$ where $\pi_{\mathbf{G}} = 0$. A question of utmost importance is to what extent the collective decision under Bayes voting rule coincides with the true graph, that is $\widehat{\mathbf{G}} = \mathbf{G}_1$. We now study the conditions under which this can be achieved, which first require the following assumptions.

**Assumption 1** (Independence of Experts' Decisions). *Experts in a decision-making body make mutually independent decisions conditioned on the truth and the input $\mathcal{D}$.*

**Assumption 2** (Bounded Competency). *There exists a constant $\tau < +\infty$ such that $\forall i \in [n]$ : $\|\log \boldsymbol{T}_i\|_{\max} \leq \tau$.*

**Condition 1** (Informativeness). *An $i$-th expert in the decision-making body is said to be informative if $\boldsymbol{T}_{i,|\mathbf{G}_k} \neq \boldsymbol{T}_{i,|\mathbf{G}_j}, \forall k \neq j$.*

A concept critical to our analysis is **informativeness**, which refers to the ability to distinguish between any two structures $\mathbf{G}_j$ and $\mathbf{G}_k$. Here the degree of informativeness of an $i$-th expert between two options $j \neq k$ is quantified by $\mathrm{KL}\left(\boldsymbol{T}_{i,|\mathbf{G}_j}; \boldsymbol{T}_{i,|\mathbf{G}_k}\right)$, where "being non-informative" is equivalent to $\boldsymbol{T}_{i,|\mathbf{G}_j} = \boldsymbol{T}_{i,|\mathbf{G}_k}$. Finally, we assume that the truth is biased in the sense that $\pi_{\mathbf{G}_1} \geq \pi_{\mathbf{G}_j}, \forall j > 1$. As to be shown in the proof, this condition is not necessary and the result still holds as long as the total informativeness level outweighs the log-odds $\pi_{\mathbf{G}_1}/\pi_{\mathbf{G}_j}$.

**Theorem 1** (Optimality under Bayes Voting). *Under Assumptions 1 and 2, the collective decision from a group of $n$ informative experts is correct under Bayes voting rule with probability at least*

$$1 - \sum_{j \geq 2} \exp\left(-n \cdot \overline{\mathrm{KL}}_{1,j}^2 \Big/ 2\tau^2\right), \tag{5}$$

*where $\overline{\mathrm{KL}}_{1,j} = n^{-1} \sum_{i=1}^{n} \mathrm{KL}\left(\boldsymbol{T}_{i,|\mathbf{G}_1}; \boldsymbol{T}_{i,|\mathbf{G}_j}\right), \forall j \in \{2, \cdots, |\mathcal{G}|\}$.*

The proof is provided in Appendix E.1. Theorem 1 provides an asymptotic guarantee for the recovery of the true graph under Bayes voting rule. It posits that the probability for the collective decision obtained from a group of informative experts to be erroneous decays exponentially in the group size and the average level of informativeness. For a finite group size $n$, the more informative each expert in the group about the true structure $\mathbf{G}_1$, the faster the group arrives the correct decision.

**Definition 1** (Identifiability from Noisy Structures). *A causal graph $\mathbf{G}$ is said to be identifiable from input $\mathcal{D}$ and a set of noisy structures $\widetilde{G} = \{\widetilde{G}_1, \widetilde{G}_2, \cdots\}$ if there exists no other graph $\mathbf{G}'$ such that $\mathbb{P}(\widetilde{G} \mid \mathbf{G}, \mathcal{D}) = \mathbb{P}(\widetilde{G} \mid \mathbf{G}', \mathcal{D})$.*

By Definition 1, Condition 1 is essentially a sufficient condition for the identifiability of causal structure from noisy counterparts. It is worth noting that informativeness is a weak condition since it only requires every two rows in any matrix $\boldsymbol{T}_i$ are distinct element-wise, which holds almost surely[4] under usual continuous models. In fact, the result in Theorem 1 even allows for some, ideally in small number, non-informative experts in the group. More interestingly, it does not require the graph to be identifiable from any given data samples (c.f Definition 2 in Appendix A.).

---

[3]The more familiar term specific to the setting of binary choices is *majority voting*.
[4]in measure-theoretic sense; see discussion in Appendix B.

## 3.2 Recovery via Aggregation of Noisy Sub-structures

The problem is now reduced to estimating the prior $\boldsymbol{\pi}$ and competence transition probabilities $\{\boldsymbol{T}_i\}_{i=1}^n$ of a selected group of experts. The caveat however is that we are dealing with an exponential space of states, thus estimating these quantities of ultra-high dimensionality is computationally infeasible. In this section, we propose a strategy to decompose the graphs into sub-structures that can be aggregated through Bayes voting with more feasible state spaces.

**Units of Aggregation.** Given a set of nodes $\mathbf{V} = \{v_1, v_2, \cdots\}$, we define a **feature level** as an abstract type of sub-structure determined by a particular of grouping of nodes in $\mathbf{V}$. A **feature space** $\Omega(\mathbf{V})$ (or $\Omega$ in short) is the set of possible configurations associated with a chosen feature level such that the union over all such groupings spans the set $\mathbf{V}$ i.e., $\bigcup_{\omega \in \Omega}\{\omega\} = \mathbf{V}$, where each element $\omega \in \Omega$ is referred to as a **feature**[5]. For example, a basic feature level is *node pairs*, where $\Omega = \{(v_i, v_j) : 1 \leq i < j \leq |\mathbf{V}|\}$ and an element $(v_i, v_j) \in \Omega$ is a feature. Feature levels represent the granularity at which features are considered and define the basic units of analysis for the graph. Higher-order levels may be of interest such as *node triplets* $(v_i, v_j, v_k)$ or *quadruples* $(v_i, v_j, v_k, v_l)$.

Given a feature level $\Omega$, each feature $\omega$ is treated as a random element with a finite space $\mathcal{S}(\omega)$ of mutually exclusive connective states among nodes in the grouping ($|\mathcal{S}(\omega)| \geq 2$), with some fixed labelling $1, 2, \cdots, |\mathcal{S}(\omega)|$. For example, one may consider adjacency between a pair of nodes $(v_i, v_j)$ – equivalent to $\mathcal{S} = \{1 : v_i \,\text{---}\, v_j, 2 : \text{no edge}\}$; if one further accounts for orientations, a possible domain is $\mathcal{S} = \{1 : v_i \rightarrow v_j, \ 2 : v_i \leftarrow v_j, 3 : \text{no edge}\}$. Let $\mathcal{S} := \{\mathcal{S}(\omega)\}_{\omega \in \Omega}$ denote the set of state spaces associated with a feature space $\Omega$. A graph structure defined over a tuple $\langle \mathbf{V}, \Omega, \mathcal{S} \rangle$ can be viewed as a joint realization of mutually inclusive events of each $\omega \in \Omega$ being assigned a certain state in its domain $\mathcal{S}(\omega)$. Hence, the set of all possible graphs can be re-defined as $\mathcal{G} := \prod_{\omega \in \Omega} \mathcal{S}(\omega)$.

**Aggregation via Feature-wise Bayes Voting.** Given a tuple $\langle \mathbf{V}, \Omega, \mathcal{S} \rangle$ of feature space and domain under analysis, by construction, there exists a ground-true set of states corresponding to every feature $\omega \in \Omega$. A feature $\omega$ can be viewed as a task subject to decision making, and an expert's prediction of a structure give rises to a set of votes on a multiplicity of tasks. Our problem can thus be treated as a general weighted voting game on a fixed set of tasks with finite choices (Baharad et al., 2011).

For a random feature $\omega \in \Omega$ with domain $\mathcal{S}(\omega)$, let $X_i(\omega) \in \mathcal{S}(\omega)$ be the random variable representing the noisy prediction from the $i$-th expert for feature $\omega$. Let $Y(\omega) \in \mathcal{S}(\omega)$ denote the true state of a feature $\omega$ and $\widehat{Y}(\omega) : (\mathcal{S}(\omega))^n \mapsto \mathcal{S}(\omega)$ be the collective decision from $n$ experts about the state of every feature $\omega \in \Omega$ based on some voting rule i.e., $\widehat{Y}(\omega)$ is a function of $X_1(\omega), \cdots X_n(\omega)$.

Let $\mathbf{G}_1(\Omega, \mathcal{S})$ return the structural pattern (e.g., a skeleton) associated with the true graph $\mathbf{G}_1$ induced by $\langle \Omega, \mathcal{S} \rangle$. Let $\widehat{\mathbf{G}}(\Omega, \mathcal{S})$ be the predicted structure by aggregating the collective decisions on all features $\omega \in \Omega$ over state space $\mathcal{S}$ from $n$ experts.

**Corollary 1.** $\widehat{\mathbf{G}}(\Omega, \mathcal{S})$ *coincides with* $\mathbf{G}_1(\Omega, \mathcal{S})$ *if and only if the voting decisions on every feature* $\omega \in \Omega$ *are correct.*

It immediately follows from the definition that a candidate graph coincides with the true graph if and only if every sub-structure is perfectly aligned. The formal analysis is provided in Appendix E.2, which further shows that the probability of $\widehat{\mathbf{G}}(\Omega, \mathcal{S})$ being correct is upper bounded by the worst-case feature accuracy. This simple result enables us to decompose the structure learning problem into a series of sub-tasks of recovering the true state of every sub-structure $\omega \in \Omega$ from its noisy versions.

By this re-definition of the graph space over $\langle \Omega, \mathcal{S} \rangle$, the prior $\boldsymbol{\pi}$ can be written as the joint distribution over the set of random variables $\{Y(\omega)\}_{\omega \in \Omega}$. Let $\boldsymbol{\pi}(\omega) := [\mathbb{P}(Y(\omega) = j \mid \mathcal{D})]_{j \in \mathcal{S}(\omega)}$ denote the true prior distribution associated with a feature $\omega$, induced by marginalizing $\boldsymbol{\pi}$ over $Y(\omega)$. We assume $\mathcal{S}(\omega)$ spans over the support of $\boldsymbol{\pi}(\omega)$. Similarly, we define $\forall i \in [n] : \boldsymbol{T}_i(\omega) := [\mathbb{P}(X_i(\omega) = k \mid Y(\omega) = j), \mathcal{D}]_{k,j \in \mathcal{S}(\omega)}$ as a result of marginalization on $\boldsymbol{T}_i$ over $\omega$. Under the appropriate conditions on the experts' informativeness about individual feature $\omega$, the result of Bayes voting rule in Theorem 1 directly applies to the task of predicting $Y(\omega)$ over its state space $\mathcal{S}(\omega)$ of reduced cardinality. We

---

[5]The term is inherited from the classic works by Friedman & Koller (2013); Friedman et al. (2013) and to be distinguished with features of data.

here note that an $i$-th expert is defined to be *informative* about a feature $\omega$ if Condition 1 is satisfied w.r.t the marginal transitions $\boldsymbol{T}_i(\omega)$ for any two state options $k, j \in \mathcal{S}(\omega)$.

## 4 ESTIMATING COMPETENCIES OF CAUSAL DISCOVERY EXPERTS

Direct graph-level Bayes voting is inherently intractable since it requires one to handle a $(2^{d^2} \times 2^{d^2})$ competence matrix for every expert. On the other hand, feature-level aggregation begins with a specification of feature level $l$ i.e., number of nodes per grouping of nodes in $\mathbf{V}$. With a careful selection of $l$, the recovery of the true causal graph can now be feasibly achieved by simultaneously recovering each feature's true state via the respective Bayes votes. Concretely, for a graph with fixed $d$ nodes, a choice of level $l$ gives rise to a feature space $\Omega$ of size $\binom{d}{l}$. Suppose that all features in $\Omega$ have a shared space $\mathcal{S}$ of exhaustive $m$ connective states ($m \geq 2$). Feature-level Bayes voting instead requires handling $\binom{d}{l}$ individual $(m \times m)$ matrices, which becomes significantly manageable for reasonably small $(l, m)$.

It now boils down to the estimation of the parameters $\theta$ which consists of the prior $\boldsymbol{\pi}(\omega)$ and the set of transition probabilities $\{\boldsymbol{T}_i(\omega)\}_{i \in [n]}$ for all sub-structures in a pre-defined space $\langle \mathbf{V}, \Omega, \mathcal{S} \rangle$:

$$\theta := \big\{\theta(\omega)\big\}_{\omega \in \Omega}, \quad \theta(\omega) := \big\{\boldsymbol{\pi}(\omega), \boldsymbol{T}_1(\omega), \cdots, \boldsymbol{T}_n(\omega)\big\}.$$

In practice, no estimation procedure is perfect and the estimates would be inevitably noisy to some extent. Before explaining how to estimate the parameters, we study the conditions under which Bayes voting remains optimal when applied to noisy parameter estimates. It is easy to see from Eq. (3) that there is still a chance for the experts to predict correctly as long as the score assigned to the true state remains maximal. Proposition 1 characterizes the probability of having a correct collective decision in this scenario, which will be called a "noisy" Bayes voting rule.

The following analyses apply to arbitrary feature $\omega \in \Omega$ with state space $\mathcal{S}(\omega)$. Let $\widetilde{\boldsymbol{\pi}}(\omega)$ and $\widetilde{\boldsymbol{T}}_i(\omega)$ represent some noisy estimates of $\boldsymbol{\pi}(\omega)$ and $\boldsymbol{T}_i(\omega)$ for a certain feature $\omega \in \Omega$. For ease of exposition, we again label 1 as the correct state of the feature $\omega$ under analysis.

**Proposition 1** (Robustness of Bayes Voting). *Suppose the estimates $\{\widetilde{\boldsymbol{T}}_i(\omega)\}_{i=1}^n$ satisfy Assumption 2 w.r.t some constant $\widetilde{\tau} < +\infty$, then under Assumption 1, the collective decision from a group of $n$ experts for a feature $\omega$ is correct under "noisy" Bayes voting rule with probability at least*

$$1 - \sum_{j \geq 2} \exp\left[-\Theta\left(n \cdot \Delta\overline{\mathrm{KL}}_{1,j}^2(\omega)\Big/2\widetilde{\tau}^2\right)\right], \tag{6}$$

*as long as $\Delta\overline{\mathrm{KL}}_{1,j}(\omega) \gg n^{-1/2}$ and $\log\left(\widetilde{\pi}_1(\omega)/\widetilde{\pi}_j(\omega)\right) = \mathcal{O}(1)$ where $\Delta\overline{\mathrm{KL}}_{1,j}(\omega) = n^{-1}\sum_{i=1}^n \left[\mathrm{KL}\left(\boldsymbol{T}_{i,|1}(\omega); \widetilde{\boldsymbol{T}}_{i,|j}(\omega)\right) - \mathrm{KL}\left(\boldsymbol{T}_{i,|1}(\omega); \widetilde{\boldsymbol{T}}_{i,|1}(\omega)\right)\right], \forall j \in \{2, \cdots, |\mathcal{S}(\omega)|\}$.*

The proof is provided in Appendix E.3. The probability of converging to the correct decision is thus driven by the deviation of the estimated competencies from the true transition distribution corresponding to the actual state. Specifically, the estimated row of the transition matrix for the true state should remain as close as possible to the ground truth, while the estimated rows for the incorrect options can vary more freely provided that they are sufficiently distinct from the latter. Fortunately, this condition is evaluated upon the total error across all experts, implying that the aggregation process tolerates the presence of bad estimates for some experts as long as the group as a whole remains informative about the true state. If the estimates are perfect, then $\Delta\overline{\mathrm{KL}}_{1,j}(\omega)$ reduces to $\overline{\mathrm{KL}}_{1,j}(\omega) = n^{-1}\sum_{i=1}^n \mathrm{KL}\left(\boldsymbol{T}_{i,|1}(\omega); \boldsymbol{T}_{i,|j}(\omega)\right)$ as equivalently defined in (5).

**Ensemble Design Choices.** We now discuss the key constraints to be considered when designing a feature-level aggregation strategy. With $n$ experts, the total number of parameters for estimation is $\binom{d}{l} \times (nm^2 - (n-1)m - 1)$. The choices of the values for $(l, m, n)$ are thus critical performance factors. It is easy to see that the number of possible states grows super-exponentially with the feature level. Furthermore, an insight from Proposition 1 is that the probability of error is summed over all incorrect states. Since the goal is to have this error probability decay rapidly to zero, it is beneficial to minimize the complexity of the feature space. Regarding the number of experts, such a choice comes

with a trade-off in computational efficiency. On one hand, it is theoretically advantageous to have large $n$ since it would lead to a faster convergence to the correct decision – ideally at an exponential rate should all experts be highly informative. On the other hand, the use of more experts introduce more quantities to be estimated, imposing higher memory and computational demands. Furthermore, there remains a question about the identifiability of the parameters that also involves the decision on number of experts. We provide a detailed discussion on this matter in the next section.

**Parameter Estimation with Optimal Transport.** For a certain sub-structure $\omega$ with unknown true state $Y(\omega) \in [m]$, we wish to estimate the parameters $\theta(\omega)$, which consist of the marginal prior probabilities $\boldsymbol{\pi}(\omega) \in \Delta^{m-1}$ and a fixed set of $n$ competence transition matrices $\{\boldsymbol{T}_i(\omega)\}_{i \in [n]}$ where each $\boldsymbol{T}_i(\omega) \in (\Delta^{m-1})^m$ with $\Delta^{m-1}$ denoting the probability simplex size $m$.

We observe a set of noisy predictions about $\omega$ as $X(\omega) = \{X_i(\omega)\}_{i \in [n]}$ with each $X_i(\omega) \in [m]$. Let $\mathcal{X}(\omega) := [m]^n$ be the space over $X(\omega)$ and $\mathcal{P}(\mathcal{X}(\omega))$ be the set of probability measures on this space. When $m, n$ are finite, we can treat $X(\omega)$ as a discrete random vector taking finite values in $\{1, \cdots, m^n\}$ and distributed according to some true distribution $\mathbb{P}_*(X(\omega) \mid \mathcal{D})$. We consider the parameter space $\Theta(\omega) \subseteq \left(\Delta^{m-1}\right)^{nm+1}$ endowed with a distance function $d_{\Theta}$. Let $\mathcal{M}(\omega) := \{\mathbb{P}_{\theta}(X(\omega) \mid \mathcal{D}) : \mathbb{P}_{\theta} \in \mathcal{P}(\mathcal{X}(\omega)), \theta(\omega) \in \Theta(\omega)\}$ be the statistical models of interest. Suppose that one can repeatedly query from each algorithmic expert and in total obtain $N \in \mathbb{N}$ i.i.d voting profiles $[x_N(\omega)]_{N \geq 1}$, we denote the empirical data distribution as $\mathbb{P}_N(X(\omega) \mid \mathcal{D}) := \left[N^{-1} \sum_{l=1}^N \mathbf{1}[x_l(\omega) = j]\right]_{j \in \mathcal{X}(\omega)}$.

We adopt the classic approach of minimum distance estimation (Bernton et al., 2019), which seeks an estimator over the parameter space $\Theta(\omega)$ that minimizes some distance between the empirical distribution $\mathbb{P}_N$ and the model distribution $\mathbb{P}_{\theta}$. Under the OT distance, as defined in (2) for some cost function $c$, the associated *minimum Kantorovich estimator* (Bassetti et al., 2006) is given as

$$\widehat{\theta}_N(\omega) = \underset{\theta(\omega) \in \Theta(\omega)}{\arg\min} \ W_c\Big[\mathbb{P}_N\left(X(\omega) \mid \mathcal{D}\right); \mathbb{P}_{\theta}\left(X(\omega) \mid \mathcal{D}\right)\Big]. \tag{7}$$

Let $\theta_*(\omega) = \{\boldsymbol{\pi}_*(\omega), \boldsymbol{T}_{*,1}(\omega), \cdots, \boldsymbol{T}_{*,n}(\omega)\}$ denote the true generative parameters that induce $\mathbb{P}_{\theta*}(X(\omega) \mid \mathcal{D}) = \mathbb{P}_*(X(\omega) \mid \mathcal{D})$. An immediate question arises as to whether the parameters $\theta_*(\omega)$ are identifiable from the observed data distribution. Another question is why the OT distance is ideal for our estimation problem. Theorem 2 addresses these two questions by first characterizing the condition for identifying the parameters, then establishing the consistency of the minimum Kantorovich estimator defined in (7). The proof is provided in Appendix E.4.

**Theorem 2** (Consistency). *For any sub-structure $\omega$ with $m$ possible states, let $\widehat{\theta}_N(\omega) = \{\widehat{\boldsymbol{\pi}}_N(\omega), \widehat{\boldsymbol{T}}_{N,1}(\omega), \cdots, \widehat{\boldsymbol{T}}_{N,n}(\omega)\}$ be the minimum Kantorovich estimates satisfying (7). As long as there are at least $(2m - 1)$ informative experts, then under Assumptions 3 and 4,*

*(i) $\widehat{\boldsymbol{\pi}}_N(\omega) \xrightarrow[N \to \infty]{\text{a.s}} \boldsymbol{\pi}_*(\omega)$, and*

*(ii) $\forall i \in [n]$ : there exists a permutation $\sigma_i \in \mathrm{S}_m$ such that $\widehat{\boldsymbol{T}}_{N,i}(\omega) \xrightarrow[N \to \infty]{\text{a.s}} \boldsymbol{P}_{\sigma_i} \boldsymbol{T}_{*,i}(\omega)$,*

*where $\mathrm{S}_m$ is the symmetric group of degree $m$ and $\boldsymbol{P}_{\sigma_i}$ is the permutation matrix associated with $\sigma_i$.*

Here we show that with at least $(2m - 1)$ informative experts, the parameters can be identified up to some row permutations of every matrix $\boldsymbol{T}_{i,*}(\omega)$. In the infinite limit, we can thus achieve the convergence of the estimates $\widehat{\theta}(\omega)$ to an equivalence class of the true parameters under label permutation, with the empirical distribution $\mathbb{P}_N$ approaching $\mathbb{P}_*$ at a rate of $\mathcal{O}(1/\sqrt{N})$ (Agarwal et al., 2019) in the general case. Theorem 2 is adapted from a well-known result regarding the consistency of the minimum Kantorovich estimator in Bassetti et al. (2006), which further requires two regularity assumptions to ensure the cost function is well-behaved. It highlights another design consideration: we should aim to keep the number of states $m$ reasonably small so that consistency can be achieved with a feasible number of experts. It is worth noting this condition is generic to the rank of each matrix $\boldsymbol{T}_i$. Given precise knowledge of the ranks, the number of sufficient experts can even be greatly reduced. Particularly, if all matrices are non-singular, then not only are as few as 3 expert necessary but also sufficient for identification for any $m \geq 2$ (Liu et al., 2023).

There remains a caveat related to the partial identifiability result in Theorem 2. The presence of permutation indeterminacy implies that some state $j$ could be mistaken for the true state under a misaligned permutation. In relation to Proposition 1, this can have an adverse effect on $\Delta\overline{\mathrm{KL}}_{1,j}(\omega)$. To resolve such ambiguity requires additional knowledge beyond informativeness regarding the structure of the experts' competence matrices. A common assumption is that the diagonal entries of each matrix $\boldsymbol{T}_i(\omega)$ are dominant i.e., $\forall i \in [n], \forall k, j \in [m] : \boldsymbol{T}_{i,j|j}(\omega) > \boldsymbol{T}_{i,k|j}(\omega)$. This implies that each expert is is more likely to correctly label the true state than to make an error. Therefore, to obtain a stable solution structure, we impose an inductive bias restricting the search space of $\{\boldsymbol{T}_i(\omega)\}_{i\in[n]}$ to those of diagonally dominance. As long as the majority of experts are truly competent, the robustness of Bayes voting can tolerate a small number of misspecified experts.

Since the goal is to minimize the number of features and state space, we simply focus on edge-level sub-structures ($l = 2$) and consider a shared state space over all features with $m = 3$ mutually exclusive states for each pair of nodes $(v_i, v_j)$ i.e., $\mathcal{S}(v_i, v_j) = \{1 : v_i \rightarrow v_j, \ 2 : v_i \leftarrow v_j, 3 : \text{no edge}\}$. Ultimately, the scope of the parameter estimation task extends to all features and experts. In Appendix B, we consolidate the above discussions into a dedicated section that provides **comprehensive guidelines** for the optimal configurations of an structural ensembling strategy.

While minimum Kantorovich estimators are theoretically appealing, the remaining practical challenge lies in computing the OT distance in Eq. (7). This task can be viewed as a parameter learning problem in a latent variable model, where $X_1(\omega), \ldots, X_n(\omega)$ are the observed noisy label variable and $Y(\omega)$ is the unobserved true label. To this end, we adopt a recent development proposed in Vo et al. (2024), which provides a tractable and likelihood-free formulation of the OT distance between the empirical and the model distributions over observables in a latent variable model. This framework additionally supports amortized optimization, allowing us to efficiently solve the parameters for all experts and features within a single training procedure. Appendix C presents details on the algorithm along with illustrative pseudo-code. For the technical proofs, readers are referred to the original paper.

## 5 RESULTS AND DISCUSSION

We assess the structure learning performance in both synthetic and real-world settings, where the estimated causal structures are compared with the ground-true ones based on the common metrics: structural Hamming distance (SHD), structural Intervention distance (SID) and F1 score for adjacency and orientation accuracies. SHD measures the smallest number of edge modifications required to recover the true graph. SID assesses the number of interventional distributions in the true causal graph that are disrupted in the estimated one. While SHD favours sparse graphs, SID penalizes overly sparse ones, resulting in a more balanced and comprehensive performance evaluation. Lower SHD, SHD ($\downarrow$) and higher F1 ($\uparrow$) are better. We generate $p$ graph predictions per expert, yielding up to $N = p^n$ possible voting profiles for a group of $n$ experts. We estimate the parameters from this collection of voting samples by solving objective (7). Bayes voting is thereafter applied on every voting profile to obtain an aggregated graph.

**Simulations.** We initially validate the theoretical results via simulations of the generative processes for noisy graph structures. We first define a prior joint distribution over edge states, where the probabilities for edge existence are sampled from $U(0, 0.5)$. For each expert, we then initialize edge-wise transition matrices, which specify the noise rates for each edge and encode the probability that an expert correctly predicts an edge's true state via the diagonal entries in the associated matrix. We examine three levels of expert competency – **strong, medium, weak** based on the distributions of the diagonal probabilities: $U(0.5, 1.0), U(0.4, 0.6)$ and $U(0.3, 0.5)$ respectively. It is worth recalling that when experts are only moderately or weakly competent, unique identifiability of the parameters is no longer guaranteed due to row-wise uncertainty. This setup aims to assess the robustness of causal discovery performance of noisy Bayes voting rule to this indeterminacy. For every edge, we iteratively sample its true state from the prior distribution and then select the corresponding transitional distribution to generate a noisy state. Repeating the procedure yields a set of noisy structures or voting samples for estimation. We evaluate the aggregated graphs from our "noisy" Bayes voting approach (`Bayes Est`) against the aggregated ones from the ground-true parameter values (`Bayes True`) and `Plurality` voting. The performance of the `Best Expert` (i.e., lowest in SHD, highest in F1) is also reported for comparative purposes.

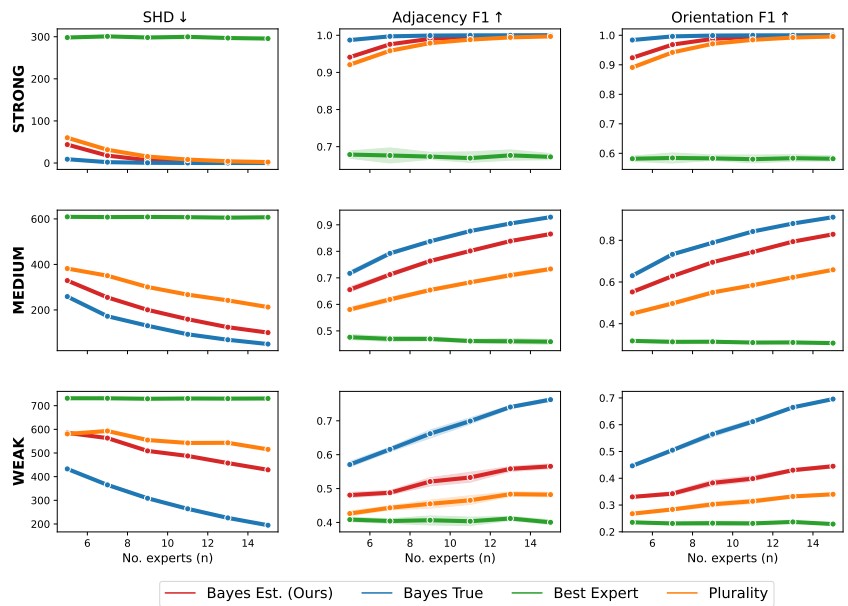

Figure 1: Simulations for graph size $d = 50$ on $50^n$ voting profiles.

**Real-world Systems.** We additionally analyze ensembles of real-world experts on 9 popular continuous and discrete benchmarks. We consider a selection of popular high-performing statistical algorithms of diverse classes of causal discovery approaches (see reviews in Appendix A). Besides `Plurality` voting, we also explore a rank-based ensemble method (`Rank`) proposed by Malmi et al. (2015) for comparison. In these experiments, $p = 50$ graph samples per expert are obtained via either bootstrapping or data partitioning. In the main text, we summarize our key findings (**F**) and present representative empirical evidence. We refer readers to Appendix F for the full results as well as the rationale behind the experimental design.

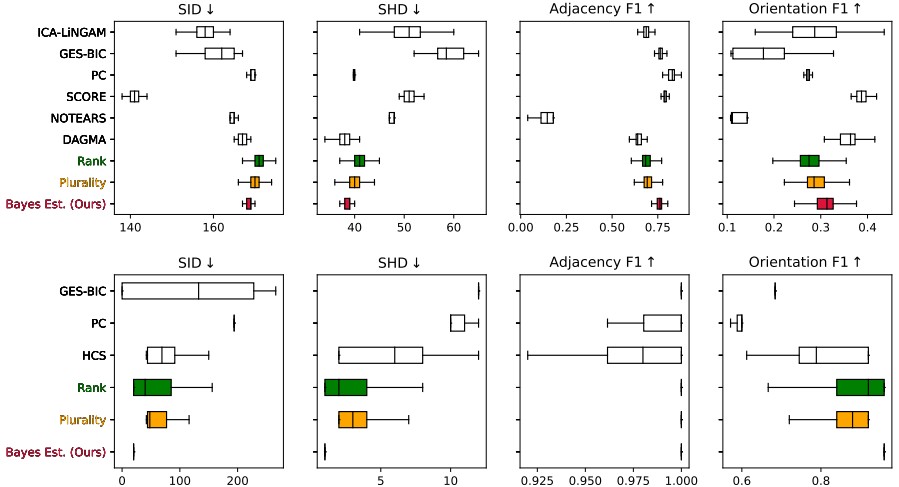

Figure 2: Experiments on **Sangiovese** (continuous, $d = 14$) and **Child** (discrete, $d = 20$) benchmarks. Causal experts are in black and ensemble-based methods are coloured.

**Key Takeaways.** Figures 1 and 2 report the average structure learning performance over 2000 random aggregated graphs and 30 model initializations. **(F1)** When the true parameters are known, the Bayes voting decision demonstrates its superior accuracy, which aligns with its theoretical optimality. In the absence of perfect knowledge, our noisy version `Bayes Est` secures the second-

best performance, approaching `Bayes True` as the number of experts increases as long as they are moderately to strongly competent. The performance gap widens with weakly competent experts, yet `Bayes Est` well surpasses `Plurality` voting and confirms its robustness across varying competency levels of experts in the ensemble. This performance pattern is consistent throughout our ablation studies on different graph sizes $d$ and voting samples $p$ (see results in Appendix F.2).

**(F2)** On the comparison with individual experts, the results are mixed, depending on the variation in experts' competency. This pertains to the effect of $\tau$ – a factor that so far remains undiscussed. In our simulations, the experts are homogenously competent, equivalent to keeping $\tau$ reasonably small, thus allowing for large $n$ to dominantly drive the probability of correctness to one. However, real-world experts tend to be heterogenously competent. Contrary to the common belief that ensemble performance is better than individual learners, our findings suggest that it is not always the case when experts' competence varies greatly. Bayes voting mitigates this situation by taking experts' competency into account and assigning more weights to competent experts, which `Plurality` and `Rank` fail to do. Our empirical evidence demonstrates the accuracy of our noisy aggregated graphs in this scenario approach that of the best causal experts in the ensemble, which is in fact a good news when the true performance of each expert is unknown in practice. Across real-world experiments (see results in Appendix F.1), we further show that our `Bayes Est` performs no worse than other ensemble-based methods, and outperforms them in most cases with lower variance.

**(F3)** The key message is that given a random voting profile from the experts, the aggregated graph obtained under `Bayes Est` on average would yield more desirable and reliable performance. In many scenarios, one may be interested in deriving a single final graph for downstream causal tasks. For this purpose, we recommend a two-phase ensembling strategy with two possible approaches: performing inter-algorithm aggregation first (via `Plurality` or `Rank`) and then intra-algorithm aggregation. Tables 1 present detailed numerical results, showing that applying `Bayes Est` on top of the set of "average" aggregated graphs produced within each expert achieves the strongest performance, where we find `Rank` + `Bayes Est` to be the most effective and stable combination.

# 6 Limitations and Conclusion

In this work, we have presented a rigorous approach to ensemble-based causal discovery, comprehensively addressing key aspects of the problem from recovery of the true causal graph, to robustness of noisy estimates and computational efficiency. Our quantitative analysis sheds light on when and how aggregated graphs perform optimally, thereby strengthening the theoretical underpinnings of ensemble-based methods.

Prior to aggregation, the proposed framework involves estimating experts' competence matrices using i.i.d. voting profiles conditioned on the input $\mathcal{D}$, which can be generated via bootstrapping the experts' graph predictions. Although some data settings (e.g., time series) may violate the i.i.d. assumption, established methods such as block bootstrapping allow the framework to remain applicable.

Our theoretical results further extend to learning general structures, including cyclic graphs. When the true graph is acyclic, under the conditions in Theorem 1 or Proposition 1, the aggregated graph is expected to be acyclic. Notably, our experimental results reinforces the result: the final aggregated graphs remain acyclic even without explicitly enforcing acyclicity constraints during optimization. In practice, conflicts can however still arise due to various computational complexities. In our setting, the aggregation procedure naturally produces posterior probabilities over the states of each feature (e.g., edges), which directly quantify the reliability of the experts' predictions. A common empirical strategy is to leverage this information post hoc: incorporating prior knowledge of acyclicity through thresholding (Zheng et al., 2020; Bello et al., 2022), where low-confidence edges are iteratively pruned until an acyclic graph is obtained.

Finally, due to the density of the paper, we currently focus on scenarios where the data inferred by the experts is sufficient, i.e., there are no latent confounders. Extending the model to account for confounders would involve introducing additional states in the features' state space, which comes with additional subtleties that require dedicated investigation. Fundamental considerations relate to, for instance, the number of confounders per pair of nodes or the selection of at least 7 reliable algorithms to deal with $m \geq 4$ states. We leave these challenges for future works to explore.

ACKNOWLEDGMENTS

Haoxuan Li is supported by the Institute for Decentralized AI, a project of the Cosmos Institute funded by the AI Safety Fund.

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

# Appendix

## Table of Contents

## USE OF LARGE LANGUAGE MODELS (LLMS)

The use of LLMs in this paper was exclusively limited to writing assistance and did not contribute to the conceptualization of the research or the implementation of the methodology.

## A  RELATED WORK

### A.1  CAUSAL DISCOVERY ALGORITHMS

Classic causal discovery algorithms can be broadly classified as *constraint-based* or *score-based* methods. The former class of methods, namely `PC` (Spirtes & Glymour, 1991) and `FCI` (Spirtes et al., 2000), detect edge existence and direction via conditional independence tests, while the latter consist of methods that optimize for DAGs based on a given objective function (Ott & Miyano, 2003; Chickering, 2002; Teyssier & Koller, 2012; Cussens et al., 2017). Research on continuous data in particular has seen significant advancements in recent years, largely due to the introduction of non-convex formulations for enforcing acyclicity. This has led to the emergence of numerous scalable DAG learning frameworks based on continuous optimization techniques, notably Lachapelle et al. (2019); Zheng et al. (2020); Yu et al. (2019); Bello et al. (2022). Constraint-based causal discovery can also be extended to discrete data with G-tests (Quine & Robinson, 1985) or $\chi^2$ tests (Cochran, 1952). Score-based methods, on the other hand, can be applied on multinomial Bayesian networks with BIC (Schwarz, 1978) or BDeu (Heckerman et al., 1995) scoring functions.

Structure learning is computationally intractable in the general case, as searching over the combinatorial space of DAGs is known to be NP-hard (Chickering, 1996; Chickering et al., 2004). An

important fact one can possibly exploits is that any DAG imposes at least one topological order and the ordering exists if and only if it is a DAG. *Ordering-based causal discovery* is an emerging line of research that focuses on the recovery of the partial orderings. The algorithm entails two stages: (1) determining a topological order and (2) subsequent post-processing to remove spurious edges. Research in ordering-based causal discovery recently takes off with the use of score matching (Rolland et al., 2022; Montagna et al., 2023; Xu et al., 2024), in which a valid causal order can be estimated by sequentially identifying the sink/leaf nodes (i.e., nodes with no outgoing edges) based on the Jacobian of the data log-likelihood.

**Definition 2** (Structural Identifiability). *A causal graph $\mathbf{G}$ is said to be structurally identifiable from observed data $X$ with distribution $\mathbb{P}(X \mid \mathbf{G})$ if there exists no other graph $\mathbf{G}' \neq \mathbf{G}$ such that $\mathbb{P}(X \mid \mathbf{G}') = \mathbb{P}(X \mid \mathbf{G})$.*

A data distribution $\mathbb{P}$ is said to be *faithful* to a causal graph $\mathbf{G}$ if every conditional independence in the data corresponds exactly to a $d$-separation implied by $\mathbf{G}$. It is a known result that the causal structure is generally non-identifiable from observational data $X$ (see Definition 2). There are however well-developed theories studying specific conditions under which identifiability is possible (Hoyer et al., 2008; Peters & Bühlmann, 2014; Zhang & Hyvarinen, 2012). Particularly, when faithfulness holds, a graph $\mathbf{G}$ becomes identifiable from data up to its Markov equivalence class, which refers to a set of causal graphs that encode the same set of conditional independencies, and $\mathbf{G}$ can be represented by a *completed partially DAG* (CPDAG). We refer readers to (Glymour et al., 2019; Vowels et al., 2022; Kitson et al., 2023) for comprehensive reviews of the recent advances in the literature. In the following, we shift the focus to ensemble-based approaches to causal discovery.

## A.2 ENSEMBLE-BASED CAUSAL DISCOVERY

Ensemble-based methods have actively advanced the field of causal discovery, moving beyond simple bootstrap aggregation to sophisticated, multi-layered frameworks designed for big data applications (Tang et al., 2019; Guo et al., 2021). These methods improve robustness and accuracy by integrating multiple causal graphs – often generated through data partitioning, algorithm diversity, or Bayesian sampling from some distribution over graphs – into a single consensus structure.

Early ensemble approaches focus on enhancing the stability of causal graph estimation, which is notoriously sensitive to small perturbations in the data. The core idea is bootstrap aggregating (bagging), that is to apply causal discovery algorithms on multiple data subsets (often generated via bootstrap sampling) and then aggregate the resulting graphs. These approaches, namely in Elidan et al. (2002); Wang & Peng (2014); Broom et al. (2012); Friedman et al. (2013), perform intra-algorithm aggregation, wherein the same causal discovery algorithm was applied across different samples. Recent developments extend this framework to a multi-phase ensemble strategy that further aggregates output structures from heterogeneous algorithms, which can be ensembled graphs obtained from an initial round of bagging. Such a hybrid approach seeks to harness the complementary strengths of diverse algorithms (Tang et al., 2019; Aslani & Mohebbi, 2023; Guo et al., 2021; Zhang et al., 2023). Since aggregation can introduce cycles, most methods also include a post-processing step to resolves violations of the acyclic constraint when specifically dealing with DAGs.

Structural aggregation methods generally fall into two categories: *distance-based* and *voting-based* approaches. In the former line of works, the goal is to find an "average" graph that minimizes a defined distance to all graphs in the ensemble. This includes methods such as MEDIAN (Jiang et al., 2001), which introduces the notion of "median" graph selecting one with the smallest graph-edit distance, or DAGBag (Wang & Peng, 2014), which leverages a family of metrics based on structural Hamming distance. Alternatively, ranking-based methods (Malmi et al., 2015; Aslani & Mohebbi, 2023) propose to use Kendall-Tau distance for aligning the partial orderings of causal DAGs. While promising, this aggregation problem is NP-hard particularly for DAGs. To address the computational challenges of high-dimensional data, several algorithms have been proposed: Wang & Peng (2014) develop a score-based method using hill-climbing search algorithm for DAG learning; Malmi et al. (2015) use a heuristic greedy search method designed to provide approximate solutions with theoretical guarantees; Aslani & Mohebbi (2023) propose a mixed-integer optimization framework where each edge's inclusion is a binary decision variable, and acyclicity constraints are handled efficiently through a lazy constraint approach.

Voting-based ensemble, on the other hand, is a simpler alternative. This strategy is often based on weighted voting mechanism, which can vary in complexity of the weighting or scoring functions. Here each individual structure or its adjacency matrix is assigned a weight, and this form of aggregation is typically performed at the edge level, where each edge inherits its weight from the same underlying mechanism. Weights can be based on edge frequency e.g., across multiple bootstrap resamples (Broom et al., 2012; Friedman et al., 2013), or variability in edge frequencies measured by coefficient of variation (Zhang et al., 2023), or goodness-of-fit Bayesian score like BDeu (Tang et al., 2019). Edge inclusion is then determined using threshold-based criteria or majority voting, reflecting the level of consensus across models. Final edge inclusion decisions are made using either simple majority voting (e.g., above 50% consensus) or threshold-based rules, allowing the ensemble method to balance confidence and consensus across multiple models.

Other directions include the work by Guo et al. (2021) that propose an ensemble causal discovery framework specifically designed for time-series data, or Mio et al. (2025) that formulate DAG selection as a multi-label classification task, aiming to identify the DAGs that are closest to the ground truth by comparing the topologies of the reconstructed graphs.

## A.3 THEORY OF VOTING

Voting theory is a branch of social choice theory that analyzes how groups can make collective decisions from individual preferences and beliefs. The field emerged from both political philosophy and mathematics, seeking to understand when and how democratic processes can produce good outcomes. The Condorcet Jury Theorem (de Condorcet, 1785; Young, 1988) provides a foundational result in voting theory: if each voter independently selects the correct option in a binary choice with probability above $50\%$, then the probability that the majority decision is correct approaches one as the number of voters grows infinitely large. This original formulation assumes identical competence, independent judgments, and majority rule.

Subsequent research has explored extensions of the theorem to address more complex and realistic voting scenarios, such as dependent voters (Berg, 1993; 1994; Berend & Sapir, 2007; Peleg & Zamir, 2012)or multiple choices (Goertz & Chernomaz, 2019). The theorem has also been extended to sequential voting where later voters observe earlier votes (Alpern & Chen, 2022), and to weighted voting systems with major and minor voters having different influence levels (Lindner, 2008). Modern works consider strategic voting and game-theoretic models, where voters behave according to Nash equilibrium strategies rather than voting sincerely (Schoenebeck & Tao, 2021).

An important extension of the Condorcet Jury Theorem is to the setting of epistemic Bayesian voting which incorporate heterogeneous voter competences and uncertainty. Notably, Nitzan & Paroush (1982) examine optimal decision rules in uncertain dichotomous choice settings, developing an optimal weighted majority rules where weights are proportional to the log-odds of each voter's accuracy to maximize the group's probability of correct decisions. This model allows for more nuanced aggregation, especially when experts' judgments are correlated or when their reliability varies across different issues.

Further advancements in Baharad et al. (2011) apply weighted aggregation across multiple issues, using performance on related questions to estimate and assign Bayesian-inspired weights for improved collective accuracy. Built on this framework, the authors then introduce an expectation-maximization (EM) algorithm that infers optimal weights solely from voting records, without requiring knowledge of the correct answers (Baharad et al., 2012). By analyzing agreement patterns, this method ranks voter expertise and facilitates statistically robust aggregation rules that outperform simple majority voting in uncertain epistemic contexts. In comparison with these past studies, our work essentially deals with a more general setting involving multiple issues and multiple alternatives, where we leverage the novel estimation algorithm based on optimal transport to solve this complex problem.

## B STRUCTURAL ENSEMBLING GUIDE

In this section, we present the details on how to configure an optimal ensemble strategy to maximize the probability of making the correct structural decisions under Bayes voting. The proposed guidelines are directly informed by our theoretical results in Proposition 1, Theorems 1 and 2.

## B.1 Choice of Aggregation Units

As discussed in Section 4, it is ideal both theoretically and computationally to focus on edge-level sub-structures, which corresponds to the feature level $l = 2$. Generally, the state spaces are not necessarily the same across edges. For example, the skeleton of a graph can be constructed from a shared domain over $\Omega$ that includes either absence or presence of undirected edges. A CPDAG on the other hand has mixed domains where the state spaces of some features consider edge directions while others do not.

We here consider a shared state space over all features with $m = 3$ mutually exclusive states for each pair of nodes $(v_i, v_j)$ e.g., $\mathcal{S}(v_i, v_j) = \{1 : v_i \rightarrow v_j, \, 2 : v_i \leftarrow v_j, 3 : \text{no edge}\}$. We note that this design choice is purely for ease of implementation and it is entirely feasible to independently solve for each feature with different state spaces. For algorithms like `PC` or `GES` that only return CPDAGs, we handle undetermined edge orientations by assuming uniform probabilities to both directions when edge existence is confirmed.

## B.2 Characteristics of Experts

There are $4$ critical aspects to be considered for a desirable pool of experts. The two most important conditions to satisfy are Assumption 1 and Condition 1. The former pertains to the condition among experts, while the latter refers to the condition within experts.

First, Assumption 1, which reflects the diversity of experts in the decision-making body, requires the experts to be mutually independent given the ground truth. This condition can be reasonably satisfied by selecting causal discovery experts that operate on distinct modelling assumptions i.e., algorithms in diverse classes of approaches, for example those outlined in Appendix A.

Second, by Condition 1, each of the experts to be informative in the technical sense that no every two row in every competency matrix is identical. Consider a competency matrix $\boldsymbol{T}_i \in (\Delta^{|\mathcal{G}|-1})^{|\mathcal{G}|}$ where each row is drawn from any distribution that has a joint density on the simplex $\Delta^{|\mathcal{G}|-1}$. It is a classic result in measure theory (e.g., see Hurder & Mitsumatsu (1991), Folland (1999)) that the set of events where every two point on probability simplex size $K$ is equal has measure zero under any law that is absolutely continuous w.r.t. the product Lebesgue measure on the simplex. This reassures us that Condition 1 holds almost surely and can be well justified for use in practice.

Third, Theorem 2 introduces an important caveat in relation to the identifiability result: beyond informativeness, the experts must also be competent – that is, they should be more likely make correct predictions than not. This corresponds to the diagonal entries of each competency matrix being dominant in their respective rows, as formally defined below.

**Condition 2** (Competency). *An $i$-th expert in the decision-making body is said to be competent if* $\boldsymbol{T}_{i,\mathbf{G}_j|\mathbf{G}_j} > \boldsymbol{T}_{i,\mathbf{G}_k|\mathbf{G}_j}, \forall j, k \in \{1, \cdots, |\mathcal{G}|\}$.

Competency implies informativeness, thus all theoretical results hold when Condition 2 is satisfied. It is worth noting that this condition is purely for the purpose of estimation, precisely to ensure unique identifiability of the parameters for the reliable recovery of the true competency matrices. While this assumption is often unverifiable in practice, it can be approximated through prior knowledge or empirical benchmarking on datasets of related domains. A natural strategy to increase the chance of meeting this requirement prefer methods with few theoretical assumptions and/or strong empirical performance. We suggest a realistic selection of such experts in Appendix F.1.

Fourth, not only should the experts be principally heterogeneous but they should also be homogenously competent. In other words, the variation among experts' competence should be minimal, such that a small realization of $\tau$ their competence can be achieved. This would contribute to minimizing the error probability, as suggested in Theorem 1.

## B.3 Number of Experts

The final question is how many of such experts is need. A straightforward answer from our theoretical results is: the more, the better. Furthermore, Theorem 2 says that the minimum number of experts sufficient for identifiability is $(2m - 1)$ where $m$ is the number of feature states. Choosing $m = 3$ implies that we need at least $5$ experts to be generally safe. As discussed in the proof in Appendix

E.4, this requirement holds under the generic condition that the Kruskal rank of the competency matrices is at least 2. With more specific knowledge about the exact rank of each expert's competency matrix, this number can be reduced to as few as 3 experts. In our empirical evaluation on real-world discrete datasets, we show that such a reduced number of experts can still yield reliable performance in several cases.

Lastly, it is worth recalling that a large number of experts introduces computational constraints related to the estimation of the quantities necessary for applying Bayes voting. Fortunately, the parameter estimation framework introduced by Vo et al. (2024) has significantly contributed to alleviating this computational burden by facilitating an end-to-end optimization procedure for all features and experts. We detail how this framework is applied in the next section.

## C  PARAMETER ESTIMATION ALGORITHM

**Problem Setup.**  We wish to estimate the parameters $\theta$ that consist of

$$\theta := \big\{\theta(\omega)\big\}_{\omega \in \Omega}, \quad \theta(\omega) := \big\{\boldsymbol{\pi}(\omega),\, \boldsymbol{T}_1(\omega), \cdots, \boldsymbol{T}_n(\omega)\big\}.$$

Vo et al. (2024) proposes an OT-based framework for estimating the parameters of general directed graphical models (of known structures) with latent variables. To understand how our problem fits into this landscape, we first examine the generative process of noisy sub-structures.

For every sub-structure $\omega \in \Omega$, the noisy data generating process goes as follows:

1. Sample true state $y(\omega) \sim \mathrm{Cat}(\boldsymbol{\pi}(\omega))$,
2. Sample noisy state $x_i(\omega) \sim \mathrm{Cat}\big(\boldsymbol{T}_{i,|y}(\omega)\big), \quad \forall i \in [n]$.

This generative setup can thus be represented by a simple directed graphical model over the set of observed variables $X(\omega) = \{X_i(\omega)\}_{i \in [n]}$ and latent variable $Y(\omega)$.

Let $\boldsymbol{e}_{y(\omega)}$ be the one-hot representation of discrete value $y(\omega) \in [m]$. Let $\boldsymbol{u}(\omega) \sim \mathbb{P}(U(\omega)) \triangleq \mathrm{Uniform}([0,1]^m)$.

If one employs the Gumbel-Softmax reparameterization trick (Jang et al., 2016; Maddison et al., 2016), the computation of $x_i(\omega)$ can be written deterministically as

$$x_i(\omega) = \psi_\theta\left[\boldsymbol{u}(\omega), y(\omega)\right] \triangleq \text{Cat-Concrete}\left(\boldsymbol{e}_{y(\omega)} \cdot \boldsymbol{T}_i(\omega)\right). \tag{8}$$

Here the Cat-Concrete($\boldsymbol{p}$) function is defined on a probability vector $\boldsymbol{p} = \left[p^{(1)}, p^{(2)}, \cdots, p^{(m)}\right]$ as

$$\text{Cat-Concrete}(\boldsymbol{p}) = \left[\frac{\exp\left\{(\log p^{(j)} + G^{(j)})/\tau\right\}}{\sum_{k=1}^{m} \exp\left\{(\log p^{(k)} + G^{(k)})/\tau\right\}}\right]_{j \in [m]},$$

with temperature $\tau$, random noises $G^{(j)} = -\log(-\log u^{(j)})$ and $u^{(j)} \sim \mathrm{Uniform}(0,1)$.

For ease of notation, we compactly denote $\boldsymbol{z}(\omega) := [\boldsymbol{u}(\omega), y(\omega)] \in [0,1]^m \times [m]$ where $\boldsymbol{z}(\omega) \sim \mathbb{P}_\theta(Z(\omega))$ with

$$\mathbb{P}_\theta(Z(\omega)) := \mathbb{P}(U(\omega))\, \mathbb{P}_\theta(Y(\omega) \mid \mathcal{D}),$$

and the subscript $\theta$ indicating the involvement of the parameters $\boldsymbol{\pi}(\omega)$ in modelling the distribution of random variable $Z(\omega)$.

**Optimization Objective.**  Let $\phi(\omega) : \mathcal{X}(\omega) \mapsto [0,1]^m \times [m]$ be a stochastic map that satisfies

$$\phi(\omega)\#\mathbb{P}_N(X(\omega) \mid \mathcal{D}) = \mathbb{P}_\theta(Z(\omega)).$$

Then Vo et al. (2024) show that for fixed $\theta$, the OT distance in (7) can be tractably computed as

$$W_c\Big[\mathbb{P}_N(X(\omega)); \mathbb{P}_\theta(X(\omega))\Big] = \min_\phi \mathbb{E}_{\boldsymbol{x}(\omega) \sim \mathbb{P}_N, \boldsymbol{z}(\omega) \sim \phi(\boldsymbol{x}(\omega))} \quad c\Big[\boldsymbol{x}(\omega); \psi_\theta(\boldsymbol{z}(\omega))\Big]. \tag{9}$$

Let us define $\mathbb{P}_\phi(Z(\omega)) = \mathbb{E}_{\boldsymbol{x}(\omega) \sim \mathbb{P}_N}[\phi(\boldsymbol{x}(\omega))]$. This expression of $W_c$ renders objective (7) as a constrained optimization problem, which can be relaxed into

$$
\min_{\theta(\omega)} \quad \min_{\phi(\omega)} \quad \left\{ \mathbb{E}_{\boldsymbol{x}(\omega) \sim \mathbb{P}_N, \boldsymbol{z}(\omega) \sim \phi(\boldsymbol{x}(\omega))} \quad c\Big[ \boldsymbol{x}(\omega); \psi_\theta(\boldsymbol{z}(\omega)) \Big] \right. \tag{10}
$$
$$
\left. + \lambda \cdot D\Big[ \mathbb{P}_\phi(Z(\omega)) \, \big\| \, \mathbb{P}_\theta(Z(\omega)) \Big] \right\},
$$

where $D$ is any arbitrary divergence measure and $\lambda > 0$ is a regularization coefficient.

The key ingredient to solving (10) is the construction of the push-forward map $\phi(\omega)$ to approximate the generative distribution for the noisy data. The authors further prove that if the family of the valid maps $\phi(\omega)$ have infinite capacity (i.e. they include all measurable functions), then (10) shares the same optimal solution with the primal objective (7) (see Theorem A.1 therein).

**Practical Algorithm.** By the above reformulation, our problem is now reduced to minimizing the reconstruction of the noisy data, which can now be solved with gradient-based optimizers. Whereas the optimization objective appears complicated, Vo et al. (2024) suggests two simplifications of (10) that are empirically shown to yield adequate estimates on problems dealing with categorical variables.

First, while in principle we need to additionally model a backward distribution to align with the uniform noises, the authors show that it is empirically safe to ignore the noise variable, which means that in our case, it is sufficient to only model $\phi(\omega)\#\mathbb{P}_N(X(\omega) \mid \mathcal{D}) = \mathbb{P}_\theta(Y(\omega) \mid \mathcal{D})$ and apply the reparameterization trick with random uniform noises to generate reconstructed samples of $x(\omega)$.

Second, the backward function $\phi$ here, essentially, returns the probability vector parameterizing the push-forward distribution $\mathbb{P}_\phi$ whose role is to mimic the model prior distribution $\boldsymbol{\pi}(\omega)$. As for the reconstruction loss, the authors also show that it is sufficient to alternatively minimize the cross-entropy loss between the target data samples $\boldsymbol{x}(\omega)$ and generative probabilities $\boldsymbol{T}(\omega)$. The **practical optimization objective** to our estimation problem is given as:

$$
\min_\theta \quad \min_\phi \quad \mathbb{E}_{\boldsymbol{x} \sim \mathbb{P}_N, \tilde{y} \sim \phi(\boldsymbol{x})} \left\{ \mathrm{CE}\big(\boldsymbol{x}; \boldsymbol{e}_{\tilde{y}}; \boldsymbol{T}\big) + \mathrm{CE}\big(\boldsymbol{x}; \boldsymbol{\pi}; \boldsymbol{T}\big) + \lambda \cdot \mathrm{JS}\big[\phi(\boldsymbol{x}) \, \big\| \, \boldsymbol{\pi}\big] \right\}, \tag{11}
$$

where we omit the notation $(\omega)$ for readability. Here the Jensen-Shannon (JS) divergence is chosen as the divergence measure and the CE loss function is computed as:

$$
\mathrm{CE}\,(\boldsymbol{x}; \boldsymbol{p}; \boldsymbol{T}) = \sum_{i=1}^{n} \sum_{j=1}^{m} -x_i^{(j)} \log\big(\boldsymbol{p} \cdot \boldsymbol{T}_{i,|j}\big) - \Big(1 - x_i^{(j)}\Big) \log\big[1 - \log\big(\boldsymbol{p} \cdot \boldsymbol{T}_{i,|j}\big)\big].
$$

We refer readers to Vo et al. (2024) for a formal algorithm (see Algorithm 1 therein). To make the paper easy to follow, we only provide an informal summary of the training procedure:

1. Initialize $\phi(\omega)$ and $\theta(\omega) = \{\boldsymbol{\pi}(\omega), \boldsymbol{T}_1(\omega), \cdots, \boldsymbol{T}_n(\omega)\}$.
2. Draw a batch of samples $\boldsymbol{x}_b(\omega) \sim \mathbb{P}_N(X(\omega) \mid \mathcal{D})$.
3. Evaluate $\phi_\theta(\boldsymbol{x}_b(\omega))$.
4. Update $\phi(\omega)$ and $\theta(\omega)$ alternately by descending (11).

**Enforcing Diagonal Dominance via Continuous Sinkhorn Operator.** To mitigate the effects of uncertainty arising from permutation indeterminacy, we impose an inductive bias on the structure of each competency matrix, based on the prior knowledge that each expert is competent in the sense that their diagonal entries should be the largest in their respective rows (see Condition 2).

The goal is essentially to reorder the rows and columns of a matrix in a way that maximizes the total sum of its diagonal entries. Formally, this corresponds to a linear assignment problem: identifying the permutation that yields the highest sum along the diagonal. While this is a classic combinatorial optimization problem typically solved using the Hungarian algorithm (Kuhn, 1955), these methods are not differentiable and therefore unsuitable for gradient-based learning.

To impose this structure while maintaining differentiability for end-to-end training, we adopt the continuous Sinkhorn operator (Mena et al., 2018), which offers a differentiable alternative by

relaxing the discrete set of permutation matrices into the continuous space of doubly stochastic matrices—square matrices where each row and column sums to one. The algorithm iteratively normalizes the rows and columns to approximate this space, effectively yielding a soft permutation matrix. Additionally, entropy regularization is introduced to encourage smoothness and numerical stability. As a result, the Sinkhorn operator produces a soft assignment that aligns high-value entries near the diagonal, thus enforcing the desired diagonal dominance in a differentiable manner.

**Training Configurations.** We optimize (11) for all features $\omega \in \Omega$ at once with a shared backward function $\phi$ modelled by a 2-layer MLP of 192 hidden units. We use Adam optimizer at learning rate of $0.001$, $\lambda = 0.001$, 500 training steps and batch size of $10,000$. We initialize the model parameters with values drawn from the normal distribution. To reduce the uncertainty due to random initializations of the parameters and selections of the graph samples, we take an average of the estimated values obtained over 30 different runs as the final estimates for $\theta$.

## D   Summary of Notation

We summarize the key notations used throughout the paper to provide a quick reference for readers.

- $\mathbb{P}$: probability measure
- $\mathbf{V}$: set of nodes in a graph
- $d$: number of nodes
- $n$: number of experts
- $\mathcal{D}$: arbitrary input data for graph prediction
- $\mathcal{G}$: space of graph structures
- $G$: true graph structure
- $\widetilde{G}_i$: noisy graph structure predicted by an $i$-th expert
- $\boldsymbol{\pi} \in [0,1]^{|\mathcal{G}|}$: prior distribution $\mathbb{P}(G \mid \mathcal{D})$
- $\boldsymbol{T}_i \in [0,1]^{|\mathcal{G}| \times |\mathcal{G}|}$: competence transition matrix $\mathbb{P}(\widetilde{G}_i \mid G, \mathcal{D})$
- $\Omega$: space of graph sub-structures
- $\omega$: arbitrary sub-structure (feature)
- $\mathcal{S}(\omega)$: state space of $\omega$
- $Y(\omega)$: true state of $\omega$
- $X_i(\omega)$: noisy state of $\omega$ predicted by an $i$-th expert
- $\boldsymbol{\pi}(\omega) \in [0,1]^{|\mathcal{S}(\omega)|}$: marginal prior distribution $\mathbb{P}(Y(\omega) \mid \mathcal{D})]$
- $\boldsymbol{T}_i(\omega) \in [0,1]^{|\mathcal{S}(\omega)| \times |\mathcal{S}(\omega)|}$: marginal competence transition matrix $\mathbb{P}(X_i(\omega) \mid Y(\omega), \mathcal{D})$
- $\theta(\omega) := \{\boldsymbol{\pi}(\omega), \boldsymbol{T}_1(\omega), \cdots, \boldsymbol{T}_n(\omega)\}$: parameters to be estimated for $\omega$
- $\Theta(\omega)$: parameter space for $\omega$
- $N$: number of empirical data samples
- $\Rightarrow$: weak convergence.

For any $i \in [n]$ and $j, k \in \{1, \cdots, |\mathcal{G}|\}$,

- $p_{i,j} \triangleq p_{i,\mathbf{G}_j} = \boldsymbol{T}_{i,\mathbf{G}_j|\mathbf{G}_j} := \mathbb{P}(\widetilde{G}_i = \mathbf{G}_j \mid G = \mathbf{G}_j, \mathcal{D})$
- $q_{i,j|k} \triangleq q_{i,\mathbf{G}_j|\mathbf{G}_k} = \boldsymbol{T}_{i,\mathbf{G}_j|\mathbf{G}_k} := \mathbb{P}(\widetilde{G}_i = \mathbf{G}_j \mid G = \mathbf{G}_k, \mathcal{D})$
- $\pi_j \triangleq \pi_{\mathbf{G}_j} := \mathbb{P}(G = \mathbf{G}_j \mid \mathcal{D})$.

For any $i \in [n]$ and $j, k \in \{1, \cdots, |\mathcal{S}(\omega)|\}$,

- $p_{i,j}(\omega) = \boldsymbol{T}_{i,j|j}(\omega) := \mathbb{P}(X_i(\omega) = j \mid Y(\omega) = j, \mathcal{D})$

- $q_{i,j|k}(\omega) = \boldsymbol{T}_{i,j|k}(\omega) := \mathbb{P}(X_i(\omega) = j \mid Y(\omega) = k, \mathcal{D})$

- $\pi_j(\omega) = \mathbb{P}(Y(\omega) = j \mid \mathcal{D})$

- $\widetilde{p}_{i,j}(\omega), \widetilde{q}_{i,j|k}(\omega), \widetilde{\pi}_j(\omega)$: noisy counterparts of $p_{i,j}(\omega), q_{i,j|k}(\omega), \pi_j(\omega)$ respectively.

# E    PROOFS

## E.1    PROOF FOR THEOREM 1

**Theorem 1.**    *Under Assumptions 1 and 2, the collective decision from a group of $n$ informative experts is correct under Bayes voting rule with probability at least*

$$1 - \sum_{j \geq 2} \exp\left(-n \cdot \overline{\mathrm{KL}}_{1,j}^2 \Big/ 2\tau^2\right), \tag{12}$$

*where $\overline{\mathrm{KL}}_{1,j} = n^{-1} \sum_{i=1}^{n} \mathrm{KL}\left(\boldsymbol{T}_{i,|\mathbf{G}_1}; \boldsymbol{T}_{i,|\mathbf{G}_j}\right), \forall j \in \{2, \cdots, |\mathcal{G}|\}$.*

*Proof.* Let us recall the scoring function for a candidate graph $\mathbf{G} \in \mathcal{G}$ in (3) :

$$S_{n,\boldsymbol{w_G}}(\widetilde{G}) = \sum_{i=1}^{n} w_{i,\mathbf{G}} \, \mathbf{1}\left[\widetilde{G}_i = \mathbf{G}\right] + b_{\mathbf{G}},$$

and the collective decision is given as $\widehat{\mathbf{G}} = \arg\max_{\mathbf{G} \in \mathcal{G}} S_{n,\boldsymbol{w_G}}(\widetilde{G})$.

For ease of notation, we omit the subscript $(n, \boldsymbol{w}_j)$ and simply denote $S_j(\widetilde{G})$ as the score assigned to a candidate structure $\mathbf{G}_j$.

Let us also define the binary random variable $\widetilde{G}_{i,j} := \mathbf{1}\left[\widetilde{G}_i = \mathbf{G}_j\right]$. With a slight abuse of notation, we sometimes treat each $\widetilde{G}_i$ as a discrete random variable that takes an index value corresponding to its structural realization. Assuming the true graph corresponds to index 1, we have $\widetilde{G}_{i,1} \sim$ Bernoulli$(p_{i,1})$, and $\forall j \geq 2 : \widetilde{G}_{i,j} \sim$ Bernoulli$(q_{i,j|1})$.

In Bayes voting rule, the score function is given as

$$S_j \triangleq S_j(\widetilde{G}) = \log \mathbb{P}(G = \mathbf{G}_j \mid \widetilde{G}, \mathcal{D}), \tag{13}$$

$$\propto \log \mathbb{P}(\widetilde{G} \mid \mathbf{G}_j, \mathcal{D}) + \log \mathbb{P}(\mathbf{G}_j \mid \mathcal{D}), \tag{14}$$

$$= \sum_{i=1}^{n} \log \mathbb{P}(\widetilde{G}_i \mid \mathbf{G}_j, \mathcal{D}) + \log \mathbb{P}(\mathbf{G}_j \mid \mathcal{D}), \tag{15}$$

$$= \sum_{i=1}^{n} \widetilde{G}_{i,j} \log p_{i,j} + (1 - \widetilde{G}_{i,j}) \log q_{i,\widetilde{G}_i|j} + \log \pi_j, \tag{16}$$

$$= \sum_{i=1}^{n} \widetilde{G}_{i,j} \left(\log p_{i,j} - \log q_{i,\widetilde{G}_i|j}\right) + \sum_{i=1}^{n} \log q_{i,\widetilde{G}_i|j} + \log \pi_j. \tag{17}$$

The second equality (15) results from Assumption 1. This final equality gives rise to the weight and bias terms of Bayes voting in (4) where

$$w_{i,j} = \log p_{i,j} - \log q_{i,\widetilde{G}_i|j}, \quad b_j = \sum_{i=1}^{n} \log q_{i,\widetilde{G}_i|j} + \log \pi_j.$$

The probability that the predicted graph $\widehat{\mathbf{G}}$ is correct is given as:

$$\Pr\left(\widehat{\mathbf{G}} = \mathbf{G}_1\right) = \Pr\left(\bigcap_{j \geq 2}\left\{S_1(\widetilde{G}) > S_j(\widetilde{G})\right\}\right), \tag{18}$$

$$= 1 - \Pr\left(\bigcup_{j \geq 2}\left\{S_1(\widetilde{G}) \leq S_j(\widetilde{G})\right\}\right), \tag{19}$$

$$\geq 1 - \sum_{j \geq 2}\Pr\left(S_1(\widetilde{G}) \leq S_j(\widetilde{G})\right). \tag{20}$$

Let $Z_{i,j} = \widetilde{G}_{i,j}\left(\log p_{i,j} - \log q_{i,\widetilde{G}_i|j}\right) + \log q_{i,\widetilde{G}_i|j} + \frac{1}{n}\log\pi_j$. We have

$$\mathbb{E}[Z_{i,1}] = p_{i,1}\log p_{i,1} + \sum_{k \geq 2} q_{i,k|1}\log q_{i,k|1} + \frac{1}{n}\log\pi_1, \tag{21}$$

$$\mathbb{E}[Z_{i,j}] = p_{i,1}\log q_{i,1|j} + \sum_{k \geq 2} q_{i,k|1}\log q_{i,k|j} + \frac{1}{n}\log\pi_j \quad \text{for } j \geq 2, \tag{22}$$

where $p_{i,j} \triangleq q_{i,j|j}$ by definition. We thus have

$$\mathbb{E}[Z_{i,1} - Z_{i,j}] = p_{i,1}\log\frac{p_{i,1}}{q_{i,1|j}} + \sum_{k \geq 2} q_{i,k|1}\log\frac{q_{i,k|1}}{q_{i,k|j}} + \frac{1}{n}\log\frac{\pi_1}{\pi_j}, \tag{23}$$

$$= q_{i,1|1}\log\frac{q_{i,1|1}}{q_{i,1|j}} + \sum_{k \geq 2} q_{i,k|1}\log\frac{q_{i,k|1}}{q_{i,k|j}} + \frac{1}{n}\log\frac{\pi_1}{\pi_j}, \tag{24}$$

$$= \sum_{k=1}^{K} q_{i,k|1}\log\frac{q_{i,k|1}}{q_{i,k|j}} + \frac{1}{n}\log\frac{\pi_1}{\pi_j}, \tag{25}$$

$$= \mathrm{KL}\left(\boldsymbol{T}_{i,|1}; \boldsymbol{T}_{i,|j}\right) + \frac{1}{n}\log\frac{\pi_1}{\pi_j}. \tag{26}$$

The score difference between the true DAG and some candidate structure $G_j$ can be written as

$$S_1 - S_j = \sum_{i=1}^{n}\left(Z_{i,1} - Z_{i,j}\right), \tag{27}$$

$$\mathbb{E}\left[S_1 - S_j\right] = \sum_{i=1}^{n}\mathbb{E}[Z_{i,1} - Z_{i,j}] = \sum_{i=1}^{n}\mathrm{KL}\left(\boldsymbol{T}_{i,|\mathbf{G}_1}; \boldsymbol{T}_{i,|\mathbf{G}_j}\right) + \log\frac{\pi_1}{\pi_j}. \tag{28}$$

Let $\overline{\mathrm{KL}}_{1,j} := n^{-1}\sum_{i=1}^{n}\mathrm{KL}\left(\boldsymbol{T}_{i,|\mathbf{G}_1}; \boldsymbol{T}_{i,|\mathbf{G}_j}\right)$, and $\pi_{1,j} := \pi_1/\pi_j$. Since experts are informative, we have $\forall j \geq 2 : \boldsymbol{T}_{i,|j} \neq \boldsymbol{T}_{i,|1}$. It follows that $\forall j \geq 2 : \overline{\mathrm{KL}}_{1,j} > 0$.

By Assumption 2, there exists a constant $\tau < \infty$ such that $\forall i \in [n] : \|\log\boldsymbol{T}_i\|_{\max} \leq \tau$. Hence, we can establish that $\forall i \in [n]$: $Z_{i,1} - Z_{i,j} \in [-\tau, \quad \tau]$.

Since the truth is biased, that is $\pi_1 \geq \pi_j > 0$ then $\mathbb{E}[S_1 - S_j]$ is strictly positive. Applying Hoeffding's inequality,

$$\Pr\left(S_1 \leq S_j\right) = \Pr\left(S_j - S_1 - \mathbb{E}[S_j - S_1] \geq \mathbb{E}[S_1 - S_j]\right), \tag{29}$$

$$\leq \exp\left(-\frac{2\left(n \cdot \overline{\mathrm{KL}}_{1,j} + \log\pi_{1,j}\right)^2}{4n\tau^2}\right), \tag{30}$$

$$= \exp\left(-\frac{n \cdot \overline{\mathrm{KL}}_{1,j}^2 + 2\overline{\mathrm{KL}}_{1,j} \cdot \log\pi_{1,j} + n^{-1}\log^2\pi_{1,j}}{2\tau^2}\right). \tag{31}$$

Substituting the above expression to (20) gives

$$\Pr\left(\widehat{\mathbf{G}} \text{ is correct}\right) = \Pr\left(\widehat{\mathbf{G}} = \mathbf{G}_1\right) \tag{32}$$

$$\geq 1 - \sum_{j \geq 2} \exp\left(-\frac{n \cdot \overline{\mathrm{KL}}_{1,j}^2 + 2\overline{\mathrm{KL}}_{1,j} \cdot \log \pi_{1,j} + n^{-1} \log^2 \pi_{1,j}}{2\tau^2}\right), \tag{33}$$

$$\geq 1 - \sum_{j \geq 2} \exp\left(-\frac{n \cdot \overline{\mathrm{KL}}_{1,j}^2}{2\tau^2}\right). \tag{34}$$

We obtain the desired lower bound due to the non-negativity of $\overline{\mathrm{KL}}_{1,j}$ and $\log \pi_{1,j}$. $\qquad \square$

### E.2 Proof for Corollary 1

**Corollary 1.** $\widehat{\mathbf{G}}(\Omega, \mathcal{S})$ *coincides with* $\mathbf{G}_1(\Omega, \mathcal{S})$ *if and only if the voting decisions on every feature* $\omega \in \Omega$ *are correct.*

*Proof.* For a given feature space and domain $(\mathbf{V}, \Omega, \mathcal{S})$, we denote $Y(\omega) \in \mathcal{S}(\omega)$ as the true state of a feature $\omega \in \Omega$, and $\widehat{Y}(\omega) : (\mathcal{S}(\omega))^n \mapsto \mathcal{S}(\omega)$ be the decision from $n$ experts about the state of $\omega$ based on some voting rule.

Structural Hamming distance (SHD) is a commonly used metric for comparing two graphs, which quantifies the number of edge additions, deletions, or reversals required to convert one graph structure into another. We thus define the generalized SHD between the aggregated structure $\widehat{\mathbf{G}}(\Omega, \mathcal{S})$ and the true structure $\mathbf{G}_1(\Omega, \mathcal{S})$ as

$$\mathrm{SHD}\left[\widehat{\mathbf{G}}(\Omega, \mathcal{S}); \mathbf{G}_1(\Omega, \mathcal{S})\right] = \sum_{\omega \in \Omega} \mathbf{1}\left[\widehat{Y}(\omega) \neq Y(\omega)\right]. \tag{35}$$

We say that $\widehat{\mathbf{G}}(\Omega, \mathcal{S})$ coincides with $\mathbf{G}_1(\Omega, \mathcal{S})$ when $\mathrm{SHD}\left[\widehat{\mathbf{G}}(\Omega, \mathcal{S}); \mathbf{G}_1(\Omega, \mathcal{S})\right] = 0$.

By definition, we have

$$\left\{\mathrm{SHD}\left[\widehat{\mathbf{G}}(\Omega, \mathcal{S}); \mathbf{G}_1(\Omega, \mathcal{S})\right] > 0\right\} = \bigcup_{\omega \in \Omega} \left\{\widehat{Y}(\omega) \neq Y(\omega)\right\}.$$

Hence, the SHD equals $0$ if and only if no feature error occurs. Consequently, the probability that $\widehat{\mathbf{G}}(\Omega, \mathcal{S})$ is correct is upper bounded by the worst-case feature accuracy:

$$\Pr\left(\widehat{\mathbf{G}}(\Omega, \mathcal{S}) \text{ is correct}\right) = \Pr\left(\bigcap_{\omega \in \Omega} \left\{\widehat{Y}(\omega) = Y(\omega)\right\}\right) \tag{36}$$

$$\leq \min_{\omega \in \Omega} \Pr\left(\widehat{Y}(\omega) = Y(\omega)\right). \tag{37}$$

$\qquad \square$

The analyses of Proposition 1 and Theorem 2 are conducted at the sub-structure level and applicable to arbitrary sub-structure $\omega$ in a pre-defined feature space $\Omega$. For brevity, we will sometimes drop the notation $(\omega)$ in the derivation whenever there is no ambiguity.

### E.3 PROOF FOR PROPOSITION 1

**Proposition 1.** *Suppose the estimates $\{\widetilde{T}_i(\omega)\}_{i=1}^n$ satisfy Assumption 2 w.r.t some constant $\widetilde{\tau} < +\infty$, then under Assumption 1, the collective decision from a group of $n$ experts for a feature $\omega$ is correct under "noisy" Bayes voting rule with probability at least*

$$1 - \sum_{j \geq 2} \exp\left[-\Theta\left(n \cdot \Delta\overline{\mathrm{KL}}_{1,j}^2(\omega)\Big/2\widetilde{\tau}^2\right)\right], \tag{38}$$

*as long as $\Delta\overline{\mathrm{KL}}_{1,j}(\omega) \gg n^{-1/2}$ and $\log\left(\widetilde{\pi}_1(\omega)/\widetilde{\pi}_j(\omega)\right) = \mathcal{O}(1)$ where $\Delta\overline{\mathrm{KL}}_{1,j}(\omega) = n^{-1}\sum_{i=1}^n\left[\mathrm{KL}\left(T_{i,|1}(\omega);\widetilde{T}_{i,|j}(\omega)\right) - \mathrm{KL}\left(T_{i,|1}(\omega);\widetilde{T}_{i,|1}(\omega)\right)\right], \forall j \in \{2,\cdots,|\mathcal{S}(\omega)|\}.$*

*Proof.* Let $\widetilde{S}_j(\omega)$ denote the score function assigned to some candidate state $j \in \mathcal{S}(\omega)$. Under Bayes voting rule, we have $\widetilde{S}_j(\omega) = \log\mathbb{P}(Y(\omega) = j \mid X(\omega), \mathcal{D})$. The collective decision is given as $\widehat{Y}(\omega) = \arg\max_{j \in \mathcal{S}(\omega)} S_j(\omega)$.

We will similarly label 1 as the correct state of $\omega$ to avoid notation overloading. The probability that the collective decision on $\omega$ is correct can be written as

$$\Pr\left(\widehat{Y}(\omega) \text{ is correct}\right) = \Pr\left(\widehat{Y}(\omega) = 1\right) \geq 1 - \sum_{j \geq 2}\Pr\left(\widetilde{S}_1(\omega) \leq \widetilde{S}_j(\omega)\right). \tag{39}$$

Let us denote the binary random variable $X_{i,j}(\omega) = \mathbf{1}[X_i(\omega) = j]$. We similarly have $X_{i,1}(\omega) \sim$ Bernoulli$(p_{i,1}(\omega))$ and $\forall j \geq 2 : X_{i,j}(\omega) \sim$ Bernoulli$(q_{i,j|1}(\omega))$.

Following the same derivations in the Theorem 1, we can define $Z_{i,j}(\omega) = X_{i,j}(\omega)\left(\log p_{i,j}(\omega) - \log q_{i,X_i|j}(\omega)\right) + \log q_{i,X_i|j}(\omega) + \frac{1}{n}\log\pi_j(\omega)$.

In this case, we are dealing with the "noisy score" induced from from the noisy estimates $\widetilde{\pi}(\omega)$ and $\{\widetilde{T}_i\}_{i\in[n]}$. Alternatively, we define $\widetilde{Z}_{i,j}(\omega) = X_{i,j}(\omega)\left(\log\widetilde{p}_{i,j}(\omega) - \log\widetilde{q}_{i,X_i|j}(\omega)\right) + \log\widetilde{q}_{i,X_i|j}(\omega) + \frac{1}{n}\log\widetilde{\pi}_j(\omega)$.

When the Bayes votes are obtained w.r.t the noisy estimates $\widetilde{\pi}(\omega)$ and $\{\widetilde{T}_i\}_{i\in[n]}$, the noisy equivalence of the expected score differences is given as:

$$\mathbb{E}\left[\widetilde{Z}_{i,1}(\omega) - \widetilde{Z}_{i,j}(\omega)\right], \tag{40}$$

$$= \sum_{k=1}^M q_{i,k|1}\log\frac{\widetilde{q}_{i,k|1}}{\widetilde{q}_{i,k|j}} + \frac{1}{n}\log\frac{\widetilde{\pi}_1}{\widetilde{\pi}_j}, \tag{41}$$

$$= \sum_{k=1}^M q_{i,k|1}\log\frac{\widetilde{q}_{i,k|1}}{q_{i,k|1}}\frac{q_{i,k|1}}{\widetilde{q}_{i,k|j}} + \frac{1}{n}\log\frac{\widetilde{\pi}_1}{\widetilde{\pi}_j}, \tag{42}$$

$$= \sum_{k=1}^M q_{i,k|1}\log\frac{q_{i,k|1}}{\widetilde{q}_{i,k|j}} - \sum_{k=1}^M q_{i,k|1}\log\frac{q_{i,k|1}}{\widetilde{q}_{i,k|1}} + \frac{1}{n}\log\frac{\widetilde{\pi}_1}{\widetilde{\pi}_j}, \tag{43}$$

$$= \mathrm{KL}\left(T_{i,|1}(\omega),\widetilde{T}_{i,|j}(\omega)\right) - \mathrm{KL}\left(T_{i,|1}(\omega),\widetilde{T}_{i,|1}(\omega)\right) + \frac{1}{n}\log\frac{\widetilde{\pi}_1(\omega)}{\widetilde{\pi}_j(\omega)}. \tag{44}$$

Let $\widetilde{\pi}_{1,j}(\omega) := \widetilde{\pi}_1(\omega)/\widetilde{\pi}_j(\omega)$. We have

$$\mathbb{E}\left[\widetilde{S}_1(\omega) - \widetilde{S}_j(\omega)\right] = \sum_{i=1}^n\mathbb{E}\left[\widetilde{Z}_{i,1}(\omega) - \widetilde{Z}_{i,j}(\omega)\right], \tag{45}$$

$$= \sum_{i=1}^n\mathrm{KL}\left(T_{i,|1}(\omega),\widetilde{T}_{i,|j}\right)(\omega) - \sum_{i=1}^n\mathrm{KL}\left(T_{i,|1}(\omega),\widetilde{T}_{i,|1}(\omega)\right) + \log\frac{\widetilde{\pi}_1(\omega)}{\widetilde{\pi}_j(\omega)}, \tag{46}$$

$$= n \cdot \Delta\overline{\mathrm{KL}}_{1,j}(\omega) + \log\widetilde{\pi}_{1,j}(\omega). \tag{47}$$

Since $\Delta\overline{\mathrm{KL}}_{1,j}(\omega) \gg n^{-1/2}$ and $\log \widetilde{\pi}_{1,j}(\omega) = \mathcal{O}(1)$, we have $n \cdot \Delta\overline{\mathrm{KL}}_{1,j}(\omega) + \log \widetilde{\pi}_{1,j}(\omega) = \Theta\left(n \cdot \Delta\overline{\mathrm{KL}}_{1,j}(\omega)\right)$ for sufficiently large $n$, implying $\forall j \geq 2 : \mathbb{E}\left[\widetilde{S}_1(\omega) - \widetilde{S}_j(\omega)\right] > 0$ eventually.

By Assumption 2, there exists a constant $\widetilde{\tau} < \infty$ such that $\forall i \in [n] : \|\log \boldsymbol{T}_i(\omega)\|_{\max} \leq \widetilde{\tau}$. We can thus establish that $\forall i \in [n]$: $\widetilde{Z}_{i,1}(\omega) - \widetilde{Z}_{i,j}(\omega) \in [-\widetilde{\tau}, \quad \widetilde{\tau}]$.

Applying Hoeffding's inequality as above, it can be shown that

$$\Pr\left(\widehat{Y}(\omega) \text{ is correct}\right) \geq 1 - \sum_{j \geq 2} \exp\left(-\frac{\left(n \cdot \Delta\overline{\mathrm{KL}}_{1,j}(\omega) + \log \widetilde{\pi}_{1,j}(\omega)\right)^2}{2n\widetilde{\tau}^2}\right), \tag{48}$$

$$= 1 - \sum_{j \geq 2} \exp\left[-\Theta\left(\frac{n \cdot \Delta\overline{\mathrm{KL}}_{1,j}^2(\omega)}{2\widetilde{\tau}^2}\right)\right]. \tag{49}$$

We obtain the desired quantity since $\Delta\overline{\mathrm{KL}}_{1,j}(\omega) \gg n^{-1/2}$, implying $n \cdot \Delta\overline{\mathrm{KL}}_{1,j}^2(\omega) \underset{N \to \infty}{\longrightarrow} \infty$. $\quad\square$

### E.4  Proof for Theorem 2

The key ingredients to the proof for Theorem 2 are the results in Lemma 1 and Lemma 2, which respectively establish the conditions for the identifiability of the parameters (by Definition 3) and consistency of the minimum Kantorovich estimator.

Lemma 1 considers a setting as follows: Suppose that there is an unobserved variable $Z$ that takes values in a $\kappa$-sized discrete domain $\{1, 2, \cdots, \kappa\}$, in which $Z$ has a non-degenerate prior $\pi_j := \mathbb{P}(Z = j) > 0$. We observe $n$ variables $\{X_i\}_{i \in [n]}$ where each $X_i$ has a finite state space $\{1, 2, \cdots, \tau_i\}$ with cardinality $\tau_i$. Let $\boldsymbol{T}_i$ be a matrix of size $\kappa \times \tau_i$ in which the $j$-th row is $[\mathbb{P}(X_i = 1 \mid Z = j), \cdots, \mathbb{P}(X_i = \tau_i \mid Z = j)]$.

The latent parameters $\theta$ contributing to the data-generating process of the observations consist of:

$$\theta := \{\boldsymbol{\pi}, \mathcal{T}\} : \quad \boldsymbol{\pi} := [\pi_j]_{j \in [\kappa]}, \quad \mathcal{T} := \{\boldsymbol{T}_i\}_{i \in [n]}.$$

**Definition 3** (Identifiability of Parameters). *The parameters $\theta(\omega)$ are identifiable if $\mathbb{P}_\theta(X(\omega)) \neq \mathbb{P}_{\theta'}(X(\omega))$, $\forall \theta(\omega) \neq \theta'(\omega)$.*

**Lemma 1** (Kruskal (1976; 1977); Sidiropoulos & Bro (2000)). *The parameters $\theta$ are identifiable up to label permutation provided that*

$$\sum_{i=1}^{n} \mathrm{Kr}(\boldsymbol{T}_i) \geq 2\kappa + n - 1, \tag{50}$$

*where $\mathrm{Kr}(\boldsymbol{T}_i)$ denotes the Kruskal rank of a matrix $\boldsymbol{T}_i$, which is defined as the largest integer $R$ such that every set of $R$ rows of the matrix is linearly independent.*

**Assumption 3** (Bassetti et al. (2006)). *Any cost function $c : \mathcal{X}^2 \to \mathbb{R}^+$ is defined by $h \circ d$ where $d$ is a distance function on $\mathcal{X}$ and $h \in \mathcal{H}$ with $\mathcal{H}$ denoting the class of all increasing, convex and continuous functions from $\mathbb{R}^+$ to $\mathbb{R}^+$, which vanish at the original and satisfy Orlicz's condition.*

**Assumption 4** (Bassetti et al. (2006)). *There exists some fixed $\theta_0$ contained in $\Theta$ such that $\mathbb{P}_{\theta_0} = \mathbb{P}_*$ and $\mathbb{P}_{\theta_0} \in \mathcal{P}_c(\mathcal{X})$ where $\mathcal{P}_c(\mathcal{X})$ is defined w.r.t a cost function $c$ as*

$$\mathcal{P}_c(\mathcal{X}) := \left\{\mu \in \mathcal{P}(\mathcal{X}) : \int_{\mathcal{X}} c(x, y)\mu(dx) < +\infty \quad \text{for some } y \in \mathcal{X}\right\}. \tag{51}$$

**Lemma 2** (Bassetti et al. (2006)). *Let $B_{\theta_0}(t) := \{\theta \in \Theta : \mathbb{P}_\theta \in \mathcal{M}, W_c(\mathbb{P}_\theta; \mathbb{P}_{\theta_0}) \leq t\}$. If*

*(i) there is some $T > 0$ such that $B_{\theta_0}(T)$ is a relatively compact set and,*

*(ii) $\mathbb{P}_{\theta_N} \Rightarrow \mathbb{P}_\theta$ for any $\theta_0$ and $(\theta_N)_{N \geq 1}$ contained in $\Theta$ such that $d_\Theta(\theta_N, \theta_0) \to 0$ as $N \to \infty$,*

*then under Assumptions 3 and 4, $\widehat{\theta}_N \to \theta_0$ $\mathbb{P}$-almost surely as $N \to \infty$.*

We are ready to prove Theorem 2.

The goal is to obtain estimates for the true parameters $\theta_*(\omega) := \{\pi_*(\omega), T_{*,1}(\omega), \cdots, T_{*,n}(\omega)\}$, where $\mathbb{P}_{\theta_*}(X(\omega)) = \mathbb{P}_*(X(\omega))$. The minimum Kantorovich estimator is the minimizer of the following OT objective over the parameter space $\Theta(\omega)$:

$$\widehat{\theta}_N(\omega) = \underset{\theta(\omega) \in \Theta(\omega)}{\arg\min} \, W_c\Big[\mathbb{P}_N\left(X(\omega) \mid \mathcal{D}\right); \mathbb{P}_\theta\left(X(\omega) \mid \mathcal{D}\right)\Big].$$

**Theorem 2.** *For any sub-structure $\omega$ with $m$ possible states, let $\widehat{\theta}_N(\omega) = \{\widehat{\pi}_N(\omega), \widehat{T}_{N,1}(\omega), \cdots, \widehat{T}_{N,n}(\omega)\}$ be the minimum Kantorovich estimates satisfying (7). As long as there are at least $(2m-1)$ informative experts, then under Assumptions 3 and 4,*

*(i) $\widehat{\pi}_N(\omega) \xrightarrow[N\to\infty]{\text{a.s}} \pi_*(\omega)$, and*

*(ii) $\forall i \in [n] :$ there exists a permutation $\sigma_i \in S_m$ such that $\widehat{T}_{N,i}(\omega) \xrightarrow[N\to\infty]{\text{a.s}} P_{\sigma_i} T_{*,i}(\omega)$,*

*where $S_m$ is the symmetric group of degree $m$ and $P_{\sigma_i}$ is the permutation matrix associated with $\sigma_i$.*

*Proof.* **We first prove the identifiability of $\theta(\omega)$.**

It is easy to see that our setting resembles one under analysis in Lemma 1: here $\kappa = \tau_1 = \cdots = \tau_n = m$. This allows us to exploit Kruskal's condition to prove the sufficiency of at least $(2m-1)$ informative and competent experts.

By the definition of informativeness in Condition 1, every pair of rows in $T_i(\omega)$ are distinct. Since any two distinct points on the $(m-1)$-dimensional simplex are linearly independent, one thus has that $\forall i \in [n] : \text{Kr}(T_i(\omega)) \geq 2$.

Thus when $n \geq 2m-1$,

$$\sum_{i=1}^n \text{Kr}(T_i) \geq 2n = n + n \geq 2m - 1 + n.$$

The inequality gives rise to Kruskal's identifiability result in (50). Invoking Lemma 1, we can identify $\theta(\omega)$ up to permutation in rows of every $T_i(\omega)$.

**We now prove the consistency $\widehat{\theta}_N(\omega)$.**

The observed distribution for noisy data $X(\omega) = \{X_i(\omega)\}_{i \in [n]}$ can be written as:

$$\mathbb{P}\left(X(\omega) \mid \mathcal{D}\right) = \sum_{j=1}^m \mathbb{P}(Y(\omega) = j \mid \mathcal{D}) \prod_{i=1}^n \mathbb{P}(X_i(\omega) \mid Y(\omega) = j, \mathcal{D}), \tag{52}$$

$$= \sum_{j=1}^m \pi_j(\omega) \prod_{i=1}^n T_{i,X_i|j}(\omega) = \pi(\omega) \cdot T_X(\omega), \tag{53}$$

where $T_X(\omega) := \left[\prod_{i=1}^n T_{i,X_i|1}(\omega), \cdots, \prod_{i=1}^n T_{i,X_i|m}(\omega)\right]^\top$ is defined as the tensor product among the $X_i(\omega)$-th columns of each $T_i(\omega)$ respectively.

With finite $(m, n)$, let us recall that the space $\mathcal{X}(\omega)$ can be defined as a finite set $\{1, 2, \cdots, m^n\}$. This means any voting profile $X(\omega)$ corresponds to some index $j \in [m^n]$. For any $\theta(\omega) \in \Theta(\omega)$, consider the induced discrete probability measure $\mathbb{P}_\theta$ over $m^n$ particles associated with weights $\{\alpha_1, \cdots, \alpha_{m^n}\}$ where every value $\alpha_j = \mathbb{P}_\theta\left(X(\omega) = j\right)$.

By Assumption 4, there exists a fixed $\theta_0(\omega) \in \Theta(\omega)$ such that $\mathbb{P}_{\theta_0}(X(\omega) \mid \mathcal{D}) = \mathbb{P}_*(X(\omega) \mid \mathcal{D})$. Let the weights over $m^n$ particles corresponding to the discrete probability measure $\mathbb{P}_{\theta_0}$ given by $\{\beta_1, \cdots, \beta_{m^n}\}$.

Let $T(\omega) := [T_j(\omega)]_{j=1}^{m^n}$. We define the metric $d_\Theta(\theta_0, \theta) = \left\|\pi_0(\omega) \cdot T_0(\omega) - \pi(\omega) \cdot T(\omega)\right\|_1$ w.r.t the usual absolute norm on $(\Delta^{m-1})^{nm+1}$.

Our OT distance can be written according to (2) as

$$W_c\big(\mathbb{P}_{\theta_0};\mathbb{P}_\theta\big) \triangleq W_c\Big[\mathbb{P}_{\theta_0}\left(X(\omega)\mid\mathcal{D}\right);\mathbb{P}_\theta\left(X(\omega)\mid\mathcal{D}\right)\Big] = \min_\gamma \sum_{j,k\in[m^n]} c_{j,k}\gamma_{j,k}, \qquad (54)$$

where $c_{j,k}$ is the transportation cost from bin $j$ to bin $k$, and $\gamma = (\tau_{j,k})_{i,k\in[m^n]} \in [0,1]^{m^n\times m^n}$ is the transport plan (or coupling) that minimizes the total transportation cost. The optimal $\gamma$ must satisfy the marginal constraints w.r.t distributions $\mathbb{P}_\theta$ and $\mathbb{P}_{\theta_0}$:

$$\forall j,k\in[m^n]: \quad \sum_{k=1}^{m^n}\gamma_{j,k}=\alpha_j, \quad \sum_{j=1}^{m^n}\tau_{j,k}=\beta_k.$$

As a result, the maximum mass that remains not transported on the diagonal is $\sum_{j=1}^{m^n}\min(\alpha_j,\beta_j)$ since the total mass is 1 i.e. $\sum_j\alpha_j=\sum_j\beta_j=1$. Thus, the total mass transported off-diagonally, across pairs $(j\neq k)$, must be at least $1-\sum_{j=1}^{m^n}\min(\alpha_j,\beta_j)$. We also have that

$$\|\alpha-\beta\|_1 = \sum_{j=1}^{m^n}|\alpha_j-\beta_j| = \sum_{j=1}^{m^n}\alpha_j+\beta_j-2\min(\alpha_j,\beta_j) = 2-2\sum_{j=1}^{m^n}\min(\alpha_j,\beta_j). \qquad (55)$$

Since $\mathcal{X}(\omega)$ is finite and the cost $c$ is non-negative, there exists a value $\eta>0$ where $c(x,y)\geq\eta$ for every $x,y\in\mathcal{X}(\omega), x\neq y$. Every unit of mass moved from some $j$ to some $k\neq j$ incurs a cost of at least $\eta$. Hence,

$$\sum_{j,k}c_{j,k}\gamma_{j,k} \geq \frac{\eta}{2}\Big\|\alpha-\beta\Big\|_1 \Rightarrow W_c\big(\mathbb{P}_{\theta_0};\mathbb{P}_\theta\big) \geq \frac{\eta}{2}\big\|\alpha-\beta\big\|_1. \qquad (56)$$

If there exists a $T>0$ such that $W_c(\mathbb{P}_{\theta_0};\mathbb{P}_\theta)\leq T$ for some $T<0$, then $\|\alpha-\beta\|_1\leq 2T/\eta$. This implies $B_{\theta_0}(T)$ is contained within a closed $L_1$-ball of radius $2T/\eta$. Then $T$ can be fixed in such a way that the above sphere is contained in $\Theta$ whenever $\theta_0$ is interior to $\Theta$, proving its relative compactness. Furthermore, the mapping $\theta\mapsto\mathbb{P}_\theta$ is continuous w.r.t the $L_1$ metric on $\Theta$, hence the weak convergence holds when $d_\Theta$ goes to zero.

Given that both conditions $(i)$ and $(ii)$ are obtained, invoking Lemma 2, we can show the consistency of the estimate $\widehat{\theta}_N(\omega)$, meaning that $\widehat{\theta}_N(\omega)$ coincides with $\theta_0(\omega)$ when $N\to\infty$.

Finally, the partial identifiability result implies that $\theta_0(\omega)$ differs from $\theta_*(\omega)$ up to label permutation. Putting all together, we have shown the convergence of $\boldsymbol{\pi}_N(\omega)$ to $\widehat{\boldsymbol{\pi}}_*(\omega)$ and each $\boldsymbol{T}_{N,i}(\omega)$ to some permutation of rows in $\widehat{\boldsymbol{T}}_{*,i}(\omega)$ in the infinite limit of sample size.

$\square$

# F    NUMERICAL RESULTS

## F.1    REAL-WORLD SYSTEMS

**Experimental Design.** We evaluate causal discovery performance using 7 real-world and 2 semi-synthetic systems with known ground-truth causal structures. The datasets are publicly accessible publicly accessible via the `bnlearn` repository.

For experiments on continuous data, we consider the real-world benchmarks: **Artic** (Huang et al., 2021), **Sachs** (Sachs et al., 2005) and **Sangiovese** (Magrini et al., 2017). We further study semi-synthetic data generated from non-linear additive noise SCMs, where the structural assignment defined in (1) takes the following form:

$$X_i := f_i\big(X_{\mathrm{pa}_i}\big) + U_i, \quad i\in[d].$$

We consider two types of systems where true structural equations $f_i$ are modelled either with Gaussian processes following a Erdos-Renyi structure **(GP-ER)**, or MLP following a Scale-Free

structure (**MLP-ER**). The noise variables have unequal variances, drawn from Uniform and Laplace distributions respectively. We refer to the systems as *semi-synthetic* since they are relatively general and widely studied in causal discovery literature. In these experiments, we consider the following causal discovery experts:

- Constraint-based: PC (Spirtes & Glymour, 1991),
- Score-based: GES-BIC – GES with BIC score (Chickering, 1996),
- Linear, non-Gaussian models: ICA-LiNGAM (Shimizu et al., 2006),
- Continuous optimization for non-linear models: NOTEARS with mean-squared-based loss; DAGMA (Bello et al., 2022) with likelihood-based loss,
- Ordering-based: SCORE (Rolland et al., 2022).

For experiments on discrete data, we analyze the datasets: **Asia** (Lauritzen, 1988), **Earthquake** (Korb & Nicholson, 2010), Child (Spiegelhalter et al., 1993), and Insurance (Binder et al., 1997) networks. In these settings, we consider only a minimum number of 3 search-based algorithms:

- Constraint-based: PC (Spirtes & Glymour, 1991),
- Score-based: GES-BIC – GES with BIC score (Chickering, 1996),
- Hill climbing search HCS (Koller & Friedman, 2009).

This goal is to study to what extent 3 experts are sufficient for effective ensemble learning in practical scenarios. It is worth recalling that theoretically, 3 experts are sufficient for (partial) identifiability of the parameters when each expert is maximally informative i.e., each competency matrix has full rank.

For continuous data, we adopt a setting where a fixed dataset is used. To generate diverse voting profiles for parameter estimation, especially for the deterministic causal discovery algorithms that generate single graph predictions, we consider the "bootstrapping" version of the expert, where the algorithm is repeatedly run on bootstrap samples i.e., subsets of the data sampled with replacement. For discrete data, we use a different approach: a large dataset is partitioned into non-overlapping subsets, and each algorithm is repeatedly run on a different partition. For both cases, we fix the number of graph samples per expert at $p = 50$.

It is important to emphasize that our selection of statistical algorithms is based purely on empirical performance. The above groups of algorithmic experts are selected because they are among the most popular and consistently high-performing methods across a variety of causal discovery benchmarks. They are also representative methods of diverse classes of causal discovery algorithms, each based on independent modelling assumptions and thereby meeting our criteria for expert diversity as discussed earlier. In fact, our framework is agnostic to the type of expert used – as long as the conditions are satisfied. This means that our approach does not exclude LLM-based experts. However, while we did test LLM-based methods on the benchmarks, they unfortunately yielded suboptimal performance and were therefore excluded from our analysis.

Regarding the ensemble-based baselines, although several ensemble-based causal discovery methods have been proposed, only Malmi et al. (2015) provide public codes for their rank-based method, which motivates its inclusion in our experiments.

**Results.**  Figures 3 to 11 report our empirical results for the real-world experiments. Table 1 reports the performance of final aggregated graphs from the proposed two-phase ensembling under different combinations of approaches, where the first step of inter-algorithm aggregation can be done by either Plurality or Rank. The methods with the superior accuracies of DAG estimation are bolded.

### F.2  SIMULATIONS

Figures 12 to 38 report the simulation studies that assess the performance of Bayes voting across various graph sizes $d \in \{10, 15, 20, 25, 30, 35, 40, 45, 50\}$ and number of graph samples per expert for estimation $p \in \{10, 30, 50\}$.

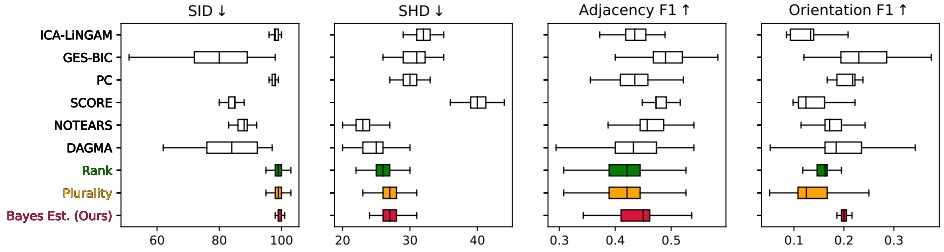

Figure 3: Experiments on **Sachs** dataset (continuous, $d = 11$).

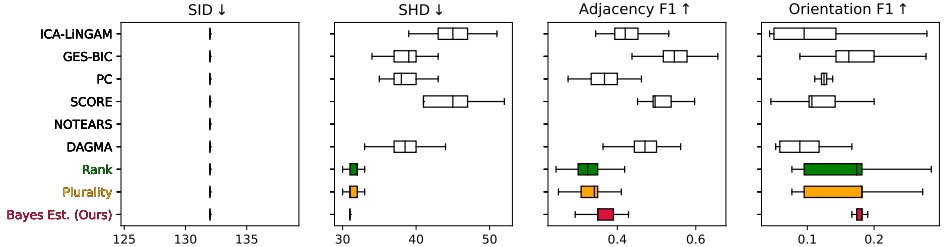

Figure 4: Experiments on **Artic** dataset (continuous, $d = 12$). SID value is upper-bounded by $d(d-1)$ and the first plot shows that all experts score the worst SID exactly of 132 at $d = 12$.

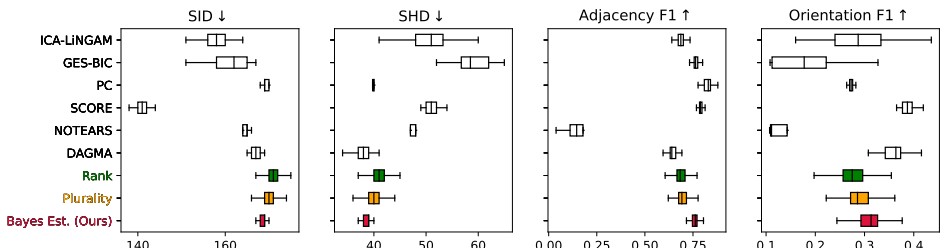

Figure 5: Experiments on **Sangiovese** dataset (continuous, $d = 14$).

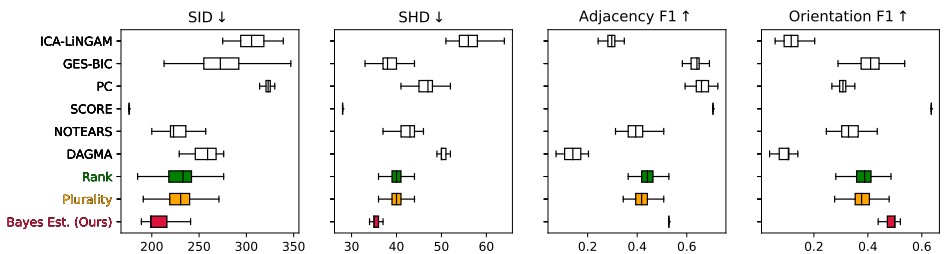

Figure 6: Experiments on **GP-ER** model (continuous, $d = 20$).

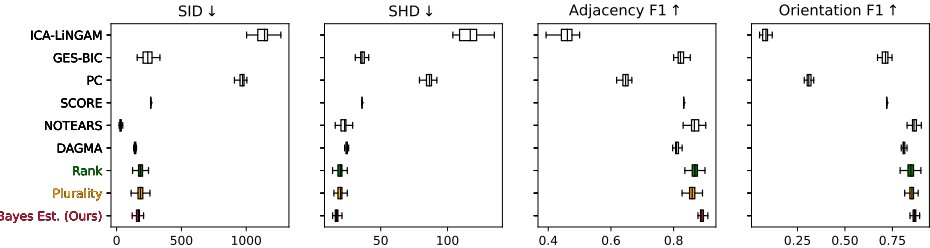

Figure 7: Experiments on **MLP-SF** model (continuous, $d = 40$).

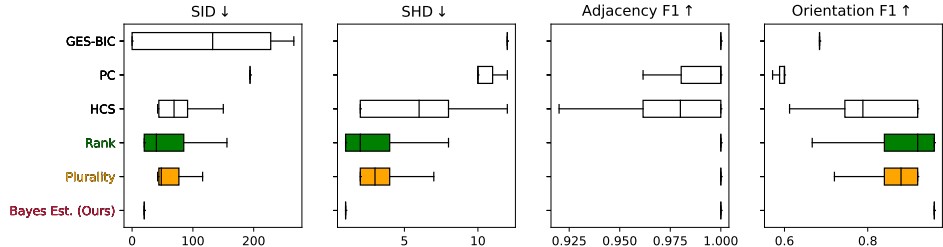

Figure 8: Experiments on **Child** dataset (discrete, $d = 20$).

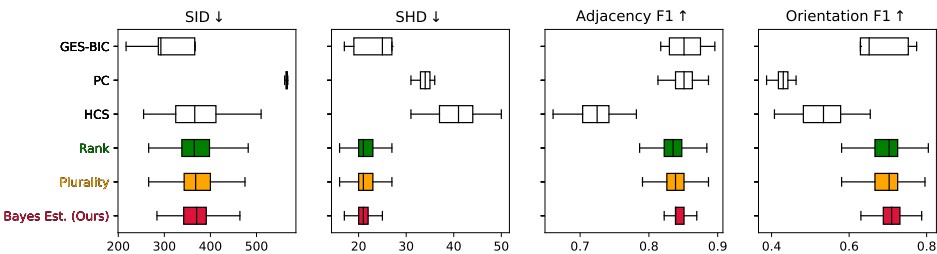

Figure 9: Experiments on **Insurance** dataset (discrete, $d = 27$).

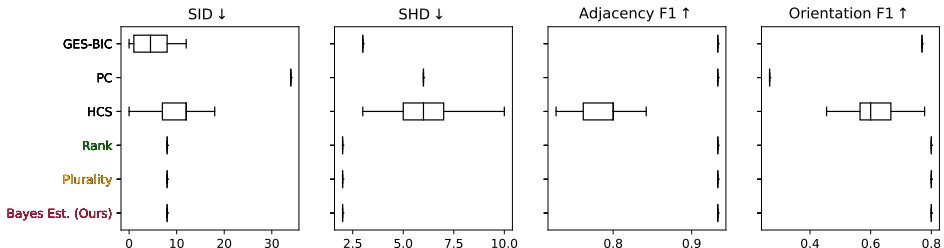

Figure 10: Experiments on **Asia** dataset (discrete, $d = 8$).

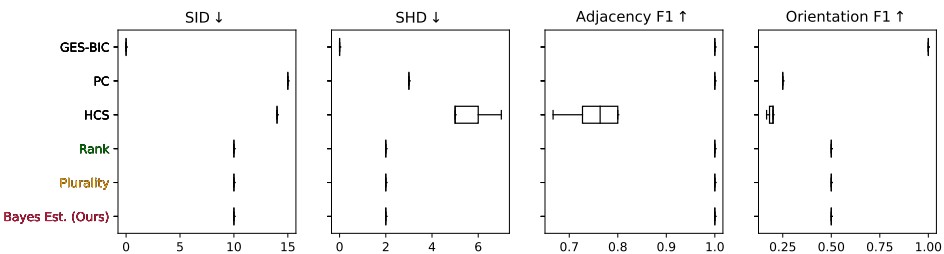

Figure 11: Experiments on **Earthquake** dataset ($d = 8$).

Table 1: Results of the final aggregated graphs from two-phase ensembling.

| Methods | SID ↓ | SHD ↓ | Average F1 ↑ |
|---|---|---|---|
| **GP-ER** (continuous, $d = 20$) | | | |
| Plurality + Rank | 204 | 38 | 0.435 |
| Plurality + Plurality | 205 | 39 | 0.412 |
| Plurality + Bayes Est. (Ours) | **199** | **35** | **0.500** |
| Rank + Rank | 205 | 39 | 0.412 |
| Rank + Plurality | 205 | 39 | 0.412 |
| Rank + Bayes Est. (Ours) | **199** | **35** | **0.500** |
| **MLP-SF** (continuous, $d = 40$) | | | |
| Plurality + Rank | 186 | 20 | 0.844 |
| Plurality + Plurality | 187 | 20 | 0.851 |
| Plurality + Bayes Est. (Ours) | 182 | 19 | 0.851 |
| Rank + Rank | 187 | 20 | 0.851 |
| Rank + Plurality | 187 | 20 | 0.851 |
| Rank + Bayes Est. (Ours) | **169** | **18** | **0.865** |
| **Sachs** (continuous, $d = 11$) | | | |
| Plurality + Rank | 99 | 26 | 0.167 |
| Plurality + Plurality | 99 | 27 | 0.162 |
| Plurality + Bayes Est. (Ours) | **99** | **27** | **0.205** |
| Rank + Rank | 99 | 26 | 0.167 |
| Rank + Plurality | 99 | 27 | 0.162 |
| Rank + Bayes Est. (Ours) | **99** | **27** | **0.205** |
| **Artic** (continuous, $d = 12$) | | | |
| Plurality + Rank | 132 | 27 | 0.211 |
| Plurality + Plurality | 132 | 27 | 0.211 |
| Plurality + Bayes Est. (Ours) | **132** | **26** | **0.241** |
| Rank + Rank | 132 | 27 | 0.211 |
| Rank + Plurality | 132 | 27 | 0.211 |
| Rank + Bayes Est. (Ours) | **132** | **26** | **0.241** |
| **Sangiovese** (continuous, $d = 14$) | | | |
| Plurality + Rank | 172 | 41 | 0.256 |
| Plurality + Plurality | 172 | 41 | 0.263 |
| Plurality + Bayes Est. (Ours) | **170** | **39** | **0.296** |
| Rank + Rank | 172 | 41 | 0.256 |
| Rank + Plurality | 172 | 41 | 0.256 |
| Rank + Bayes Est. (Ours) | **170** | **39** | **0.293** |
| **Child** (discrete, $d = 20$) | | | |
| Plurality + Rank | 20 | 1 | 0.960 |
| Plurality + Plurality | 44 | 2 | 0.920 |
| Plurality + Bayes Est. (Ours) | 20 | 1 | 0.960 |
| Rank + Rank | 20 | 1 | 0.960 |
| Rank + Plurality | 20 | 1 | 0.960 |
| Rank + Bayes Est. (Ours) | 20 | 1 | 0.960 |
| **Insurance** (discrete, $d = 27$) | | | |
| Plurality + Rank | 349 | 20 | 0.725 |
| Plurality + Plurality | 374 | 22 | 0.697 |
| Plurality + Bayes Est. (Ours) | 367 | 22 | 0.703 |
| Rank + Rank | 349 | 20 | 0.725 |
| Rank + Plurality | 349 | 20 | 0.725 |
| Rank + Bayes Est. (Ours) | **342** | **20** | **0.731** |

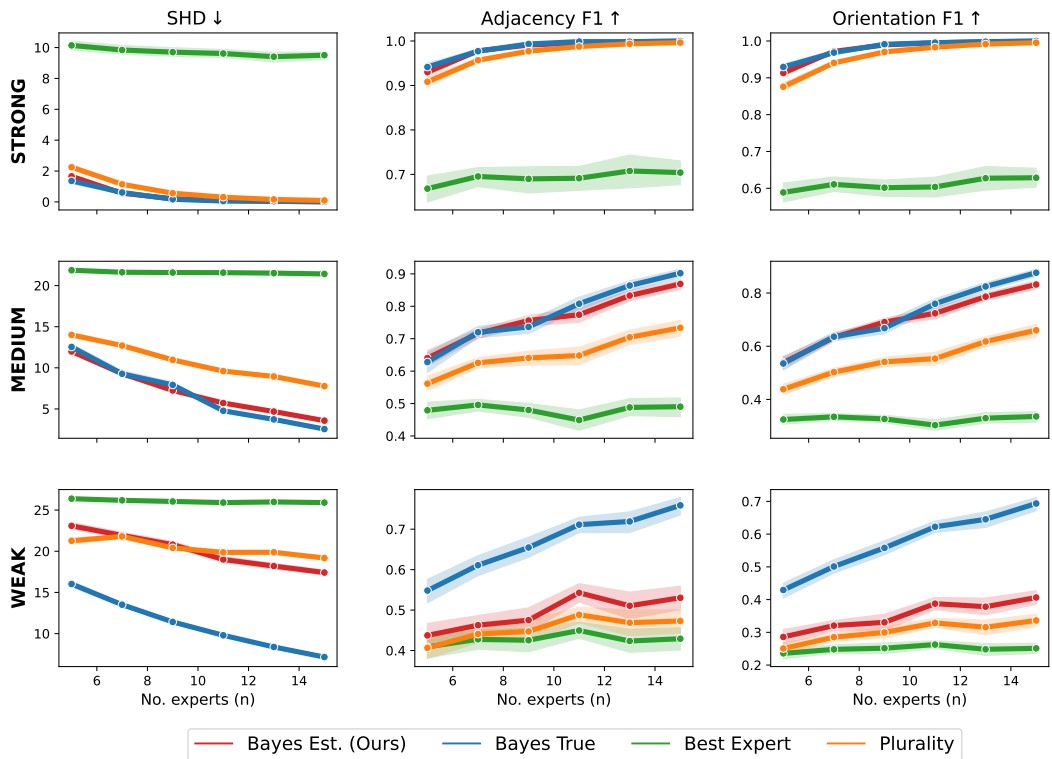

Figure 12: Simulations for graph size $d = 10$ on $50^n$ voting profiles.

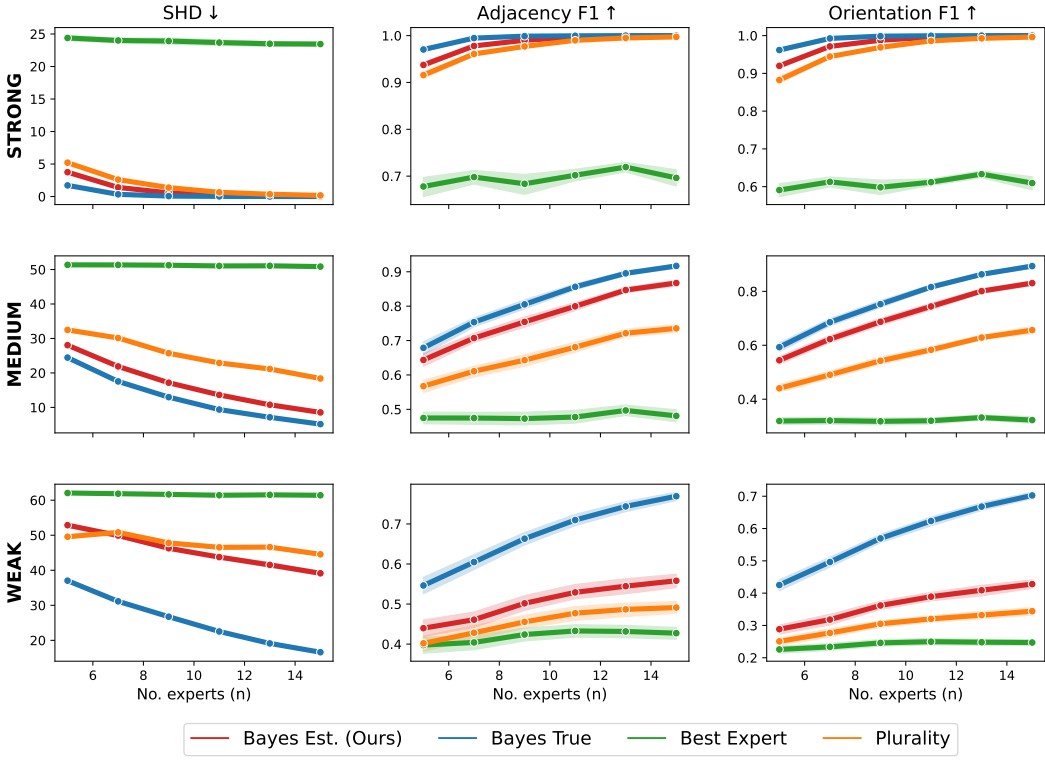

Figure 13: Simulations for graph size $d = 15$ on $50^n$ voting profiles.

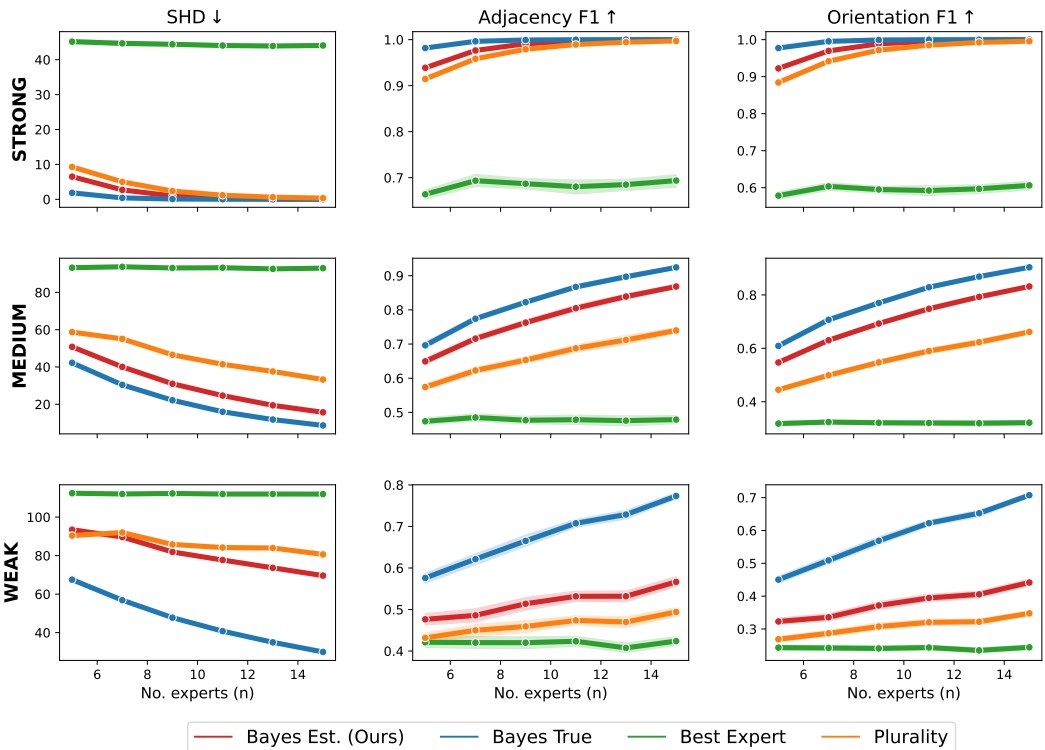

Figure 14: Simulations for graph size $d = 20$ on $50^n$ voting profiles.

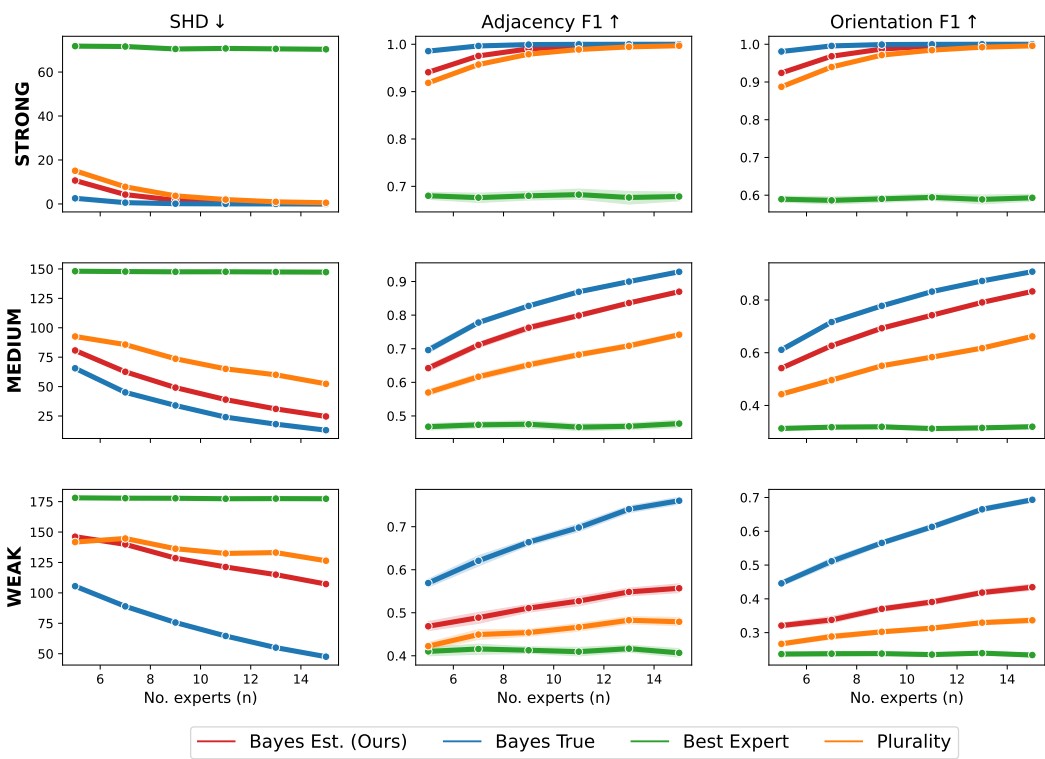

Figure 15: Simulations for graph size $d = 25$ on $50^n$ voting profiles.

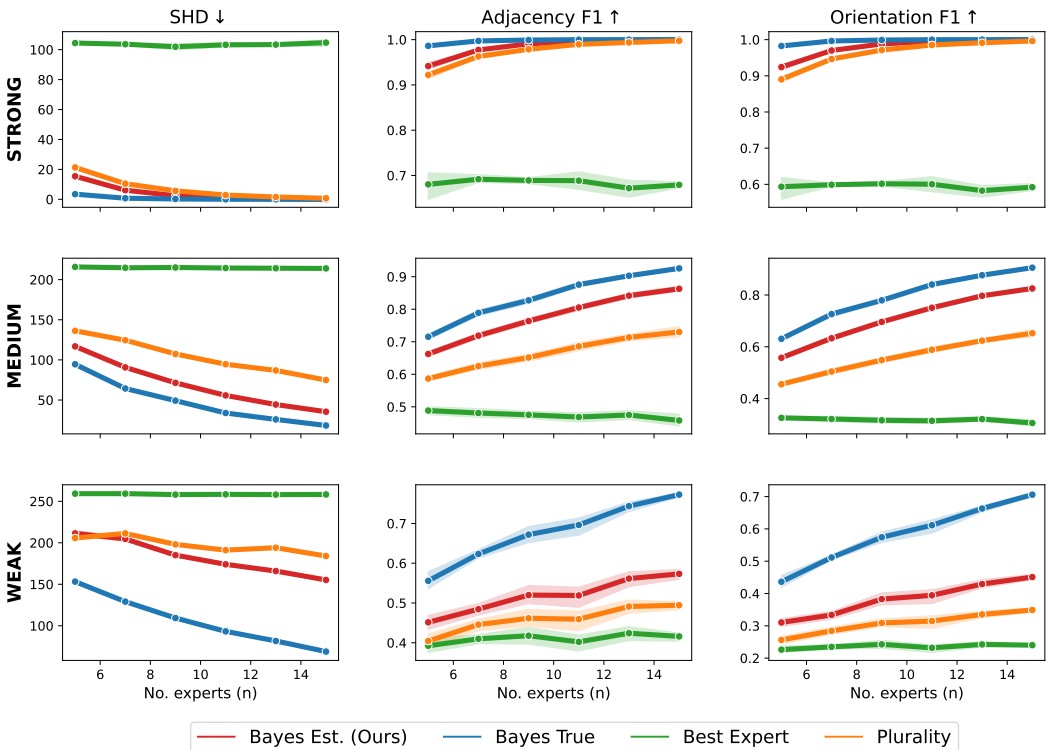

Figure 16: Simulations for graph size $d = 30$ on $50^n$ voting profiles.

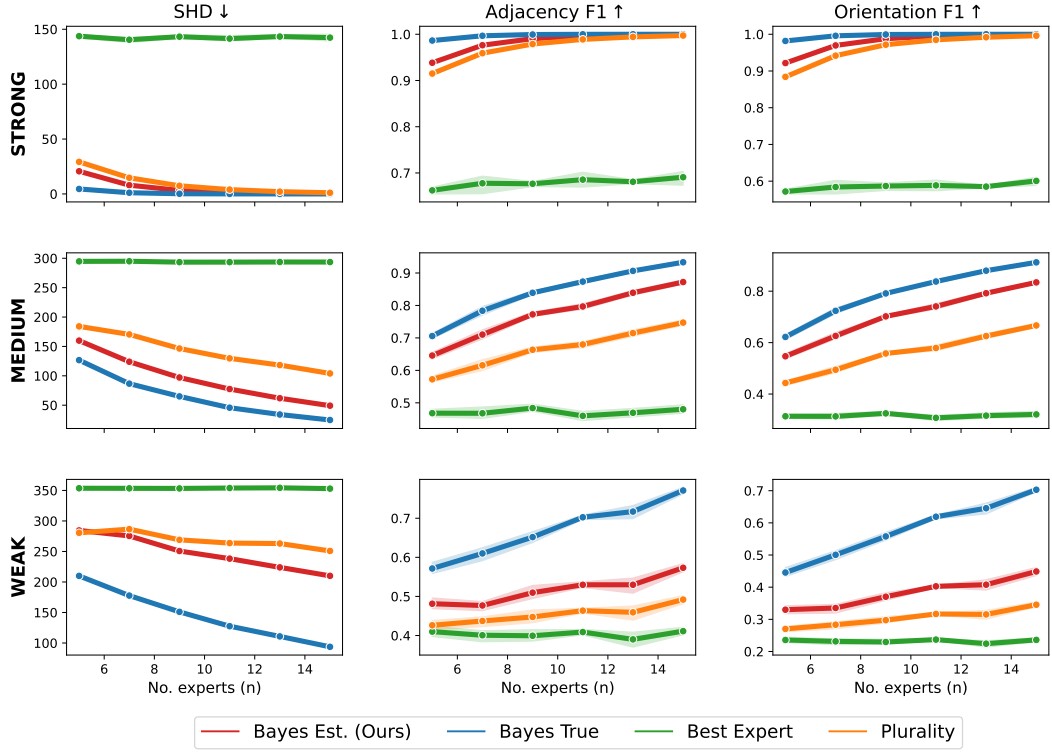

Figure 17: Simulations for graph size $d = 35$ on $50^n$ voting profiles.

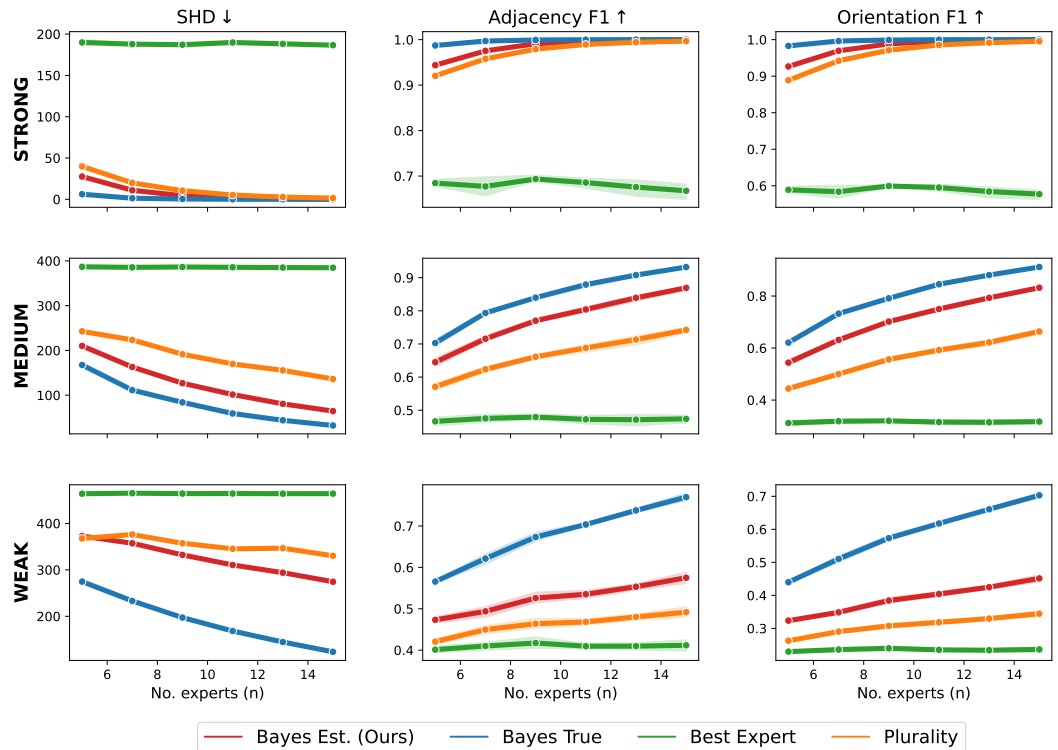

Figure 18: Simulations for graph size $d = 40$ on $50^n$ voting profiles.

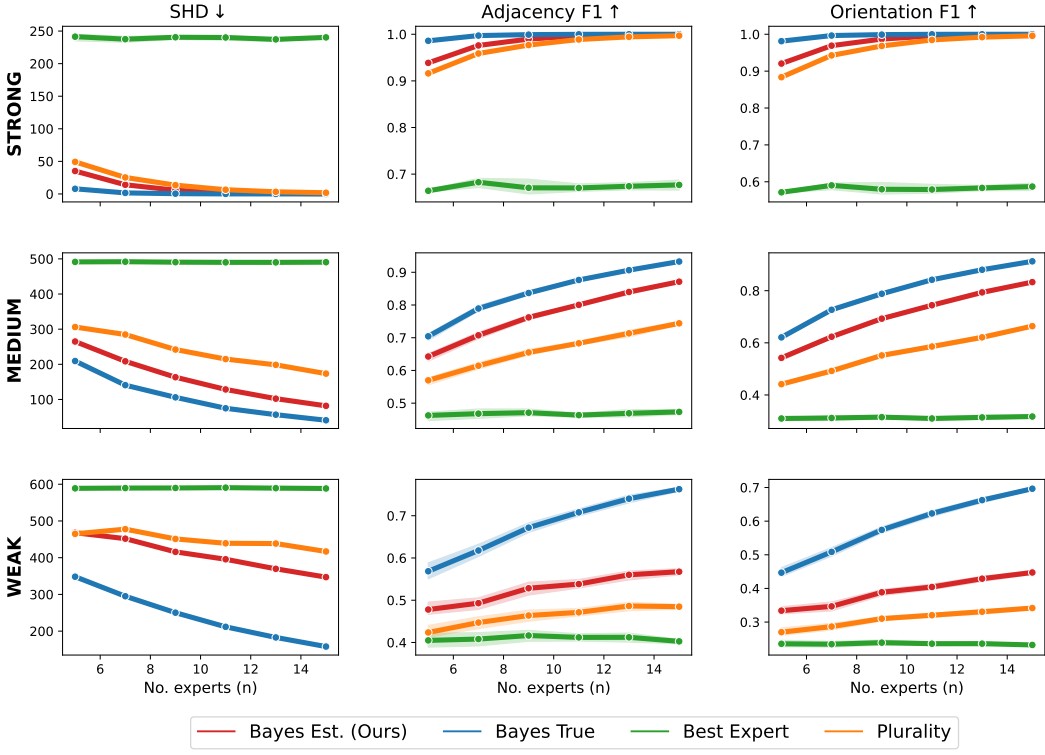

Figure 19: Simulations for graph size $d = 45$ on $50^n$ voting profiles.

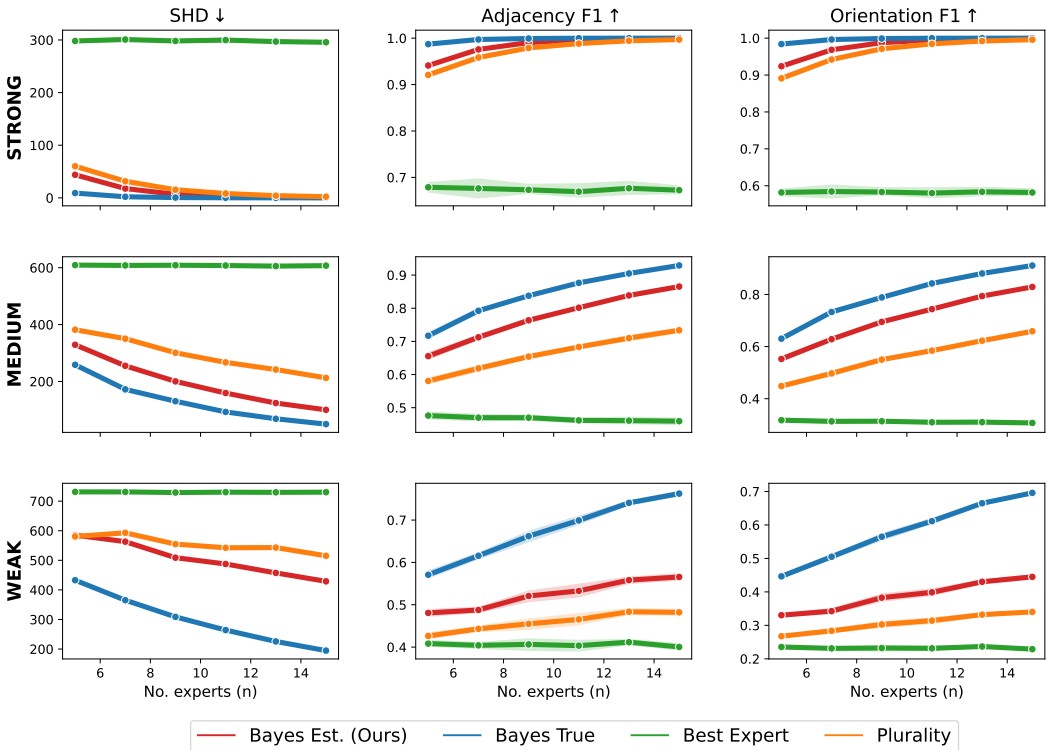

Figure 20: Simulations for graph size $d = 50$ on $50^n$ voting profiles.

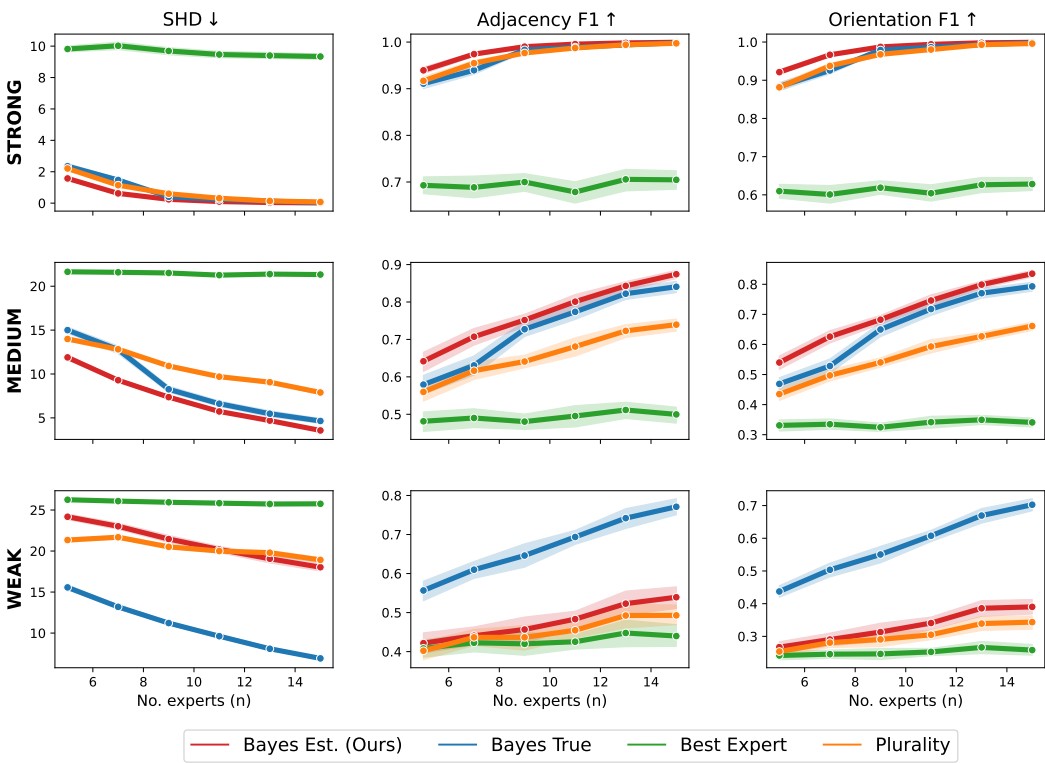

Figure 21: Simulations for graph size $d = 10$ on $30^n$ voting profiles.

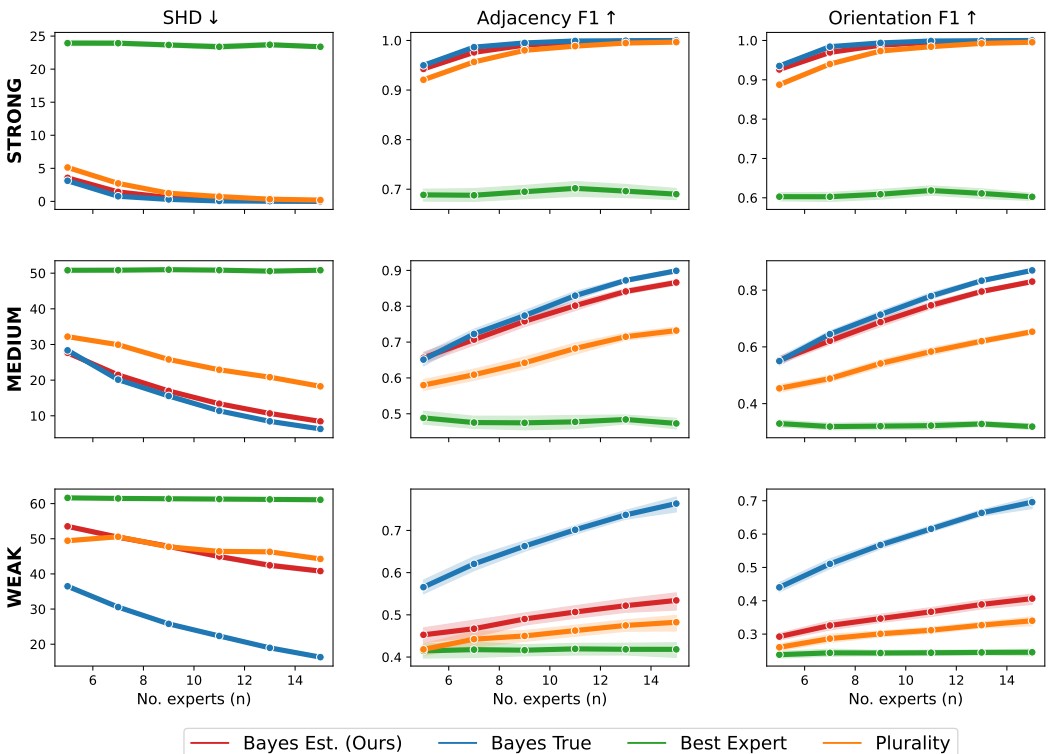

Figure 22: Simulations for graph size $d = 15$ on $30^n$ voting profiles.

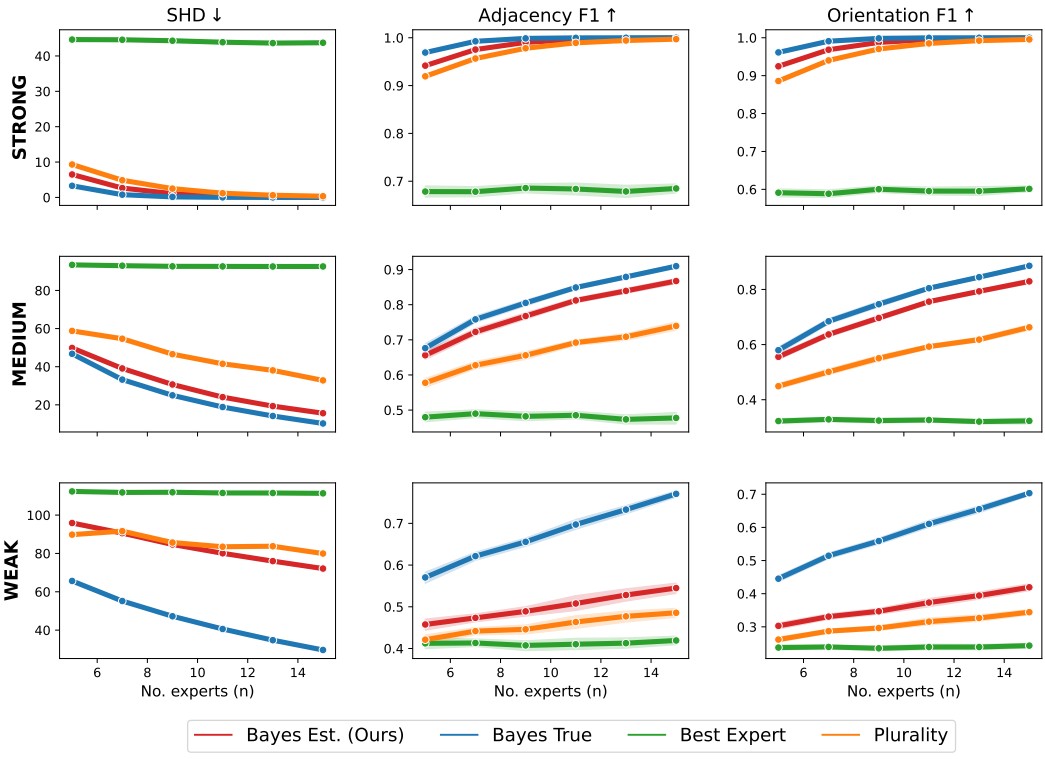

Figure 23: Simulations for graph size $d = 20$ on $30^n$ voting profiles.

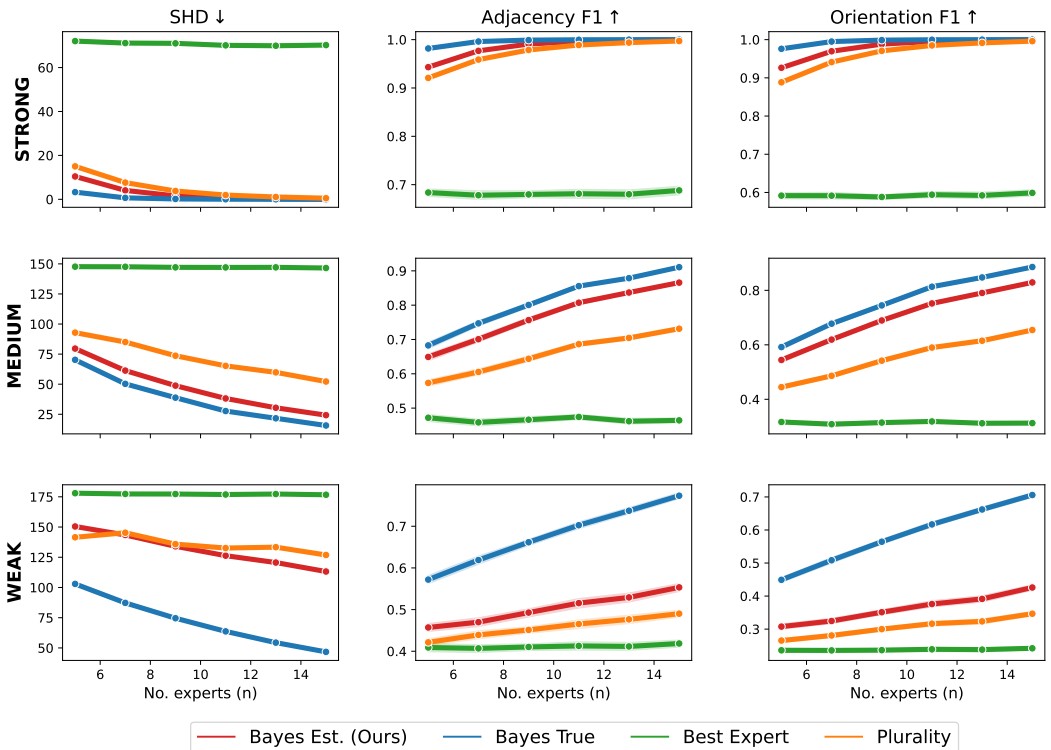

Figure 24: Simulations for graph size $d = 25$ on $30^n$ voting profiles.

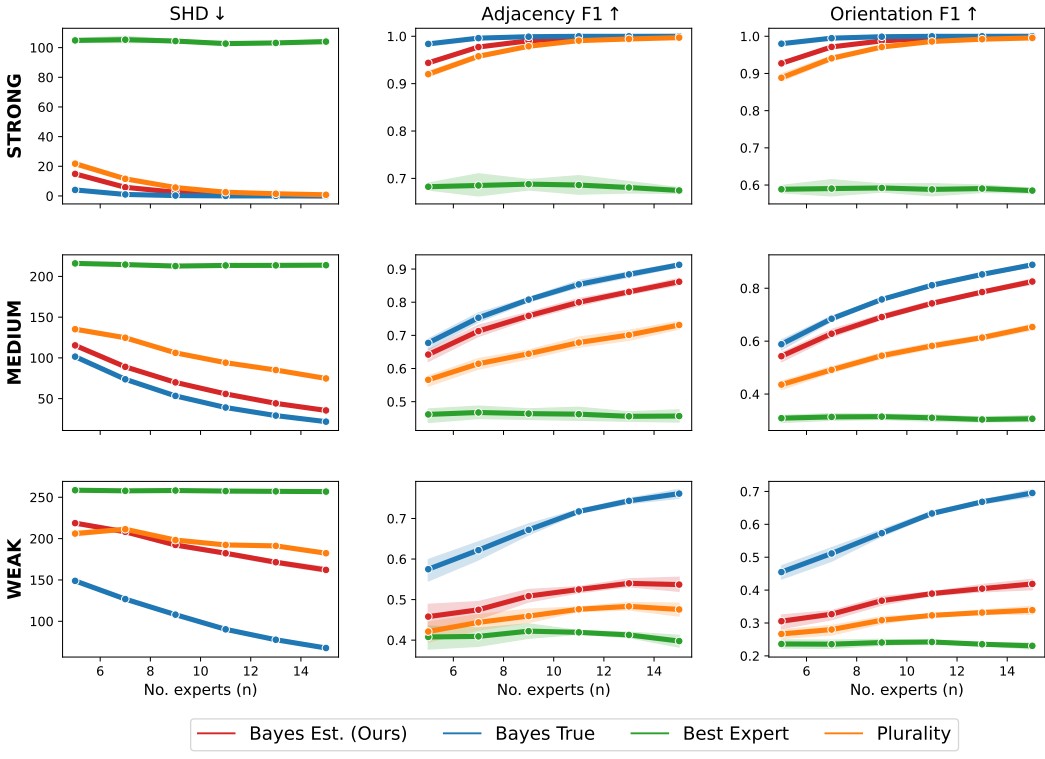

Figure 25: Simulations for graph size $d = 30$ on $30^n$ voting profiles.

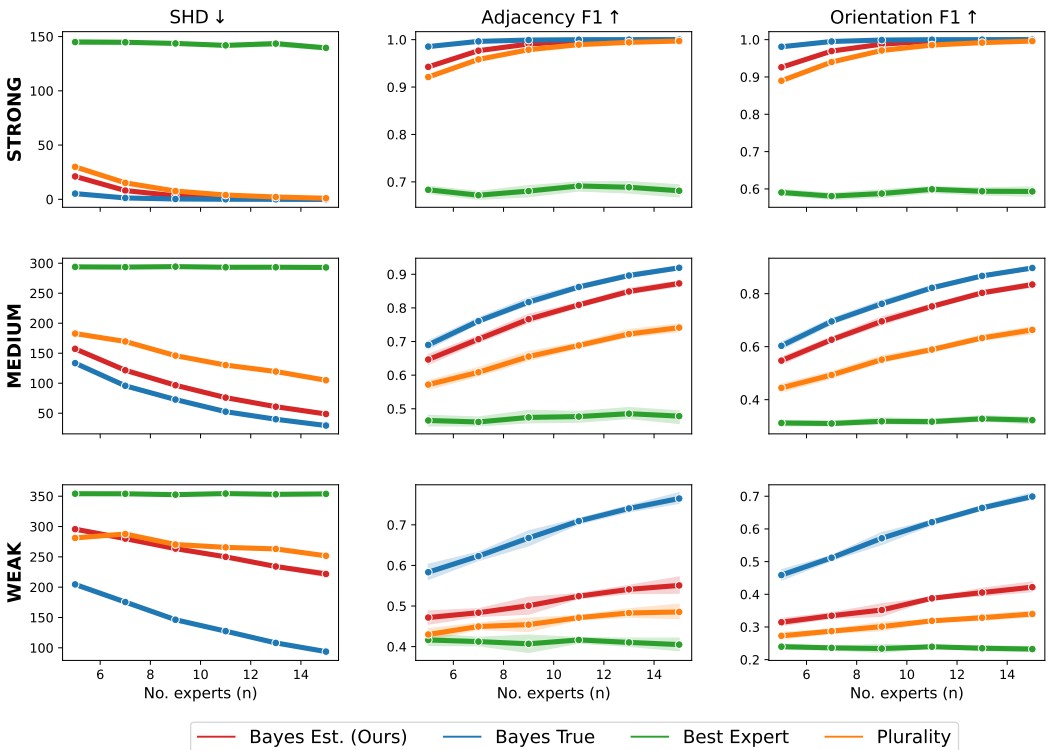

Figure 26: Simulations for graph size $d = 35$ on $30^n$ voting profiles.

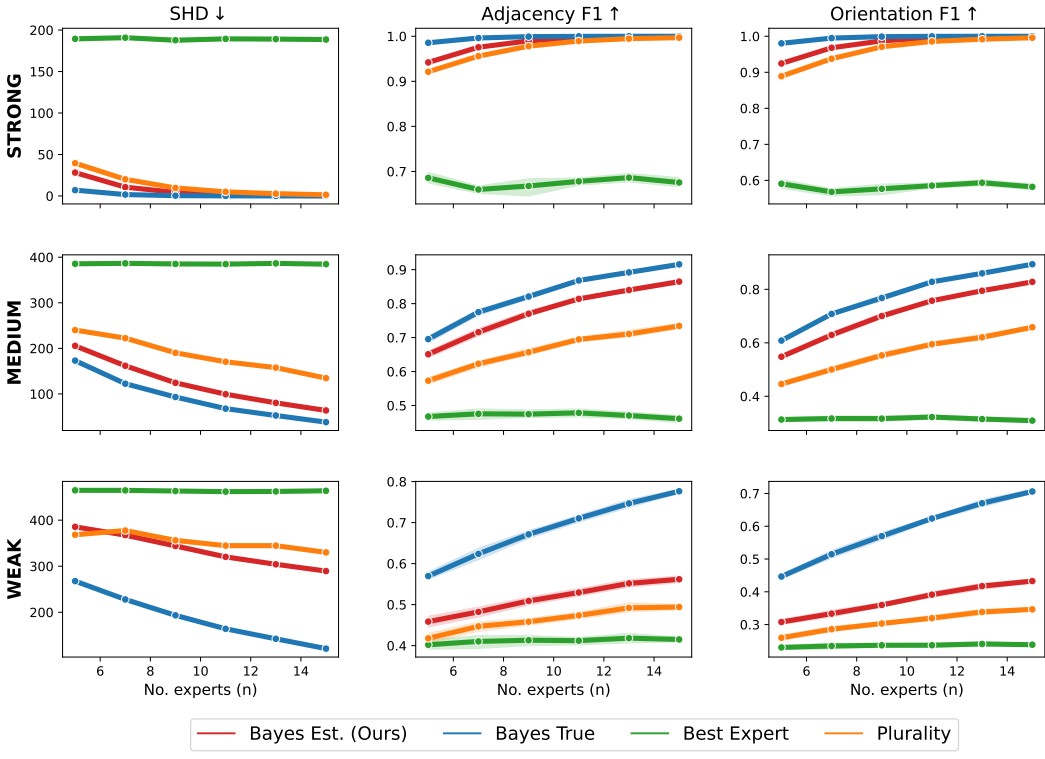

Figure 27: Simulations for graph size $d = 40$ on $30^n$ voting profiles.

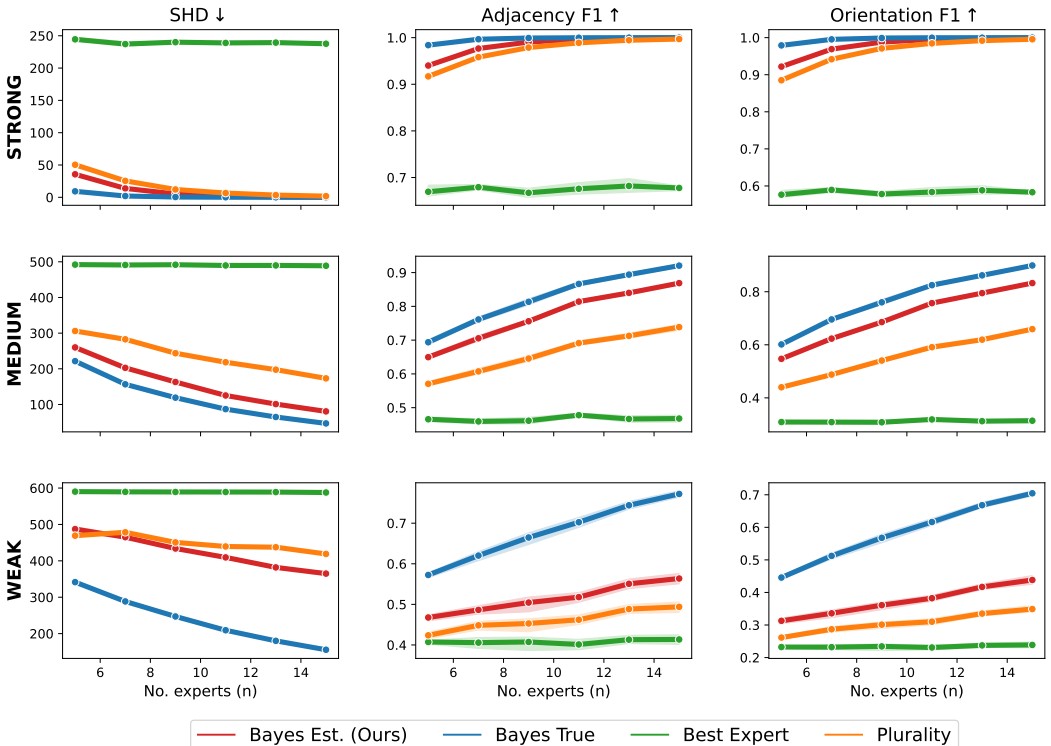

Figure 28: Simulations for graph size $d = 45$ on $30^n$ voting profiles.

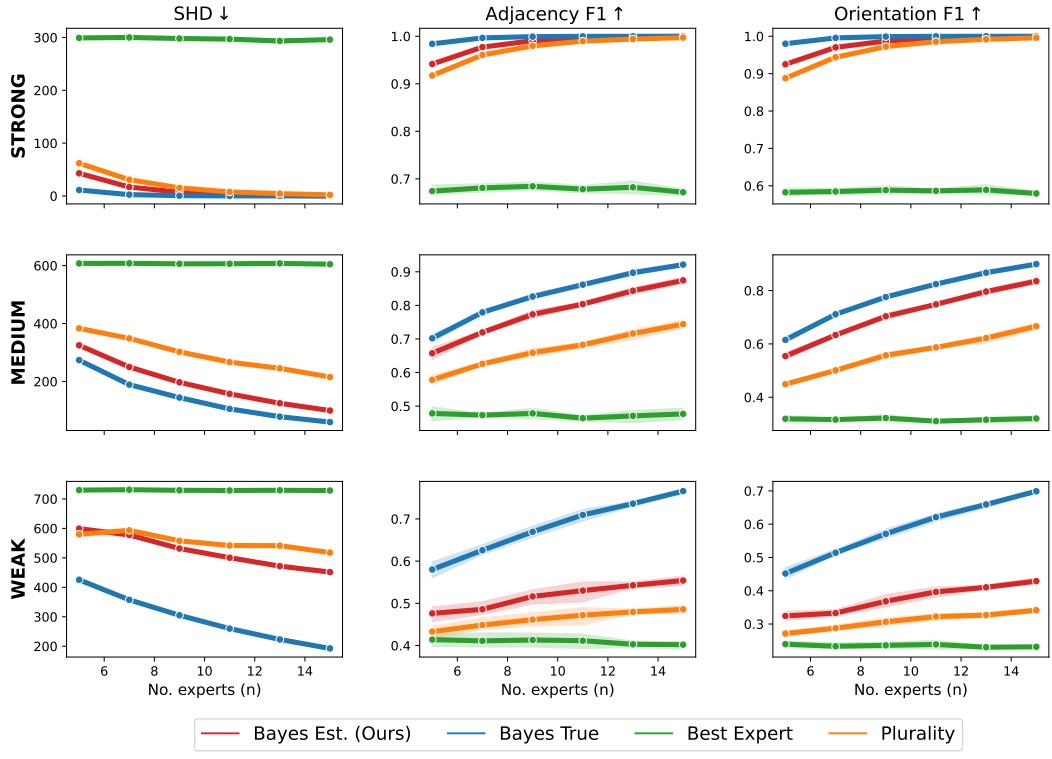

Figure 29: Simulations for graph size $d = 50$ on $30^n$ voting profiles.

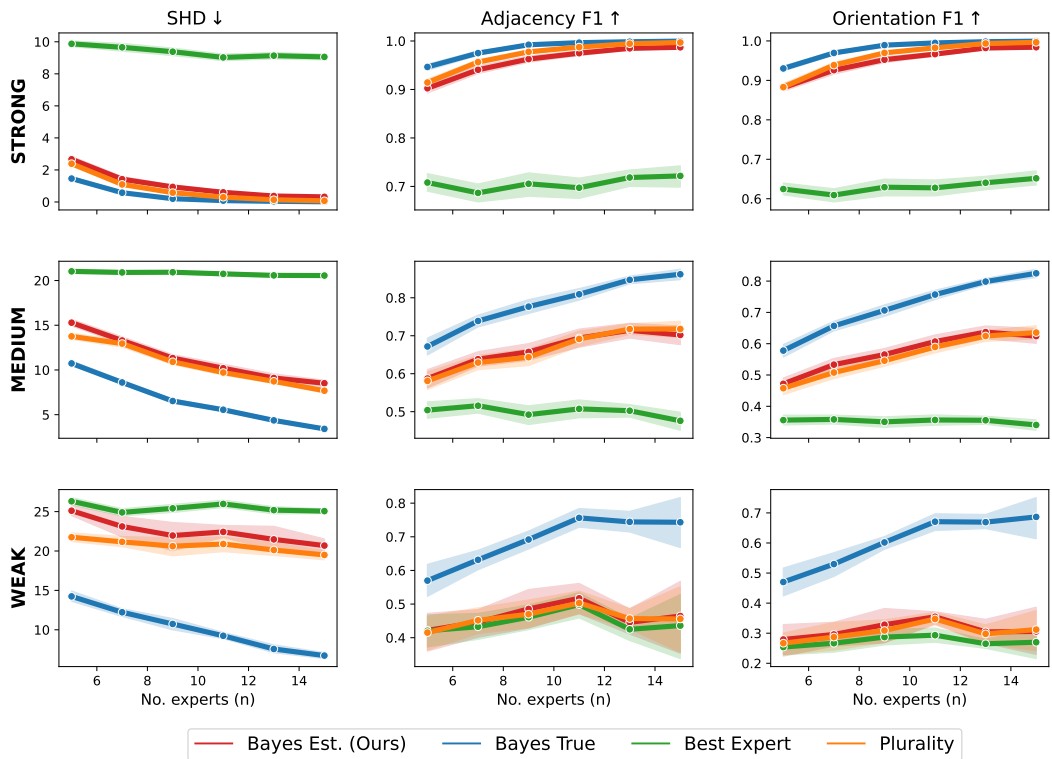

Figure 30: Simulations for graph size $d = 10$ on $10^n$ voting profiles.

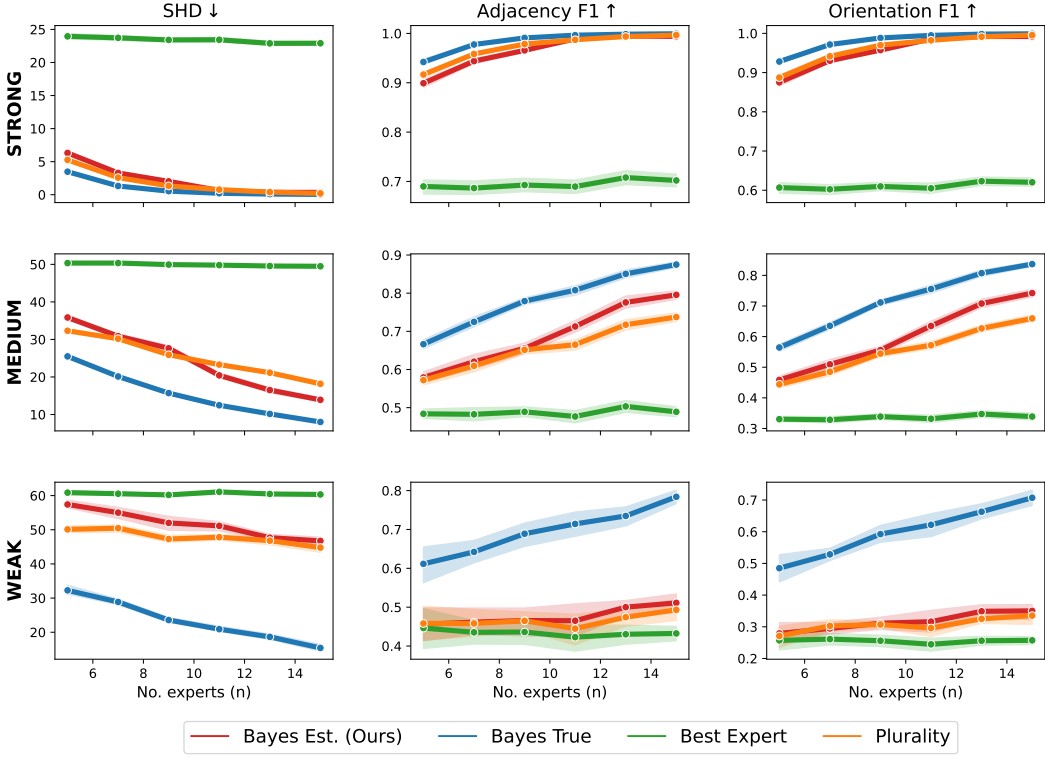

Figure 31: Simulations for graph size $d = 15$ on $10^n$ voting profiles.

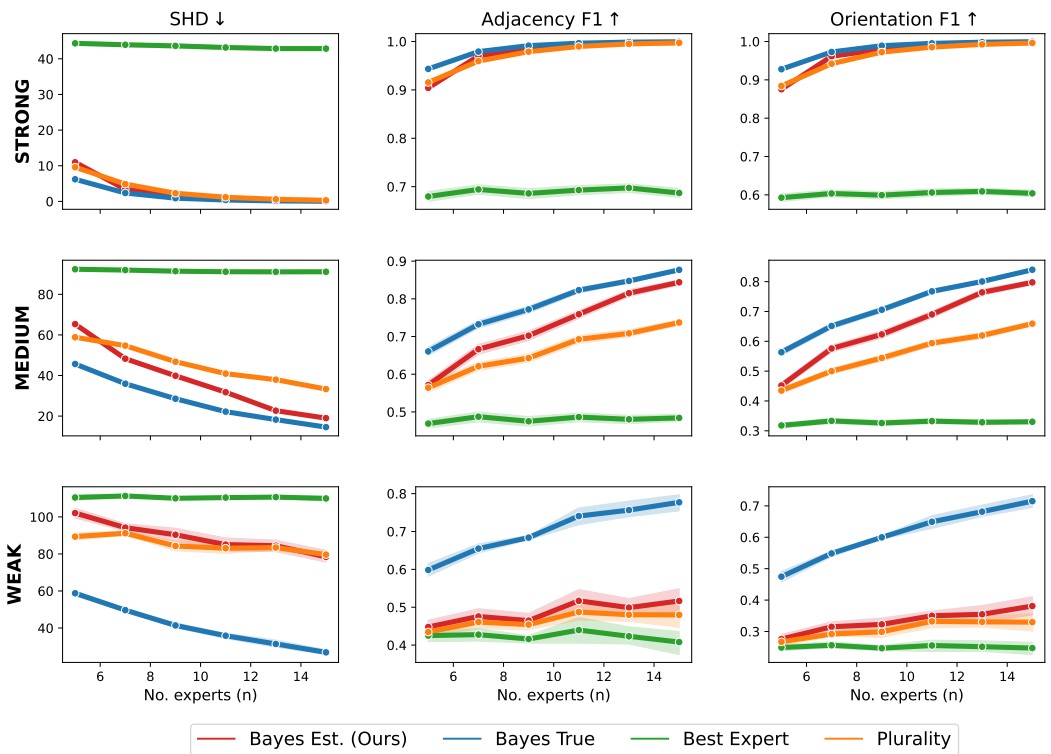

Figure 32: Simulations for graph size $d = 20$ on $10^n$ voting profiles.

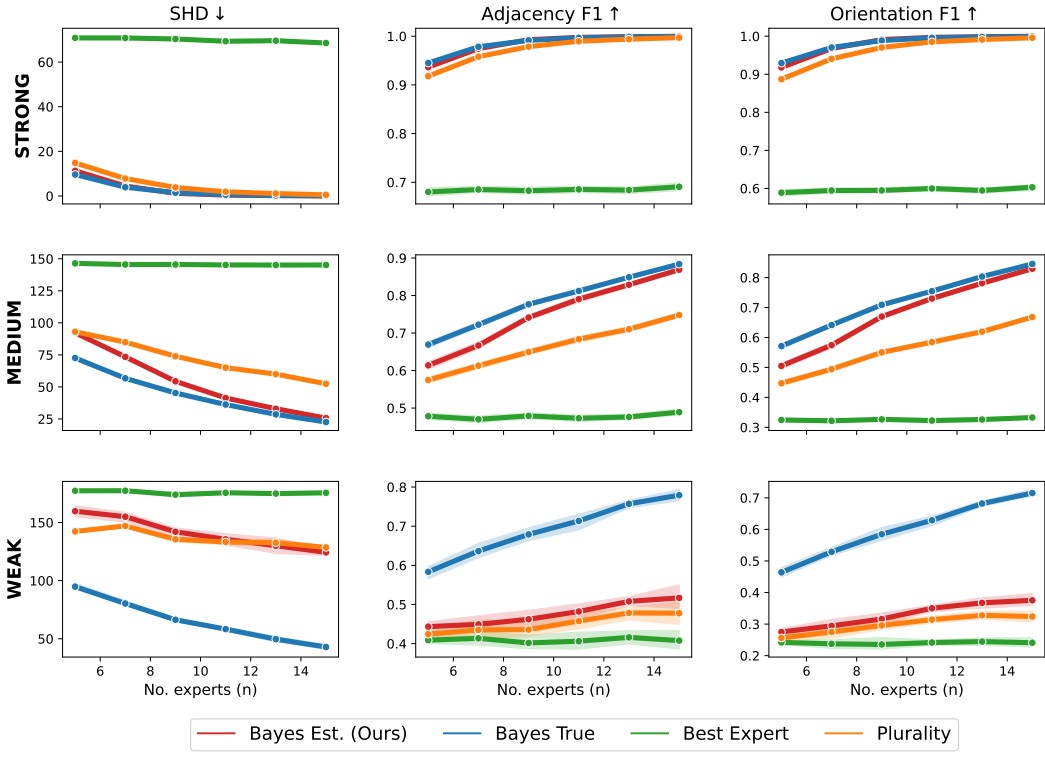

Figure 33: Simulations for graph size $d = 25$ on $10^n$ voting profiles.

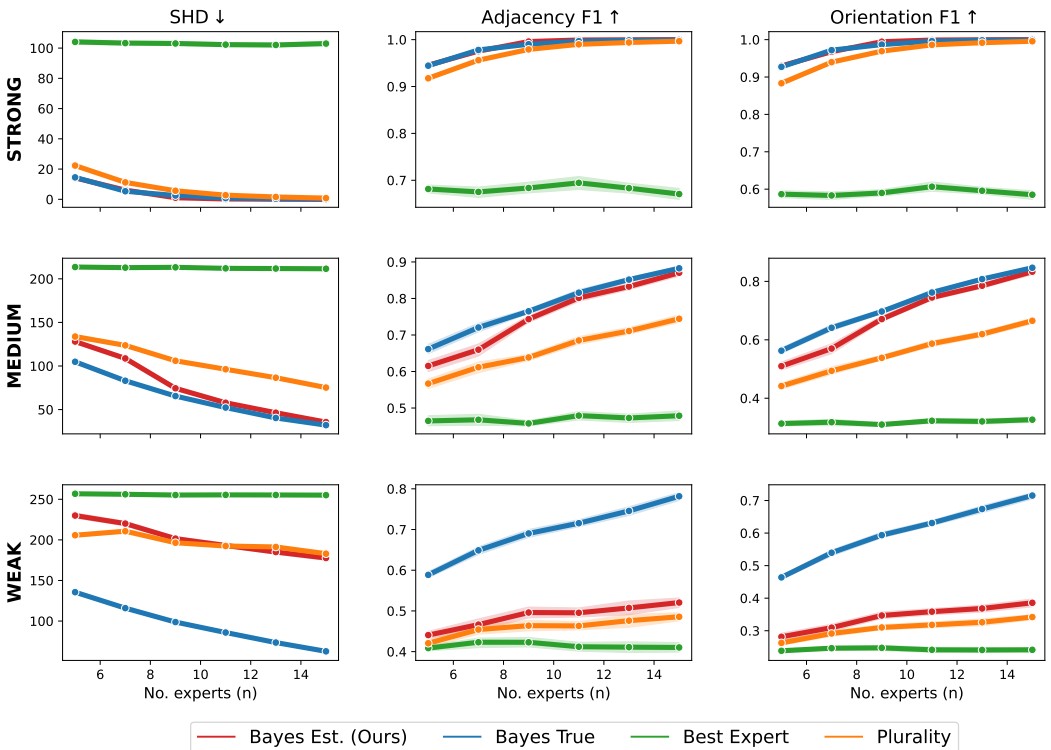

Figure 34: Simulations for graph size $d = 30$ on $10^n$ voting profiles.

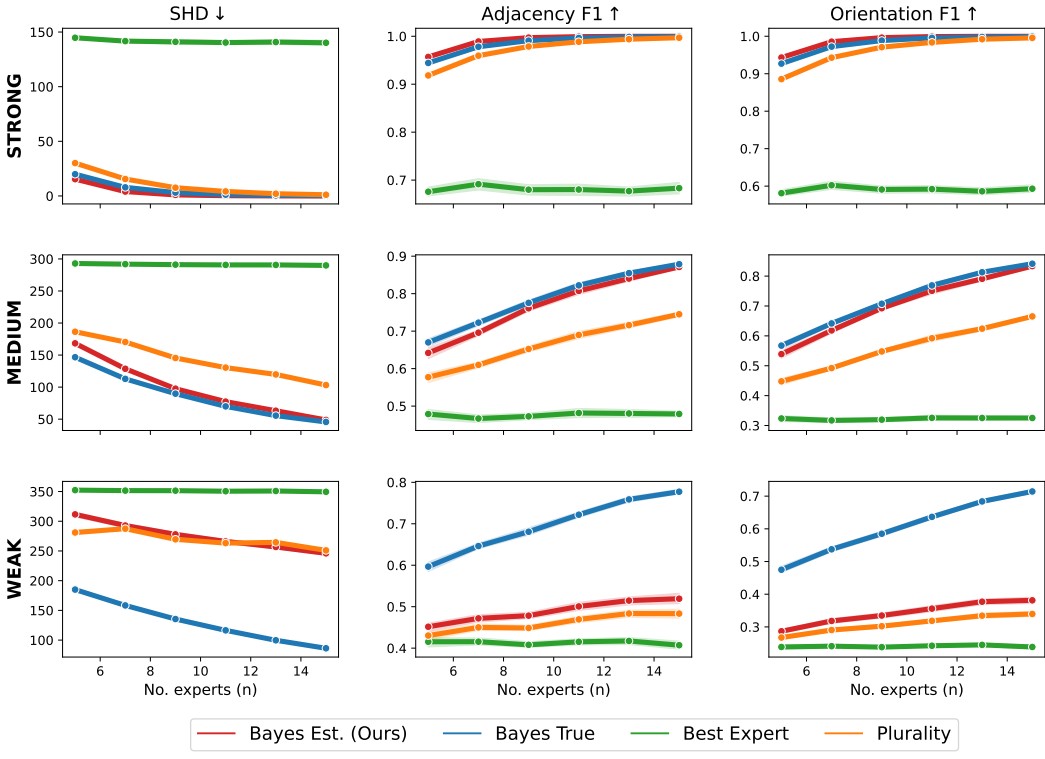

Figure 35: Simulations for graph size $d = 35$ on $10^n$ voting profiles.

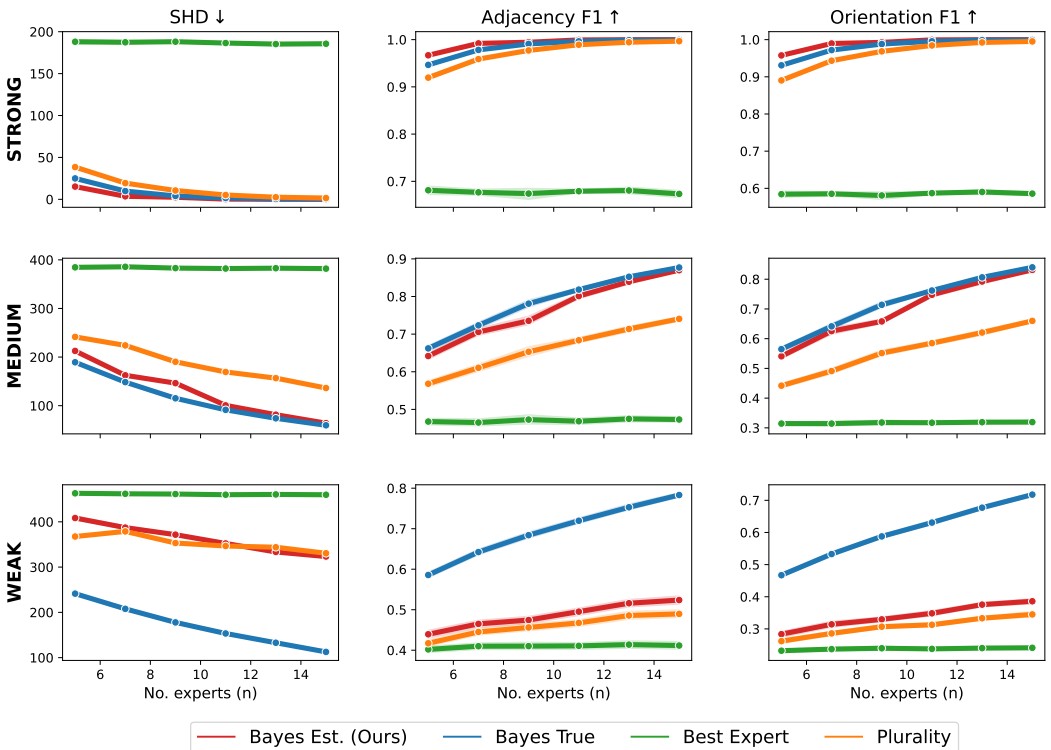

Figure 36: Simulations for graph size $d = 40$ on $10^n$ voting profiles.

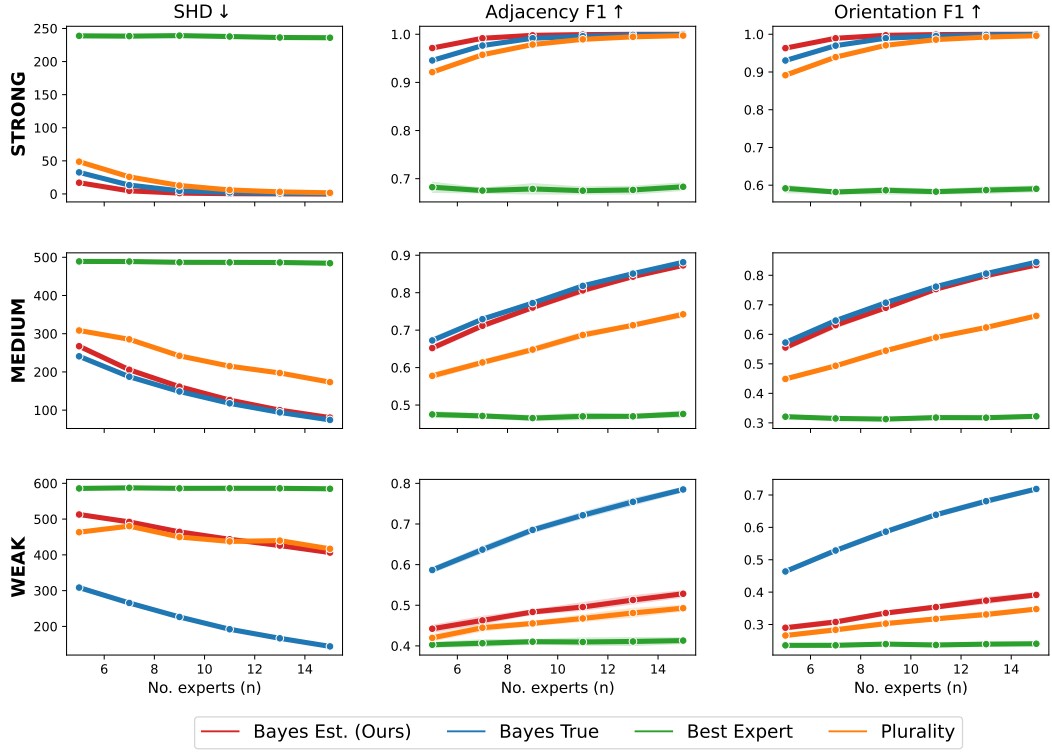

Figure 37: Simulations for graph size $d = 45$ on $10^n$ voting profiles.

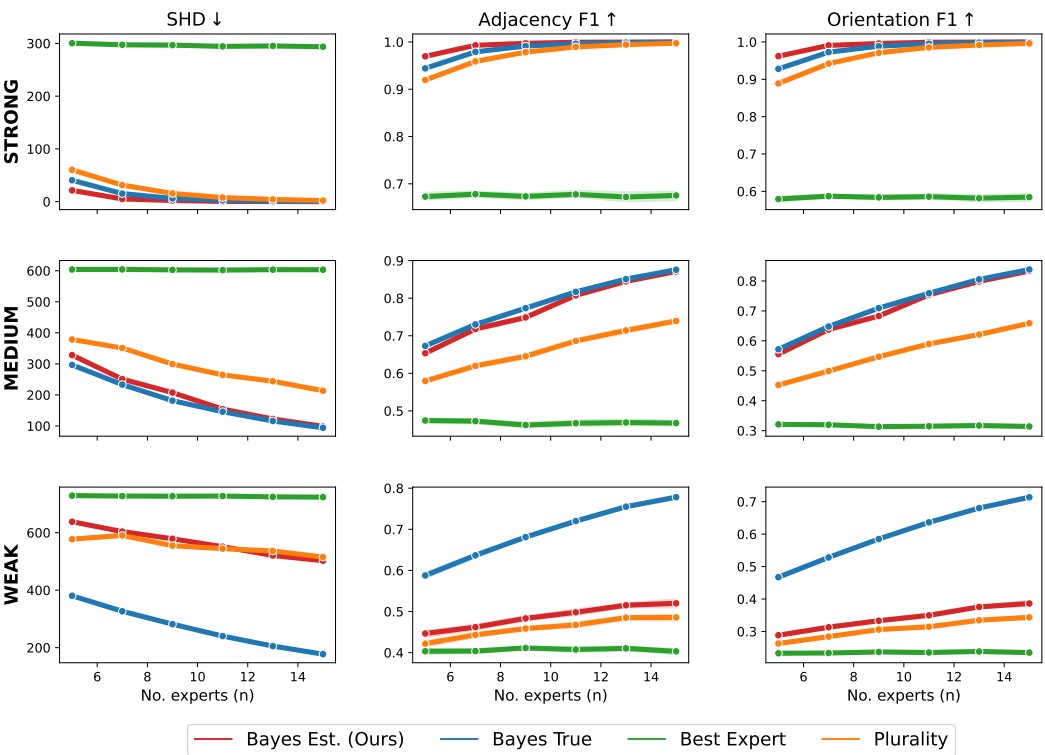

Figure 38: Simulations for graph size $d = 50$ on $10^n$ voting profiles.

