# OpenReview forum: "Causal Discovery in the Wild: A Voting-Theoretic Ensemble Approach"
_ICLR.cc/2026/Conference — ICLR 2026 Poster_

### Official Review · Reviewer_mcKH · 2025-10-27

**Soundness:** 3
**Presentation:** 3
**Contribution:** 3
**Rating:** 6
**Confidence:** 3

**Summary:**

This paper proposes a principled ensemble framework for causal discovery by treating multiple causal structure learning algorithms (or multiple runs of one algorithm) as noisy experts. It introduces a voting-theoretic model based on Bayes-optimal weighted voting to aggregate DAG structures. Theoretical analysis  of the proposed method (including the version under estimated parameters) are provided.

**Strengths:**

- This paper provides first voting-theoretic error bounds for structure ensembling in causal discovery, particularly useful as many algorithm candidates are available.
- The error bounds offer insights into how ensemble size, diversity, and competency for algorithm design affect the performance of the ensembling.
- The experiments combine synthetic and real datasets, showing consistent performance and wide algorithm coverage.

**Weaknesses:**

- the motivation of this work (or ensembling) is not about the structure learning is bad (low $\mathbb{P}(\widetilde{G} | G,\mathcal{D})$) and we want to aggregate them. But the focus is on the fact that causal dscovery methods rely on very different identifiability assumptions, thus different data generating process $\mathbb{P}(\mathcal{D} | G)$. So the "competency" is often zero once the identifiability assumptions are violated. How do we evaluate them under the same probability measure? and how does this change the problem setup and estimation of $\theta$'s?
- It is helpful to include a runing examples when introducing "feature" level to illustrate how thinking in "feature" level instead of graph level helps reduce the complexity of the problem.
- See questions.

**Questions:**

- How is $\mathbb{P}(G | \mathcal{D})$ or $\mathbb{P}(\widetilde{G} | G,\mathcal{D})$ defined? What is the mariginal distribution $\mathbb{P}(G)$ and the data generating process $\mathbb{P}(\mathcal{D} | G)$? Why is $\mathbb{P}(G | \mathcal{D})$ called "prior" rather than "posterior"?
- Why does switching the focus to "feature" level from graph level make the computation for feasible? The "feature"s are essentially sub-structure in the graph, e.g. edges. How to compare the size of two levels quantatively?
- In proposition 1, it seems we do not need $\widetilde{\pi}$ to be close to $\pi$?
- In Theorem 2, what is the intuition that we need at least $2m-1$ experts? More experts leads to more parameters to estimate. Does the estimation/identification of $\pi$ come from number of experts?

---

> ### Author Response · Authors · 2025-11-18
> **Our Responses (1/2)**
>
> We thank the reviewer for your valuable time and very insightful questions raised. We kindly categorize them and provide responses as follows:
>
> ### Clarifications about data-generating process
>
> **Notations**
> * $D$: fixed input experts receive to make predictions
> * $G$: random variable representing the true graph
> * $\tilde{G}_i$: random variable representing the noisy prediction from an $i$-th expert.
>
> &nbsp;
>
>
> **1. What is the data?**
>
> The reviewer might have misunderstood about the "data".
>
> The setting is that we have a group of causal discovery experts, each providing their prediction about the true causal structure given some fixed input $D$ (can be arbitrary, not necessarily data samples). As a result, we have a collection of graph predictions, which we call noisy graphs. The data-generating process here refers to the process of generating these noisy graph predictions, not the input $D$.
>
>
> **2. What is $P(\tilde{G} \mid G, D)$?**
>
> The next question is how to model this generative process, say the noisy graph $\tilde{G}_i$ from the $i$-th expert.
>
> Given the true graph $G$, a noisy graph $\tilde{G}_i$ is assumed to be sampled from a distribution $P(\tilde{G}_i \mid G, D)$. The distribution is parameterized by a row-stochastic vector indicating the probabilities that the noisy graph takes a certain configuration.
>
> The entry at which  $\tilde{G}_i$ aligns with the true graph $G$ represents the probability the expert predicts correctly, and incorrectly otherwise. In this sense, this vector can be seen as a representation of the expert's competence or reliability of their decision.
>
> Since the true graph is unknown, we impose uncertainty on $G$. Stacking the vectors corresponding to different possible realizations of $G$ gives us the competence transition matrix as defined in Line 132.
>
> **3. What is $P(G \mid D)$?**
>
> It refers to the distribution over the true graph $G$ conditioned on $D$. In a decision-making context, these probabilities represent prior bias or preference towards certain configurations (i.e., those with high probabilities), independent from the experts' decisions.
>
> Because $D$ is assumed fixed here, we use the term "prior" to distinguish it from the posterior $P(G \mid \tilde{G}, D)$, which is used in our scoring function (see Line 163 and Eq. 4).
>
>
> **4. So the “competency” is often zero once the identifiability assumptions are violated?**
>
> The answer is NO. As discussed, you can see that expert's competency is quantified via this row-stochastic transition matrix, so by construction **competency cannot be a zero.**
>
> Suppose the true graph is some $G^{\*}$ and $D$ are data samples. When the graph is not identifiable from $D$, a precise chacterization is that the distribution $P(\tilde{G} \mid G^{*}, D)$ does not peak at $G^{\*}$, and the probabilities spread out across various options. This means that the expert is more likely to make mistakes, but the probability at the correct entry can still be non-zero.

---

> ### Author Response · Authors · 2025-11-18
> **Our Responses (2/2)**
>
> ### Questions about feature level
>
>
> **1. How thinking in “feature” level instead of graph level helps reduce the complexity of the problem.**
>
> At the graph level, Bayes voting requires computing, for every possible graph, a score determined by the prior distribution over graphs and each expert’s competence transition matrix. However, this is infeasible: for a graph with $d$ nodes, the number of possible graph configurations is as enormous as $2^{d^2}$, and each competence matrix is of size $2^{d^2} \times 2^{d^2}$. Even storing such matrices is impossible for realistic graph sizes, let alone estimating or using them.
>
>
>
> To address this, we decompose the graph into features i.e., a
> grouping of a subset of nodes, where the number of nodes in the grouping is called a feature level, denoted as $l$. Each feature is thus a local substructre over $l$ nodes and presumably has a set of $m$ connectivity states.
>
>
> Instead of performing Bayes voting over the entire graph, we apply Bayes voting *feature by feature*. For each substructure, we  require only (i) the marginal prior over its $m$ states and (ii) the marginal competence matrix, which is of size $m \times m$.
>
> This results in a dramatic reduction in complexity. Rather than estimating a massive global competence matrix, we estimate competence matrices only for local features.
>
> For a graph of $d$ nodes, a choice of level $l$ gives rise to a feature space $\Omega$ of size $\binom{d}{l}$, where $\Omega$ refers to the set of features at a given decomposition level. The total number of parameters becomes $\binom{d}{l} \cdot m^{2}$. Because $l$ is under our control, we choose a level that is computationally feasible. In this work, we show that $l = 2$  i.e., features corresponding to individual edges -- is not only tractable but also sufficient and, feasibly ideal for aggregation.
>
>
>
> **2. How to compare the size of two levels quantitatively?**
>
> If we understand correctly, we guess that the reviewer refers to the size of $\Omega$, which is $\binom dl$ as explained above.
>
>
> &nbsp;
>
> ### Questions about $\pi$
>
> **1. In proposition 1, it seems we do not need $\hat{\pi}$ to be close to $\pi$?**
>
> Yes, that is correct. The beauty of voting theory is that we do not need to estimate the scoring quantities perfectly on an element-wise basis. It is sufficient that the scores be correct up to proportionality such that the true label receives the highest votes.
>
> &nbsp;
>
> **2. In Theorem 2, what is the intuition that we need at least $2m - 1$ experts? Does the estimation/identification of $\pi$ come from number of experts?**
>
> Yes, the condition of at least $2m-1$ experts applies to the identification and estimation of both $\pi$ and the matrices $\{T\}_{i \in [n]}$.
>
> Here is one intuitive way to think about the identifiability condition:
>
> In our setting, we observe only the noisy graph predictions produced by the composition of hidden factors: the competence transition matrices and prior probability vector. Each expert acts as a separate "view", providing a distinct source of information about these latent factors.
>
> Each view offers a footprint or snapshot of the underlying components. The informativeness of a view is determined by the degree of linear independence among the rows of its transition matrix: the more linearly independent these rows are, the richer the information the view provides. Identifiability asks how much information, or how many such views, is required to uniquely recover the latent factors.
>
> The intuition is straightforward: with more views, it becomes increasingly difficult for any alternative set of latent factors to reproduce all observed data simultaneously. When these views are sufficiently rich (suffiency quantified by conditions like ours), no alternative decomposition can explain the observations, and the latent components can be uniquely identified.
>
> **More experts leads to more parameters to estimate.**
>
> Yes, that is correct. While reducing the problem to the feature level makes it tractable, it does not eliminate the underlying difficulties. We still must handle important constraints involving parameter identifiability, consistent estimation, and computational trade-offs. Our paper offers a comprehensive solution that addresses each of these concerns systematically.
>
> ---
> ### Thank you!
> We sincerely appreciate the reviewer’s time and effort in evaluating our work. We hope to have adequately addressed your concerns. Kindly let us know if you have any further questions.

---

### Official Review · Reviewer_94Bz · 2025-10-30

**Soundness:** 3
**Presentation:** 3
**Contribution:** 2
**Rating:** 6
**Confidence:** 4

**Summary:**

The authors study ensemble-based causal discovery, where the results of several causal discovery algorithms are aggregated into a single final graph. In particular, they divide the graph into different "features", such as possible edges, and perform feature-wise aggregation. Following a typical aggregation approach, they assume there are $n$ independent experts, and seek to approximate the Bayes optimal voting rule by estimating the transition probabilities of each expert on each feature and the prior over each feature. To estimate these quantities, they assume iid samples from each expert, and use minimum distance estimation.

The authors provide two sets of experimental results. In the first set of results, they use a simulation where, instead of using causal discovery algorithms, they specify each expert by a deterministic (but unknown) transition matrix from the true graph to its predicted graph, and show that their procedure outperforms both the best expert and the plurality voting rule. In the second set of results, they consider actual causal discovery algorithms such as ICA-LiNGAM, GES-BIC, PC, SCORE, NOTEARS, and DAGMA, on 9 benchmarks from the bnlearn repository, with the proposed method achieving the best results on the SHD metric and competitive results on Adjacency F1 and Orientation F1 metrics.

**Strengths:**

**Originality:** To the best of my knowledge, the use of more sophisticated aggregation techniques for causal discovery is relatively novel.

**Quality:** The paper is of good quality, with a good balance of theoretical motivation and experimental investigation.

**Clarity:** The motivation for going beyond plurality voting and other simple aggregation techniques is well-articulated.

**Significance:** The results are promising and may help push causal discovery towards more practical reliability.

**Weaknesses:**

**Theorem presentation/novelty:** Theorem 1 in this paper seems to be an extension of Theorem 1 in [1] to multi-class classification and unbalanced truths. Overall, I think the nature of both Theorems 1 and 2 are obscured by an early overcommitment to the causal discovery setting: the nature of the results is really about general aggregation of experts with categorical outputs. I'm not an expert in the area, but aggregation is a very well-studied subject in statistical learning theory, and the paper would benefit from contextualizing itself within this field before turning the attention to causal discovery. A better understanding of aggregation would also address some issues with the theoretical assumptions, discussed in the next point.

**Theorem assumptions:** A key assumption (Assumption 1) is that the experts make independent decisions (conditioned on the truth). However, in practice, each of the causal discovery experts would be using the same dataset (or, you would have to reduce the sample size and divide a dataset of size $M$ into $n$ datasets of size $M/n$), i.e., their decisions are only independent when conditioning on both the ground truth *and* the dataset. Further, to estimate the expert transition matrices, the theory requires another $N$ independent voting profiles from each expert. Overall, these assumptions are far different from how any aggregation method would be used in practice, and seem to be too strongly based on intuitions from crowdsourcing/voting theory. At the very least, I think it is essential that the paper emphasizes these limitations and better describes the mismatch between theory and practice (e.g., the experiments use bootstrapping to get different graph samples for each expert, which would not be independent). Ideally, the authors would be able to use more sophisticated analysis to handle these issues, but in reality I think it's fair to leave that direction for future work.

[1] Berend et al. (2015), A Finite Sample Analysis of the Naive Bayes Classifier

**Questions:**

1. Is the SID plot in Figure 1 an error?
2. Since you aggregate at the feature level, there could be issues with combining the aggregated features into a single graph. For example, with plurality voting, if expert 1 gives the graph (1->2, 2->3), expert 2 gives the graph (2->3,3->1), and expert 3 gives the graph (3->1,1->2), then the aggregated features would be 1->2, 2->3, and 3->1, so combining these features gives a cyclic (rather than acyclic) graph. How do you handle such issues? If you don't, what is the justification?

---

> ### Author Response · Authors · 2025-11-18
> **Theoretical Questions (1/2)**
>
> Thank you for your valuable time and the support of an acceptance. We kindly provide our responses as follows:
>
> &nbsp;
>
> ## About theoretical presentation
>
> Theorem 1 indeed generalizes classic results on the optimal decision rule for
> unsupervised multiclass classification setting with unbalanced labels, and we are aware of the result in [1]. The causal discovery setting also introduces several specific challenges that we aim to highlight in our work. These aspects include the intractable space of classes that motivates feature-level aggregation, parameter identifiability, estimation consistency, robustness under noisy estimation and computational trade-offs, which prevent a direct application of the existing results.
>
> Because multiclass classification is a well-established problem in machine learning, we assumed readers would already be familiar with the setting, thus decided to present the theory directly in the context of causal discovery to keep the paper focused, especially given the page limit.
>
> Based on the overall feedback from the reviewers, we agree with that a clearer discussion of the problem framing would improve readability. We have revised the paragraph at the beginning Section 3.1 for this purpose, which will be updated in the final paper when an extra page is given.
>
> *"Essentially, we have a body of $n$ decision-making experts, each voting for one alternative $\widetilde{G}_i$ out of $|\mathcal{G|}$ possible options. The voting setup can be cast as a general problem of multi-class classification with unbalanced truth over an exponentially large space of $|\mathcal{G}|$ classes. Thus, recovering the true causal structure in our setting is equivalent to finding the optimal decision rule mapping from the data of $n$ noisy predictions to the ground-truth graph label, a classic unsupervised learning problem now placed over a combinatorially large label space typical of causal discovery settings. We first introduce our proposed vote aggregation rule and establish the conditions for recovering the true causal structure from its noisy predictions."*
>
>
>
>
> &nbsp;
>
> ## About theoretical results
>
>
> 1/ Thank you for pointing it out. We would like to clarify that, in Assumption 1, the correct statement is that **the experts’ decisions are independent conditional on both the ground truth and the input data $D$**. Since $D$ is fixed and conditioned on throughout the analysis, we did not explicitly state this in the original text. As can be verified in our proof, we indeed rely on this joint conditioning for the factorization - NOT conditioning on the ground truth alone.
>
>
> &nbsp;
>
> 2/ Because of this, from Section 4, we made the conditioning on $D$ implicit and dropped the notation to keep the presentation clean. Precisely, all related quantities should be conditional
> i.e., $P_{N}(X(\omega) \mid D)$ or $P_{\theta}(X(\omega) \mid D)$. So, we in fact only require the graph samples to be i.i.d **conditionally** on the original input $D$, which can be satisfied with boostrap samples. The usage of bootstrapping comes with the supposition that given the knowledge from $D$, the variation in these graph samples captures experts' competency-related factors in making decisions.
>
> The reviewer has raised a very insightful point about scenarios where the bootstrap samples may be (conditionally) non i.i.d. We agree that such situations necessitate the use of sophisticated methods like block bootstrapping. Given the maturity of statistical theory, we believe that these situations are manageable within the existing framework.
>
> We recognize that ambiguity in our notations led to a misunderstanding, and we thank the reviewer for highlighting these useful comments. We have revised the notation accordingly and reflected on the role of i.i.d assumption in the discussion on limitations.

---

> > ### Comment · Reviewer_94Bz · 2025-11-23
> >
> > Thank you for the clarifications.
> >
> > I think the revised paragraph at the beginning of Section 3.1 will be helpful. I agree that the combinatorially large label space is the interesting part compared to the traditional settings, and this aspect should be emphasized as part of the novelty.
> >
> > Thanks for clarifying the theoretical assumption. Mathematically, I think it's better to keep the conditioning on $D$ explicit - I understand the desire for notational simplicity, but it shouldn't be accomplished at the sacrifice of mathematical correctness/clarity. I am happy to hear that you've revised the notation to accurately reflect the theoretical assumptions. The conditional independence given the data is a much easier pill to swallow; I'm not too concerned about the more complicated bootstrapping scenarios, but agree it is worth mentioning.

---

> ### Author Response · Authors · 2025-11-18
> **Other Questions (2/2)**
>
> **1. Is the SID plot in Figure 1 an error?**
>
> SID quantifies the distance between two structures in terms of their induced intervention distributions. So yes, it is an error-type metric: the lower SID, the better.
>
> &nbsp;
>
> **2. How do you handle cyclic graph? Justifications?**
>
> It is first important to emphasize that our theoretical results apply to generic structure learning problems, since in practice
> we often do not know whether the underlying graph is acyclic or contains feedback loops. So precisely speaking, a situation like this would occur when one had prior knowledge, like acyclicity, about the true structure and the question would be how our framework accomodates such constraints.
>
> From a theoretical standpoint, if the true graph is acyclic, then under the ideal conditions specified in Theorem 1 or Proposition 1, namely a large number of experts with sufficient competence, the aggregated graph should itself be (or at least nearly) acyclic. Remarkably, this is exactly what we observe in our experiments: our reported final aggregated graphs remain acyclic even without imposing any acyclicity constraints.
>
> From an practical standpoint, we agree that conflicts might arise due to computational complexities. Our empirical suggestion is that one can incorporate this information in a post-processing step. Many causal discovery algorithms like NOTEARS or DAGMA output a weighted (instead of binary) adjacency matrix. A common practice is to threshold the matrix and prune low-weight edges until acyclicity is achieved. In our setting, the aggregation procedure naturally yields posterior probabilities over the states of each feature (e.g., edge), which directly quantify the reliability of all experts' prediction. One could apply the same thresholding heuristic to remove low-confidence edges.
>
> ---
> ### Thank you!
> We sincerely appreciate the reviewer’s time and effort in evaluating our work. We hope to have adequately addressed your concerns. Kindly let us know if you have any further questions.

---

> > ### Comment · Reviewer_94Bz · 2025-11-23
> >
> > 1. Sorry, let me clarify the question. I know that SID is an an error metric and I misreferenced the figure. I meant to refer to Figure 2, where the left plot (for SID) only contains a line rather than a box.
> >
> > 2. This response is sensible and I think these points should be mentioned in the paper. For downstream tasks, many users may care about whether the returned graph is cyclic, so it would be valuable to include the empirical suggestion of thresholding as an optional part of the code.

---

> ### Author Response · Authors · 2025-11-24
> **Clarification on SID in Figure 2**
>
> Regarding Figure 2, the flat lines in some figures mean that the method produces identical results across all runs. The boxes indicate that there is variation in the results.
>
> ---
> Thank you for your prompt responses and all the valuable comments. We have started incorporating the revisions into the updated paper. Feel free to let us know if you have any further concerns or questions.

---

> > ### Comment · Reviewer_94Bz · 2025-11-25
> >
> > I understood that the flat line would indicate that the results are identical across all runs. However, I find it highly unlikely that **all** methods have the **same** SID with **no variability** across runs, especially when the other metrics have non-negligible variability across methods and runs. It is much more likely that there is a bug in the computation of the SID. Please carefully check this result.

---

> ### Author Response · Authors · 2025-11-26
>
> Thank you for your engagement in the discussion. Your valuable time is highly appreciated. We provide explanations as follows:
>
> **We believe our computation of SID is correct.**
>
> We first note that there are many other figures (namely Fig. 9 and 10) in which the behavior of SID agrees with other metrics. The SID values across all experiments are computed using the **same implementation**. Specifically, the computation of SID is based on the standard R package provided by Peters and Bühlmann (2013). We did not modify the codes and directly ran R codes to evaluate SID. All we did was to supply the adjacency matrices and we did carefully check that the corresponding arguments for the true and estimated graphs are correct.
>
> ### 1. How is SID computed?
>
> The situation highlighted by the reviewer can indeed occur due to the nature of SID formulation. To understand why, let us briefly recap how it is defined.
>
> Given a true graph $G$ and an estimated graph $H$, we say the intervention distribution from node $i$ to node $j$ is correctly inferred by $H$ if for *all* observational distributions that are Markov w.r.t $G$,
> $$p_{G}(x_j | do(X_i = x_i)) = p_{H}(x_j | do(X_i = x_i))$$
>
> In Proposition 7,  Peters and Bühlmann (2013) also explained how an estimated graph $H$ may contain additional edges, yet still yield the same set of intervention distributions as $G$, and therefore a low or even zero SID, while the same situation is strongly penalized by SHD.
>
> As a result, **it is possible for two structurally different graphs to produce similar SID scores**, essentially because SID depends on the induced interventional factorizations, not on topological similarity per se.
>
> ### 2. What happens in Figure 2?
>
> Based on the definition, one can see that for a graph of $d$ nodes, the SID is upper bounded by $d(d-1)$. The Artic dataset in Figure 2 has $12$ nodes. What happens is that all experts score the worst SID precisely at $132$, meaning for every pair of nodes, there is at least one observational distribution Markov w.r.t $G$ with which the predicted interventions do NOT align.
>
> ---
> We hope the explanations clear up the doubts. We have released the codes in the supplementary materials for full reproducibility. We have also saved the experts' predicted graphs in our experiments, which we can make available if needed. We do not provide here because we are unsure whether external links are permitted in the response. If allowed, we would be happy to provide them as evidence.
>
>
> *Reference:*
>
> Peters, J., & Bühlmann, P. (2013). Structural intervention distance (sid) for evaluating causal graphs.

---

> > ### Comment · Reviewer_94Bz · 2025-11-26
> >
> > Thank you for the explanation and for pointing out the other figures where SID has some variability - this addresses my concerns and I don't think there's an error. The code is unnecessary, this explanation is sufficient.
> >
> > I think many readers will have a similar reaction, so I highly recommend addressing this point in the paper or even removing the SID plot from Figure 2 (since it doesn't provide any information beyond the fact that all experts have the worst possible SID, which you can just state in the text). Directly explaining this point will avoid potential confusion/skepticism.

---

> > > ### Author Response · Authors · 2025-11-26
> > > **Thank you!**
> > >
> > > We have well noted the comments and will include the explanation in the final version. Thank you again for the active engagement. We truly appreciate the your time and very useful feedback. Should there be anything else we should pay attention to, please kindly let us know.

---

### Official Review · Reviewer_kwwE · 2025-11-01

**Soundness:** 2
**Presentation:** 3
**Contribution:** 3
**Rating:** 6
**Confidence:** 3

**Summary:**

This work aims to address three fundamental limitations of ensemble-based causal discovery. They claim existing methods are largely heuristic, lack theoretical guarantees and guidance on how ensemble design choices affect performance. The paper proposes a principled voting-based framework for structural ensembling, establishing conditions under which the aggregated structure recovers the true causal graph. They concluded that their analysis yielded a theoretically justified weighted voting mechanism that informs optimal choices regarding the number, competency, and diversity of causal discovery experts in the ensemble. Their model was tested on both synthetic and real world data to verify effectiveness and robustness.

**Strengths:**

S1: Extends the novel estimation algorithm based on optimal transport in a clearly non-trivial way.

S2: Solid mathematical formulation with proofs or guarantees, with clear connection between intuition and formal results.

S3: Comprehensive empirical validation with experiments on diverse, well-chosen benchmarks

**Weaknesses:**

W1: The solution approach is hard to follow with very limited intuitive explanation of complex maths used.

W2: It is not clear how the paper addressed the well-defined problems with existing ensemble methods.

W3: Authors did not share the code therefore reproducibility is questionable.

W4: The problem the paper is trying to solve (existing methods are largely heuristic, lack theoretical guarantees) is not clearly motivated and lacks practical importance.

**Questions:**

Q1: When learning the structure from noisy experts or base learners, how is the competence of an expert measured, before creating the competence transition matrix?

Q2: Different experts/base methods use different parameters to identify causal links, how did you reconcile and consolidate these parameters?

Q3: The proposed solution using the Optimal Transport framework is not clear.

Q4: Can this proposed approach identify nonlinear causal links, i.e often found in dynamic systems like climate systems with feedback loops?

Q5: Did you encounter situations were weighted linear voting rule say false positives, i.e the experts voted the presence of an edge in a graph when there exist no edge in the true graph? How did you reconcile such situations?

---

> ### Author Response · Authors · 2025-11-18
> **Our responses (1/2)**
>
> Thank you for your valuable time and the support of an acceptance.
>
>
> It appears that there is a misunderstanding about our approach. It is possibly due to the clarity issues in our presentation, for which we apologize. With the extra page allowed in the camera-ready version, we will move additional content from the appendix into the main text to improve readability.
>
> &nbsp;
>
> ## Significance of the problem
>
>
> **First, we strongly believe that our problem is practically important.** We provide justifications as follows:
>
>
> 1/ Causal discovery aims to recover the underlying causal structure from given data. A large number of causal discovery algorithms exist, but each relies on its own set of modeling assumptions that are often unverifiable in practice. As a result, applying different methods to the same dataset can yield markedly different causal graphs. This raises a fundamental question: *which algorithm should we trust?*
>
> 2/ This challenge motivates ensemble-based approaches: shifting the focus from selecting a single "best" algorithm to aggregating the predictions of many - a form of leveraging "wisdom of the crowd".  This immediately leads to important questions: *How should we combine these graphs? Under what conditions on the experts can we guarantee that the aggregated graph is correct?*
>
> 3/ This lack of theoretical guarantees is the key limitation of existing ensemble causal discovery methods. We believe this poses **a critical issue for real-world, especially high-stakes applications.** Imagine a pharmaceutical company that uses causal discovery methods to identify causes of a disease before committing a multi-million-dollar investment in drug development. Without understanding when and why a method succeeds, we cannot meaningfully assess its risks to make well-informed decisions.
>
> We strongly argue that a heuristic ensemble method cannot substantiate itself as an adequate solution to the original problem, as it offers no more reliability in causal discovery than the individual algorithms it seeks to combine.
>
> &nbsp;
>
> ## Our solution approach
>
> We now briefly outline our approach to solving it, which may altogether clarify part of the reviewers' questions.
>
>
> Our work precisely aims to fill the mentioned gap. Here we adopt a Bayesian perspective and explicitly model the generative process that gives rise to the predictions produced by each causal discovery algorithm (treated as an “expert”).
>
> Our framework is based on Bayes voting - a weighted linear voting rule that accounts for two key information: (1) a prior probabilities over possible graph configurations, and (2) experts' compentency or reliability, represented by the competence transition matrices. Each row in the matrix quantifies the probability that an expert predicts (in)correctly, given  a particular realization of the true graph
>
> Given these quantities, the Bayes voting rule provides an optimal aggregation principle under standard Assumption 1.
> More importantly, this formulation allows us to analyze the probability that the aggregated graph is close to the true graph, along with the conditions under which recovery can be achieved with high probability. These results are critical to ensuring reliability of a method since in practice the true graph is unknown.

---

> ### Author Response · Authors · 2025-11-18
> **Our responses (2/2)**
>
> ## Other comments/questions
>
> **W3: Code release is available!**
>
> It is mentioned in Line 1490 that we do have released codes. They can be found in the supplementary material.
>
> &nbsp;
>
> **Q1: How is the competence of an expert measured, before creating the competence transition matrix?**
>
> As explained, the compentence of an expert is directly quantified by the competence transition matrix. These matrices are parameters of the data-generating model and unknown. To apply Bayes voting, we need to estimate these quantities. Section 4 in the paper details how this is done, to what extent the true parameters can be recovered, and how robust the decision rule is when the estimates are imperfect.
>
>
> &nbsp;
>
> **Q3: The proposed solution using the Optimal Transport framework is not clear.**
>
> In relation to Q1, it is seen that our problem reduces to estimating the competence matrices. Now we are dealing with a classic statistical task of parameter estimation. We here adopt the standard approach of minimum distance estimation: to seek the parameters that minimizes some distance $D$ between the data-generating model and empirical data distribution, where data refers to the observed experts' graph predictions (or called voting profiles).
>
> The proposed framework simply arises from the fact that we choosing $D$ as the optimal transport distance (defined in Eq. 2), which admits a tractable form as shown in Vo et al. (2024). This form also gives rise to a likelihood-free method that supports amortized optimization, which serves our purposes well.
>
> Due to space limit, a dedicated discussion of this formulation is deferred to Appendix D.2. In that section, we briefly recap the setup, outline the optimization objective and include illustrative pseudo-code. For the technical proofs, readers are referred to the original paper.
>
>
>
> &nbsp;
>
> **Q2:  How did you reconcile and consolidate different parameters?**
>
> Our framework conveniently bypasses this complexity by operating solely on the predicted graphs from the experts. The input to our method is simply the set of experts' predicted graph and it outputs a final aggregated one. The specification of the competence matrices provides an elegant way to capture the heterogeneity arising from different strategies and parameter settings across experts, enabling direct comparison of the algorithms' effectiveness.
>
>
>
> &nbsp;
>
> **Q4: Can this proposed approach identify nonlinear causal links, or feedback loops?**
>
> Yes, the theoretical results are applicable to generic structure learning problem and do NOT assume linearity or acyclicity. In fact, most our experiments were conducted on non-linear and real-world causal models.
>
> &nbsp;
>
>
> **Q5: Did you encounter situations were weighted linear voting rule say false positives? How did you reconcile such situations?**
>
> To the first part of the question: yes, it is possible to have false positives.
>
> However, it is important to note that **in practice, we do not know the true graph**. Hence one cannot tell whether a predicted (sub-)graph is correct or not.
>
> This is exactly why our theoretical results play an extremely important role: they chacterize the conditions on the number of experts and their competency (quantitatively via the transition matrices) that one needs to **maximize the probability** that the aggregated graph matches the true one. The choice of experts involved is the factor we can control, and in practice one can use simulated or benchmark data to evaluate the quality of a selected group of experts, for example how frequently they produce false negative/positive predictions. This enables us to identify and select experts that performs most effectively and reliably for the task.
>
>
> ---
> ### Thank you!
>
> We sincerely appreciate the reviewer’s time and effort in evaluating our work. We hope to have adequately addressed your concerns. Kindly let us know if you have any further questions.

---

### Official Review · Reviewer_239W · 2025-11-08

**Soundness:** 3
**Presentation:** 1
**Contribution:** 3
**Rating:** 6
**Confidence:** 2

**Summary:**

This paper proposes an ensemble framework for causal discovery task. More specifically, the authors analyze the conditions under which the predicted graph matches the true graph. They also provide theoretical justification for different size, competency and diversity levels of causal discovery experts. Finally, they show performance on synthetic and real-world datasets.

**Strengths:**

The proposed method offers a strong contribution as it provides practitioners with a guide on how to configure ensemble strategies while combining outputs of multiple causal discovery experts for optimal performance. It’s interesting how the authors avoid searching the exponential space by decomposing the graphs into sub-structures.

**Weaknesses:**

Below I provide my comments:

## **Major**
* The paper is little convoluted. The steps of the proposed algorithm can be organized in a better way.   Section 3.1 is notation-heavy, with lots of notations, making it quite hard to keep track of all notations.
* A simulation of the voting mechanism is needed, i.e., how $\hat{Y}(w)$ is constructed and what its relation is with $\hat{G}(\sigma, \mathcal{S})$, how the transition matrix looks like and how it contributes.
* Some of the discussion from Appendix D can be transferred to the main paper (Section 2) to provide readers with a better understanding of the concepts. Many concepts in the main paper cannot be connected without proper preliminaries or background.
* How parameter estimation with optimal transport works is not clear. The authors should focus more on explaining the main algorithm intuitively.
* The authors did not compare with other existing distance-based or voting-based structural aggregation algorithms. For example, the authors cited papers such as Mio et al. (2025), Guo et al. (2021), (Tang et al., 2019), Aslani & Mohebbi, 2023. They should compare their performance with these ensemble frameworks.
* In figure 2, are the causal experts (black) executed once to record their performance? In such case, the proposed ensemble framework should perform better always. Would that be a fair comparison? I would request the authors to point me if I misunderstood this.

---

## **Minor**
* There are too many versions of the graph: $G$, $\mathbf{G}$, $\mathcal{G}$, $\tilde{G}$, $\tilde{G}_i$, $\hat{G}$.
* Some visualization or example of $T$, $\theta$ would be nice.
* Line 140: shouldn’t the incorrect prediction be a sum over all $q_{i, G_j | G_1}$?
* It is not clear intuitively what the bias term indicates in Equation 3.
* Lines 200–201: should it be an equal sign?

**Questions:**

Below I share my questions.

## **Question**
* What does convergence mean in the “converging to the correct decision”?
* Line 346: at least $2m - 1$ experts are needed. For a three-node directed graph having three features {(v1, v2), (v2, v3), (v1, v2)}, what will be the value of *m* and how many experts do we need?  Is it number of experts always feasible in real-world setups?
* If $\mathbf{G}_1$ is the true graph, does the proposed algorithm work with a vector of probabilities $T_i[:, G_1]$ instead of the matrix?
* How large is the competence transition matrix $T_i$? Is it of size $|\mathcal{G}| \times |\mathcal{G}|$? Is it very large and hard to learn?
* In Condition 1, the authors consider that two rows are distinct element-wise. To my understanding, each index of $T_i$ represents $T_i := [{P}(\tilde{G}_i = G_j |  G = G_k,  D)]$. A row represents the probability distribution of different true graphs while the prediction is fixed. On the other hand, a column represents the distribution of different predictions while the truth is fixed.
Now, when the true graph is different, if the algorithm provides different probability distributions over the predicted graphs, that should indicate the algorithm’s capability to distinguish between two graphs, right? Why then do the authors use rows in condition 1 and not columns of the matrix?

---

> ### Author Response · Authors · 2025-11-18
> **Our responses (1/2)**
>
> Thank you for the your valuable time and very insightful questions raised. We kindly categorize them and provide responses as follows:
>
>
> ## Questions about methodology
>
> **1. What does convergence mean in the "converging to the correct decision''?**
>
> It means the probability that the aggregated graph, under Bayes voting rule, coincides with the true graph goes to $1$ w.r.t factors like number of experts $n$ approaching infinity, while keeping the others fixed.
>
> &nbsp;
>
> **2. For a three-node directed graph having three features, what will be the value of $m$ and how many experts do we need? Is it number of experts always feasible in real-world setups?**
>
> When there are no latent confounders, for each edge-level feature, one can consider a state space: $(1: v_i \rightarrow v_j, 2: v_i \leftarrow v_j, 0: \text{no edge})$, where $m = 3$. Then, our theory suggests in the general case, $n = 5$ experts are sufficient to identify the competence matrices and $n = 3$ are necessary i.e., the parameters cannot be identified with fewer than $3$. In special case when all experts are fully informative, meaning the matrices has full rank, then $n = 3$ are both necessary and sufficient. This number is entirely feasible and exactly what we considered in our experiments.
>
> &nbsp;
>
> **3. How large is the competence transition matrix $T_i$? Is it of size $|\mathcal{G}| \times |\mathcal{G}|$? Is it very large and hard to learn?**
>
> At the graph level, it is indeed $|\mathcal{G}| \times |\mathcal{G}|$ and very large. This is what motivates our development from Section 3.2, shifting from graph-level aggregation to feature-level aggregation.
>
> For a graph with $d$ nodes, one competence matrix at the graph level induces at most $2^{d^2} \times 2^{d^2}$ parameters. Meanwhile, a choice of level $l$ gives rise to a feature space $\Omega$ of size $\binom{d}{l}$, where $\Omega$ refers to the set of features at a given decomposition level. In this case, the number of parameters per expert becomes $\binom{d}{l} \cdot m^{2}$.
>
> Because $l$ is under our control, we choose a level that is computationally feasible. In this work, we show that $l = 2$ - i.e., features corresponding to individual edges - is not only tractable but also sufficient and, feasibly ideal for aggregation. Hence, the remainder our paper focuses on establishing the optimality and robustness of  feature-level aggregation.
>
> &nbsp;
>
> **4. A row represents the probability distribution of different true graphs while the prediction is fixed. On the other hand, a column represents the distribution of different predictions while the true is fixed. Why then do the authors use rows in condition 1 and not columns of the matrix?**
>
> We believe there is some misunderstanding about our setup. It is in fact the other way around: each **row** corresponds to one possible configuration of the **true graph**, while each **column** corresponds to one configuration of a **predicted graph**. Each $T_i$ is a row-stochastic matrix. Our theoretical results adopt the row convention throughout.
>
> &nbsp;
>
> **5. Does the proposed algorithm work with a vector of probabilities of $T_{i,|1}$ instead of the matrix?**
>
> Our work begins with the Bayes voting rule used to aggregate graph predictions from multiple experts, where the decision rule is defined w.r.t the competence transition matrices as shown in Eq. (4).
>
> Our theoretical analysis proceeds in two main directions. First, we characterize the conditions on the matrices to determine the extent to which the true graph can be recovered. To apply the Bayes voting rule, the problem thus reduces to estimating each expert’s competence transition matrix. This motivates the use of an optimal transport algorithm for estimation. The Wasserstein-based framework is a generic approach to estimating latent variable models with well-defined parameters.
>
> In this context, we presume that the question pertains to whether the recoverability conditions can be established solely on the row vector $T_{i,|1}$ (i.e. probability vector corresponding to the true label). This setting in fact corresponds to the classical asymptotic results in Jury’s theorem for majority/plurality voting (see Fey, M 2003). Hence, the nature of the required parameters depends on the chosen voting rule. Bayes voting rule is a more general approach, which explicitly models uncertainty in the true labels and subsumes plurality voting as a special case (see Line 160).
>
>
> *Fey, M.: A note on the Condorcet Jury Theorem with supermajority voting rules. Soc Choice
> Welfare 20(1), 27–32 (2003).*

---

> ### Author Response · Authors · 2025-11-18
> **Our responses (2/2)**
>
> ## Questions about experiments
>
> **1. Comparisons with existing structural aggregation algorithms**
>
> As mentioned in Lines 1487-1490, only the rank-based method proposed by Malmi et al. (2015) provides publicly available code. We were unable to locate released code for the other ensemble-based methods.  Among these, Aslani and Mohebbi (2023) and Tang et al. (2019) include relatively detailed descriptions. We attempted to reproduce their algorithms based on the information provided in the papers but were unsuccessful, respectively due to their usage of complex optimization solvers and intricate distributed data workflows.
>
> &nbsp;
>
> **2. In figure 2, are the causal experts (black) executed once to record their performance?**
>
> The causal experts are NOT executed once. They were evaluated on various bootstrap subsets of data. For methods subject to initialization randomness like NOTEARS, DAGMA), they are also run 10 different seeds each subset. The average results have taken into account the randomness of the algorithm.
>
> &nbsp;
>
> ## Other Comments/Questions
>
> **1. About notation/presentation**
>
> We apologize for the difficulty in following our notations. We have strived to make the main text self-contained, and a notation summary was provided in Appendix B for quick reference. With the extra page allowed in the camera-ready version, we will move additional content from the appendix into the main text to improve clarity and ease of reading.
>
> &nbsp;
>
> **2. About  $\hat{Y}(\omega)$ and $\hat{G}(\Omega, \mathcal{S})$**
>
> For each feature, we define a corresponding competence matrix that characterizes each expert’s ability to predict that feature. This matrix is obtained by marginalizing the global (graph-level) competence matrix.
>
>
> Let $\omega$ denote a feature, for example, an edge $(i,j)$. Viewing the entire graph as a joint realization of all its features, we aggregate the predictions of individual features to obtain the final predicted graph, where the predicted value of a feature $\omega$, denoted by $\hat{Y}(\omega)$ is obtained by applying Bayesian voting on the corresponding marginal (feature-level) competence matrix. Given a predefined feature space $\Omega$ and feature state space $\mathcal{S}$, this final predicted graph is denoted by $\hat{G}(\Omega, \mathcal{S})$.
>
> &nbsp;
>
> **3. How parameter estimation with optimal transport works is not clear.**
>
> Due to space limitations, a detailed discussion of the optimal transport (OT) formulation is provided in Appendix D.2. In that section, we briefly recap the setup, outline the optimization objective and include illustrative pseudo-code for readability. For the technical proofs, readers are referred to Vo et al. (2024).
>
>
> &nbsp;
>
> **4. Line 140: shouldn't the incorrect prediction be a sum?**
>
> Apologies if there was a confusion. The notation $q_{i,G_j|G_1}$ was intended to denote the probability of making a **particular** wrong prediction $G_j$. Our results take individual probabilities into account.
>
>
> &nbsp;
>
> **5. It is not clear intuitively what the bias term indicates in Equation 3.**
>
> In a decision-making setting, you can think of the bias term as a prior belief or institutional preference for certain options over others, independent of the experts’ votes. In some cases, it also serves as a calibration factor—adjusting for options that are popular but systematically suboptimal (e.g., offering low utility to the community).
>
> &nbsp;
>
> **6. Lines 200-201: should it be an equal sign?**
>
> Apologies for the typo. Yes, it should be an equal sign.
>
>
> ---
> ### Thank you!
> We sincerely appreciate the reviewer’s time and effort in evaluating our work. We hope to have adequately addressed your concerns. Kindly let us know if you have any further questions.

---

### Author Response · Authors · 2025-12-01
**Summary of Rebuttal**

As the rebuttal period concludes, we thank the reviewers for your time and unanimous support of an acceptance. We especially appreciate Reviewer 94Bz’s active engagement and are pleased that all raised concerns have been addressed. We provide below a concise summary of the rebuttal for quick reference.

## Contribution in one line

Our paper introduces a principled Bayes voting framework to aggregate predicted graphs from multiple causal discovery experts.

## Key results in paper
* **Sec 3.1:** Characterization of the probability that the aggregated graph under Bayes voting recovers the true graph.
* **Sec 3.2:** Equivalence and reduction of global graph-level aggregation to local feature-wise aggregation.
* **Sec 4:** Robustness of Bayes voting under noisy estimates, parameter estimation for feature-level Bayes voting, identifiability and consistency of the parameters.
* **Appendix D.1:** Theoretically grounded guidelines for practical design of optimal ensembles.

### Clarifications about & where to navigate:
* Motivation, significance, and solution approach → Responses to Reviewer kwwE (Part 1)
* Key notations and problem setup → Responses to Reviewer mcKH (Part 1)

&nbsp;
## Common Questions & Brief Answers

**(Reviewers 239W & mcKH) How does reducing graph-level to feature-level aggregation helps efficiency?**

→ Direct graph-level Bayes voting is intractable: a graph of $d$ nodes requires scoring all $2^{d^2}$ possible graphs and handling $2^{d^2} \times 2^{d^2}$ competence matrices. We thus propose to decompose the graph into local features of level $l$, each with $m$ connectivity states. Bayes voting is then applied feature-by-feature using the marginal priors and individual $m \times m$ transition matrices.

A choice of level $l$ gives rise to a feature space $\Omega$ of size $\binom{d}{l}$, where $\Omega$ refers to the set of features at a given decomposition level. The number of parameters in estimating the matrices (per expert) reduces to $\binom{d}{l} \cdot m^{2}$, which is manageable for small $l$. We show $l=2$ i.e. edge-level features is not only tractable but also sufficient and, feasibly ideal for aggregation.


**(Reviewers 239W & kwwE) How does parameter estimation via optimal transport work?**

→ For parameter estimation, we adopt the classic approach of minimum distance estimation: to seek the parameters that minimizes some distance $D$ between the data-generating model and empirical data distribution, where data refers to the observed experts' graph predictions. The optimal transport framework simply arises from the fact that we choose $D$ as the optimal transport distance (defined in Eq. 2). The formulation is tractable, likelihood-free, enables efficient amortized optimization across features, and scales well to large ensembles. Full details, objective, and pseudo-code are provided in Appendix D.2.


**(Reviewers kwwE & 94Bz) Can the method handle nonlinear relationships or feedback loops?**

→ Yes. Our theoretical results apply to generic structure learning without assuming linearity or acyclicity. When acyclicity is required, our method accommodates standard post-hoc practice of thresholding low-confidence edges.

&nbsp;
## Revisions

**Revisions already made:**
* Fixed typo in Definition 1
* Revised Assumption 1 and Sec 4 notations to explicitly condition on input $D$

**Revisions to be made with an extra page in camera-ready version**
* Move parts of Appendix D and E to the main text
* Expand on the paragraph at the start of Section 3 to clarify the setup (per part 1 response to Reviewer 94Bz)
* Add recommendations in the Conclusion for handling acyclicity constraint and non-i.i.d. graph samples

---

### Meta-Review · Area_Chair_Y74a · 2025-12-05

**Summary:**

The paper provides the first theoretically grounded ensemble method for causal discovery. The main contribution is in providing guarantees of the recovering of the true graph depending on the number, diversity and competence of experts in the ensemble.

The method boils down to multiple and independent Bayes votes on sub-graphs (called features), and applying optimal transport for estimating the expert transition matrices/competencies. Dividing the problem into subproblems (the subgraphs) is what makes the method scalable and tractable for structure learning. Apart from this point, no other characteristic of the method is specific to causal discovery, and constraints on the estimated graph (such as acyclicity) would be incorporated post-hoc.

**Reviewer Concerns:**

Still standing concerns:
1. Limited comparison with existing structural aggregation algorithms: Only results for the rank-based approach of Malmi et al. (2015) are reported in the paper. The authors could not reproduce the code for the methods of Aslani and Mohebbi (2023) and Tang et al. (2019).
2. Convoluted presentation: The presentation of the paper could be improved by providing an example of the method execution on a small graph, by providing intuition for the mathematical derivation, and by simplifying the notation.
3. Limited novelty wrt statistical aggregation results: The paper boils down to a novel application (to causal discovery) of existing results in the statistical learning theory.

Resolved major concerns:
Lack of practical importance, expert conditional-independence given the ground-truth graph and the input/data

**Reviewer Scores:**

I don't think the scores would have significantly changed.

---

### Decision · Program_Chairs · 2026-01-26

Accept (Poster)